# EMERGENCE OF ALIGNMENT AND LOCAL ELASTICITY IN TWO-LAYER NEURAL NETWORKS

## ABSTRACT

Investigating phenomena such as Alignment and Local Elasticity is essential for understanding feature space of Neural Networks and enhancing performance across a wide range of tasks. In this context, we investigate the emergence of these phenomena in two-layer neural networks performing a classification task. This paper reveals Alignment and Local Elasticity emergence condition after one step of training are identical. In particular, we demonstrate that intra-class features are more aligned when the inner product of their mean and the covariance of the training data-label i.e. *train-unseen similarity* is large, with stronger Local Elasticity occurring under this condition. We validate our theory through experiments with a two-layer network showing that both Alignment and Local Elasticity improve as the train-unseen similarity increases. Furthermore, we claim that our analysis provides both theoretical and practical insights into the relationship between train-unseen similarity, alignment, and the improvement of clustering performance on unseen data for neural networks trained on similar domain data. This is supported by experiments, including a multi-layer CNN setup and detailed discussions. Specifically, we show that higher train-unseen similarity improves Recall@1 in two-layer networks and that Alignment and Recall@1 exhibit a positive correlation in metric learning. We also present novel techniques for deriving operator norm bounds of non-centered Sub-Gaussian matrices, extending conventional regression analysis with standard Gaussian assumptions to the binary classification setting.

## 1 INTRODUCTION

Representation learning has been advanced thanks to the introduction of deep learning (Goodfellow et al., 2016; Bengio et al., 2014), surpassing the generalization performance of the conventional machine learning techniques (Bach, 2016; Sánchez & Perronnin, 2011). However, the underlying feature training dynamics that enable deep network to learn more generalizable features An et al. (2023); Radford et al. (2021) remain unclear, prompting studies aimed at theoretically resolving this issue (Damian et al., 2022; Abbe et al., 2021). To understand the learning dynamics, we argue that the following *three challenges* must be addressed: First, under what conditions does learning occur (He & Su, 2019)? Second, to what extent does learning take place under those conditions e.g. Local Elasticity (Dan et al., 2023)? Third, how are the resulting features structured after learning e.g. Alignment (Wang & Isola, 2022; Beaglehole et al., 2024)?

One approach to addressing this challenge is the Neural Tangent Kernel (NTK) (Jacot et al., 2020). NTK studies have explored the alignment structure of features and the concept of Local Elasticity in NTK (Seleznova et al., 2023; Chen et al., 2020; Atanasov et al., 2021; Shan & Bordelon, 2022). However, the NTK operates under a lazy training regime, and its empirical variants exhibit significant discrepancies in modeling neural networks (Chizat et al., 2020; Vyas et al., 2022; Yang & Hu, 2022). This makes it challenging to conduct a theoretical analysis of feature learning without additional assumptions, such as whitened data, feature block structure, or label awareness. On the other hand, Conjugate Kernel (CK) approaches have been studied (Pennington & Worah, 2017; Fan & Wang, 2020; Benigni & Péché, 2022), with a key distinction from NTK in their ability to facilitate the analysis of feature learning (Ba et al., 2022; Dandi et al., 2023; Moniri et al., 2024), thereby offering a framework for explaining generalization performance. Building on these properties, we claim that the CK feature learning model not only explains the generalization performance on test

data from the same distribution as the training data but also offers a structural analysis of features derived from data sampled from unseen distributions that differ from the training distribution.

Deep representations are used in problems where the distributions are "unseen" or "almost similar but different downstream task data" such as in transfer learning (Yosinski et al., 2014; Weiss et al., 2016; Bozinovski, 2020; Galanti et al., 2022), linear probing (Kumar et al., 2022; He et al., 2020; Kornblith et al., 2019), and metric learning (Huang et al., 2024). In these applications, learned features remain effective for data outside the training distributions, even though statistical theories suggest that perfect extrapolation is not attainable (Balestriero et al., 2021; Kang et al., 2024; Armstrong, 1984). Therefore, it is essential to investigate such a problem to advance the deep learning theory. Specifically, this paper investigates the emergence conditions of *Alignment Structure* and *Local Elasticity* for data from unseen distributions to address the *three challenges* mentioned above.

## 1.1 RELATED WORKS

**Conjugate Kernel**  Many works (Benigni & Péché, 2021; Louart et al., 2017; Hu & Lu, 2022; Goldt et al., 2020) study the CK, which models neural networks and enables the analysis of the structure of the first layer in two-layer networks after the Gradient Descent. Ba et al. (2022) analyze regression tasks in the teacher-student setups to study feature learning in the proportional regime. They demonstrate that neural networks exhibit superior performance compared to linear models, particularly at higher learning rates since the feature learning reflects the structure of the teacher's weights. Moniri et al. (2024) utilize Hermite decomposition to analyze how nonlinear features are learned based on the polynomials. Ba et al. (2023) theoretically compute the condition when neural networks learn the low-dimensional structure of the dataset with spiked covariance Gaussian distribution data. Bietti et al. (2022) analyze the loss landscape and sample complexity which enables us to learn a single-index model. Ba et al. (2022); Moniri et al. (2024); Ba et al. (2023); Bietti et al. (2022) argue that in teacher-student settings for solving regression problems with centered Gaussian distributions, neural network features can learn the structure of the teacher, thereby improving generalization performance. Unlike these studies, we extend the two-layer network setting to classification with non-centered Sub-Gaussian distributions and examine the phenomena that arise when the network is exposed to input drawn from a distribution different from train distributions. To the best of our knowledge, our work provides the first analysis of non-centered training distributions. We believe this contributes a framework that can be further utilized in analyzing classification.

**Alignment Structure**  Alignment has been used with various definitions in the study of neural network structure and applications. For instance, there are studies on the following: intra-class feature alignment (Deng et al., 2022; Wang & Isola, 2022), feature-weight alignment (Papyan et al., 2020), feature-label alignment (Shan & Bordelon, 2022; Atanasov et al., 2021), feature-gradient alignment (Ziyin et al., 2024). We are interested in **intra-class feature alignment**. Therefore, in the following, "**Alignment**" will refer to **intra-class feature Alignment**. As training progresses, the Alignment where features of a given class align towards a single point has been observed (Papyan et al., 2020). It is linked to generalization performance on unseen distributions (Liu et al., 2018). For example, some works claim that increasing intra-class alignment of train distributions with inductive bias improves task performance on unseen distributions, particularly in metric learning. (Wang et al., 2018; Liu et al., 2017). However, to the best of our knowledge, the conditions under which alignment strongly emerges have not been established. In this work, we demonstrate that the emergence of higher alignment is governed by the relationship between the training data and the input data distribution. Specifically, we show that in a binary classification problem, where $\beta$ represents the covariance vector of the training data and labels, and $\mu$ is the mean of the unseen class conditioned distributions, a larger inner product $\left| \beta^\top \mu \right|$ i.e. **train-unseen similarity** leads to higher alignment, thereby providing a theoretical basis for alignment.

**Neural Collapse (NC) and Unconstrained Layer-Peeled Model (ULPM)**  research is related to intra-class feature and feature-weight alignment. NC (Papyan et al., 2020) addresses the phenomena that occur with the features and the weights of the classifier head at the final stages of classifier training. At this stage, phenomena related to alignment occur: First, Variability Collapse, i.e. intra-class feature alignment, and Second, self-duality, i.e. feature-weight alignment. Several studies propose the ULPM to analyze NC treating features and weights as unconstrained free variables (Ji et al., 2022; Tirer & Bruna, 2022; Zhu et al., 2021; Fang et al., 2021). However, ULPM, unlike

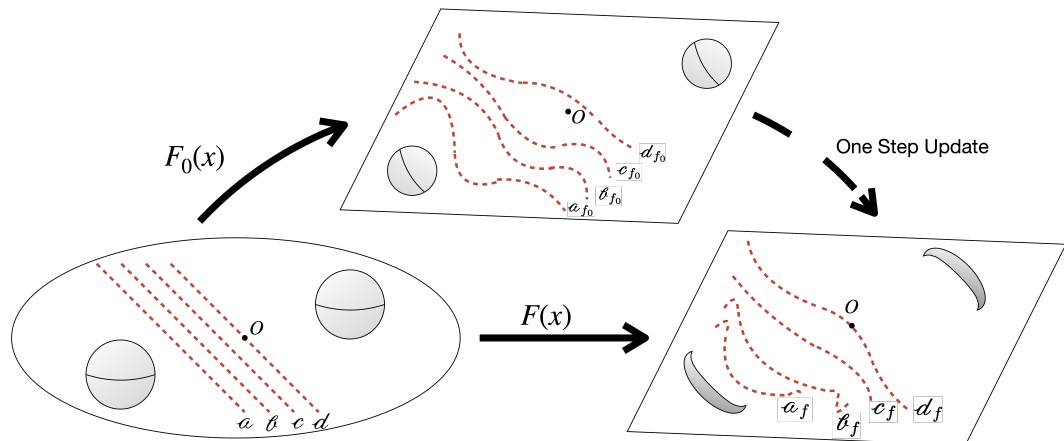

Figure 1: Emergence of Alignment and Local Elasticity: The Neural Networks feature $F(x), F_0(x)$ from data points surrounding the training data ( i.e. unseen data) are influenced by the gradient step, leading to both phenomena. We denote unseen data as $a, b, c, d$ and two classes of training data are represented as a sphere. Notably, distribution $a$, being closer to the training data, undergoes stronger Alignment and Local Elasticity i.e. the intra-class inner product is enlarged, and the features undergo substantial movement during this single step, compared to other distant distributions. $b, c, d$.

the CK model we use, assumes the features as free variables, which limits its ability analysis about input the data distribution and, consequently, prevents studying the structure of the features. This motivates and provides the need to explore internal features using CK.

**Studies on the concept of Local Elasticity (LE)** have been established after observing that data points closer to the training samples are updated more significantly than those farther away (He & Su, 2019). Thus, Local elasticity has been informally described using terms such as "similarity/closeness". In other words, it is argued that the greater the "similarity" between the training data and the input data, the higher the elasticity of the feature. Subsequently, in He & Su (2019), the elasticity score was formalized as a metric to quantify this informally defined notion of "similarity". Meanwhile, there have been attempts to theoretically understand LE. Zhang et al. (2021) model the learning process of neural networks using SDE to verify its occurrence, but they have a limitation that actual neural networks are not utilized as our CK modeling. Dan et al. (2023) sort training steps into two phases by whether LE occurs or not using Gradient Flow, but they only empirically observed the basic condition of LE i.e. feature of "similar" sample is updated more, without engaging in theoretical exploration. However, with theoretical assumption and analysis, we establish that this similarity can be measured and expressed as **train-unseen similarity**.

Additional related works are discussed in Appendix C.

### 1.2 OVERVIEW

This section provides basic definitions and informal Theorems of the results of Alignment and Local elasticity, which will be detailed in section 4. The phenomenon described here is also illustrated in Figure 1. Let $\theta$ be the set of every randomly initialized parameter of a neural network, let $\mathbf{d}, \mathbf{N}$ denote the dimensions of the data space and feature space, respectively, for $x \in \mathbb{R}^{\mathbf{d}}$ denote $F(x) \in \mathbb{R}^{\mathbf{N}}$ is trained feature and $F_0(x)$ is initialized feature and $c$ is given class conditioned distribution. Feature represents a network output obtained by peeling off the last task layer. The **Alignment score** and **Elasticity score** are defined as follows:

**Definition 1.1** (Alignment score)**.** The **Alignment score** is defined as the expected inner product between the features $F(x)$ for two i.i.d. samples of $c$ : $\mathbb{E}_{x,x'\sim c,\theta}[F(x)^{\top}F(x')]$.

**Definition 1.2** (Elasticity score similar[1] to He & Su (2019)). The **Elasticity score** is defined as the expected $L2$ distance between $F(x), F_0(x)$ for sample of $c$: $\mathbb{E}_{x \sim c, \theta}[\|F(x) - F_0(x)\|^2]$.

These two definitions are informally expressed as the Theorem below, which is an approximation with high probability in the proportional regime for a two-layer neural network after one step training and a Gaussian assumption of given class conditional distribution $c$.

**Theorem 1.3** (Alignment and Elasticity score (Informal of Theorem 4.2, 4.3)). *Let $\mathbf{n}$ be the number of data points. Assume $x, x' \sim \mathcal{N}(\mu, \Sigma)$ be i.i.d random vectors drawn from the arbitrary class conditional distribution given mean $\mu \in \mathbb{R}^{\mathbf{d}}$ and Covariance $\Sigma \in \mathbb{R}^{\mathbf{d} \times \mathbf{d}}$ and the network allows for Hermite expansion. Let $\beta \triangleq \frac{1}{n\sqrt{N}} X^\top y$ from given training datasaet $(X, y)$. Then **train**($\beta$)-**unseen**($\mu$) similarity $|\beta^\top \mu|$ and $\beta^\top \Sigma \beta$ is governing the **Alignment** and the **Elasticity score**.*

Following Theorem, **Alignment** and **Elasticity score** approximately increase as **Train-unseen similarity** $|\beta^\top \mu|$ and $\beta^\top \Sigma \beta$ grow i.e. , with fixed covariance $\Sigma$, both scores are approximately polynomial to $|\beta^\top \mu|$, which is the similarity between the training sample distributions and the arbitrary class data distributions.

It can be interpreted that the closer the unseen distribution is to the training data ( i.e. , higher **Train-unseen similarity**), the stronger the effect of Local Elasticity (LE) becomes, and leading to a stronger Alignment of features. These implications can be observed in section 4, where comparable formulae for the two phenomena are derived, demonstrating their simultaneous occurrence and correlation.

For theoretical analysis, we define two-layer networks with elementwise activation function that allows Hermite decomposition to decompose a one step trained feature function into initialized features and polynomial functions. This decomposition is explained thoroughly in subsection 3.1 and subsection 3.2. The decomposed feature is analyzed using unseen data distributions assumed to follow Gaussian distributions.

This paper also verifies the following supplementary contributions during our theoretical analysis. We expand the previous two-layer network analysis method, which is based on regression tasks with standard Gaussian train distribution into binary classification with non-centered Sub-Gaussian distribution. This assumption makes two-layer model available to analyze classification problems in further works or any non-centered Sub-Gaussian training data, which is more discussed in section 3.

Finally, we conduct experiments that empirically verify our analyses using a synthetic dataset where classes of evaluation set are consecutively distant from the training set in section 5.

## 2 PROBLEM STATEMENTS

**Notations** Let $\|\cdot\|$ be $L^2$ or the operator norm. Let $\odot$ be the Hadamad product. Let $A^{\circ k}$ be the Hadamad power. Let $C, c > 0$ be absolute constants, and let $\kappa \in \mathbb{R}$ be a constant that may change from line to line. Define $[d] \triangleq \{1, 2, \cdots, d\}$. Let $\mathbf{1}_{\text{condition}}$ be 1 if the condition is true and 0 otherwise. The operator $\text{diag}(\cdot)$ creates a matrix with the elements of the input vector placed along the diagonal. Let $n!! \triangleq \prod_{k=0}^{\lceil \frac{n}{2} \rceil - 1}(n - 2k)$ be double factorial. For simplicity, we define $(-1)!! = 0!! = 1$. For two positive sequences $A_n$ and $B_n$, we write $A_n = \Theta(B_n)$ if there exist constants $c_1, c_2 > 0$ s.t. $c_1 B_n \le A_n \le c_2 B_n$ for sufficiently large $n$. Similarly, $A_n = \Theta_{\mathbb{P}}(B_n)$ indicates that the relationship holds with high probability as $n \to \infty$. We say $A_n = o(B_n)$ if, for every $\epsilon > 0$, there exists $N \in \mathbb{N}$ such that $A_n \le \epsilon B_n$ for all $n \ge N$. For a vector $x \in \mathbb{R}^n$, the expression $x[i]$ denotes the $i$-th element of $x$. For a matrix $A \in \mathbb{R}^{n \times m}$, $A[i]$ denotes the $i$-th column of $A$, and $A[i:j]$ denotes the columns from $i$ to $j$. Additionally, $A[:]$ refers to all elements of $A$.

**Hermite Polynomials** We employ the probabilist's Hermite polynomials (Moniri et al., 2024; Szegő, 1975; Grad, 1949; Bienstman, 2023). The $n$-th Hermite polynomials, $H_n(\cdot)$, are defined by the recurrence relation: $H_{n+1}(x) = xH_n(x) - nH_{n-1}(x)$, for $n \ge 1$, with the initial conditions $H_0(x) = 1, H_1(x) = x$. Using this recurrence, we have $H_2(x) = x^2 - 1, H_3(x) = x^3 - 3x, \cdots$.

---

[1]Unlike the definition in the original paper, which uses network's predictions, this paper examines Elasticity in the feature level.

## 2.1 PROBLEM SETTINGS

**Proportional Regime** We consider a two-class classification problem with classes $c_1$ and $c_2$, using two-layer neural networks in the proportional regime. Here, $\mathbf{n}$, $\mathbf{d}$, and $\mathbf{N}$ are sample size, data dimension, and feature dimension, respectively. We perform our analysis under the following regime: $n/d \to \psi_1, N/d \to \psi_2$ as $\mathbf{n}, \mathbf{d}, \mathbf{N} \to \infty$, where $\psi_1, \psi_2 \in (0, \infty)$. This setup reflects a scenario where the network width scales proportionally to the data size, aligning with common scaling practices in modern machine learning models.

**Training Data** Let $\mathscr{D} = (X, Y)$, where $X \in \mathbb{R}^{\mathbf{n} \times \mathbf{d}}, Y \in \{-1, 1\}^{\mathbf{n} \times 2} \subseteq \mathbb{R}^{\mathbf{n} \times 2}$, represent the training dataset. For any data point $(x, y)$, $y = (1, -1)^\top$ if $x \sim c_1$ and $y = (-1, 1)^\top$ if $x \sim c_2$, where $x \sim c_i$ indicates that $x$ belongs to class $c_i$. We denote the $i$-th column of $Y$, $Y[i]$, as $y_i \in \mathbb{R}^{\mathbf{n}}$. It follows that $y_1 = -y_2$. For every $i$-th row of $X[:][1 : \lfloor \mathbf{n}/2 \rfloor]$, we have $X[:][i] \sim c_1$, for every $i$-th row of $X[:][\lfloor \mathbf{n}/2 \rfloor + 1 : \mathbf{n}]$, we have $X[:][i] \sim c_2$. Let $\tilde{\mathscr{D}} = (\tilde{X}, \tilde{Y})$ an i.i.d. copy of $\mathscr{D}$.

**Evaluation Data** In this paper, we employ the **"Unseen"** dataset as the **Evaluation** dataset, which is drawn from a distribution different from the one used to generate the training dataset. We assume that "Unseen" samples follow a Gaussian distribution $x \sim \mathcal{N}(\mu, \Sigma)$, where $\mu \in \mathbb{R}^{\mathbf{d}}$ and $\Sigma \in \mathbb{R}^{\mathbf{d} \times \mathbf{d}}$.

**Network Structure** We consider two-layer networks to be fully connected. The initial weight of the first layer, $W_0 \in \mathbb{R}^{\mathbf{d} \times \mathbf{N}}$, is initialized as $W_0[i] \sim Unif(\mathbb{S}^{\mathbf{d}-1})$ for $i \in [\mathbf{d}]$. We denote $W$ as the one-step trained weight. The initial weights of the second layer, $a_c \in \mathbb{R}^{\mathbf{N}}$ for $c \in \{1, 2\}$, are initialized as $a_c \sim \mathbf{N}(0, \frac{1}{\mathbf{N}} I)$. For an input $x$, we define the initialized feature as $F_0(x) \triangleq \sigma(W_0^\top x)$ and the one-step trained feature as $F(x) \triangleq \sigma(W^\top x)$. The network output is defined as the following two-dimensional vector: $\left(\frac{1}{\sqrt{\mathbf{N}}} F(x)^\top a_1, \frac{1}{\sqrt{\mathbf{N}}} F(x)^\top a_2\right)^\top$. The network is designed to output $y = (1, -1)$ for $c_1$ and $y = (-1, 1)$ for $c_2$.

**Optimization Problem** Denote $\theta = \{W, a_1, a_2\}$ as the set of all network parameters. However, for feature analysis, we only train $W$ and use $a_1, a_2$ for calculating gradient. To classify the given data, we introduce the Mean Squared Error (MSE) loss

$$L(X, y; \theta) = \frac{1}{2n} \sum_{c \in \{1,2\}} ||y_c - \frac{1}{\sqrt{\mathbf{N}}} \sigma(XW) a_c||^2. \tag{1}$$

The weight update formula for the first layer is given by $W' = W + \eta \sqrt{\mathbf{N}} G$, where $\eta$ is the learning rate and $G$ is the negative gradient of $L(X, y; \theta)$ with respect to $W$ expressed as

$$G = -\frac{\partial L}{\partial W} = -\frac{1}{\mathbf{n}} \sum_{c = \{1,2\}} \left[ X^\top \left[ (\frac{1}{\sqrt{\mathbf{N}}} (\frac{1}{\sqrt{\mathbf{N}}} \sigma(XW) a_c - y_c) a_c^\top) \odot \sigma'(XW) \right] \right]. \tag{2}$$

Now, we introduce the assumptions for our theoretical analysis.

**Assumption 2.1** (Activation Function). *Let $\sigma(x)$ be an element-wise activation* s.t. $\sigma, \sigma', \sigma''$ is *bounded by $\lambda_\sigma$ almost surely ( a.s. ). For $z \sim \mathcal{N}(0, 1)$, it admits a Hermite decomposition* i.e. *$\sigma(z) = \sum_{k=0}^\infty c_k H_k(z)$, where $c_k = \frac{1}{k!} \mathbb{E}_z[\sigma(z) H_k(z)]$. Note that $\mathbb{E}[\sigma(z)] = c_0$ and $\mathbb{E}[z\sigma(z)] = c_1$. We denote $c_{\perp_{0,1}} \triangleq \sqrt{\mathbb{E}[\sigma^2(z) - c_1^2]}$. We assume $c_0 = 0, c_1 \neq 0$ and $c_k^2 k! \leq C k^{-3/2-w}$, for some constants $C, w > 0$.*

**Assumption 2.2** (Learning Rate). *$\eta = \Theta(\mathbf{n}^\alpha)$, $\frac{l-1}{2l} < \alpha < \frac{l}{2l+2}$. $l \in \mathbb{N}$.*

**Assumption 2.3** (Training Data Structure). *Let the class-conditional training data distributions $c_1$ and $c_2$ be Sub-Gaussian (Vershynin, 2018; Cole & Lu, 2024; Cao et al., 2021; Jambulapati et al., 2020; Sivakumar et al., 2015; Bombari et al., 2022; Bazinet et al., 2024).*

*Remark* 2.4 (MSE for Classification). Note that utilizing MSE in classification is as well-established as using softmax-cross entropy, especially in theoretical analyses of classification problems (Han et al., 2022; Zhou et al., 2022).

**Note 2.5** (Sub-Gaussian Training Data Distribution). *The data structure described in Assumption 2.3 allows us to transform the analysis of CK solving linear regression under Gaussian assumptions (Ba et al., 2022; 2023; Moniri et al., 2024) to classification problems. This extension can open new avenues for theoretical analyses of deep representations in classification tasks.*

# 3 ANALYSIS OF FEATURE IN THE PROPORTIONAL REGIME WITH MSE CLASSIFICATION SETTING AND SUB-GAUSSIAN DATA

In this section, we analyze the learning dynamics of a neural network in a single training step, assuming the training data $\mathcal{D}$ originates from two distinct Sub-Gaussian distributions with non-zero means. To achieve this, we decompose the gradient (equation 2) using Hermite decomposition, which allows us to extract the essential rank-one matrix structure. As a result, we approximate the one-step trained feature function $F(x) = \sigma((W_0 + \eta\sqrt{\mathbf{N}}G^\top x))$ as $F_l$ by deriving its Hermite expansion, which serves as a key step in deriving our main theorem. The entire process is carried out asymptotically in the proportional regime.

## 3.1 RANK-ONE APPROXIMATION OF THE FIRST GRADIENT

In this section, we follow the proof structure of Ba et al. (2022) to decompose gradient in our classification learning setting. Unlike their assumption of centered Gaussian training data, we consider non-centered Sub-Gaussian data distributions. In this process, we apply a novel approach involving the concentration of the operator norm on a random matrix. Also, since our framework is not in a teacher-student setting, we use class labels instead of a teacher function.

Starting from equation 2, by performing an orthogonal decomposition of the first Hermite expansion term and the remainder of $\sigma(x)$, we express $\sigma(x) = c_1 x + \sigma_\perp(x)$. The gradient $G$ is then decomposed as follows $G_0 = \mathbb{A} + \mathbb{B} + \mathbb{C}$ i.e.

$$
G_0 = \overbrace{\frac{c_1}{\mathbf{n}\sqrt{\mathbf{N}}}X^\top(y_1 a_1^\top + y_2 a_2^\top)}^{\mathbb{A}} + \overbrace{\frac{1}{\mathbf{n}\sqrt{\mathbf{N}}}X^\top(y_1 a_1^\top + y_2 a_2^\top) \odot \sigma'_\perp(XW_0)}^{\mathbb{B}}
$$
$$
- \frac{1}{\mathbf{n}\mathbf{N}}X^\top \sigma(XW_0)(a_1 a_1^\top + a_2 a_2^\top) \odot \sigma'(XW_0) \dots \mathbb{C}. \tag{3}
$$

We derive the norm bound for the terms $\mathbb{A}$, $\mathbb{B}$, and $\mathbb{C}$ in Lemma F.1. Using these bounds, we establish the following Proposition 3.1. For the proof, please refer to Appendix F.

**Proposition 3.1.** *Under the assumptions in subsection 2.1, and when $\mathbf{n}$ satisfy $1 - \kappa'\frac{\log^2 \mathbf{n}}{\sqrt{\mathbf{n}}} > 1/2$, the following holds:*

$$
||G_0 - \mathbb{A}|| \le \kappa\frac{\log^2 \mathbf{n}}{\sqrt{\mathbf{n}}}||G_0|| \quad w.p. 1 - C(\mathbf{n}e^{-c\log^2 \mathbf{n}} + e^{-c\mathbf{n}}). \tag{4}
$$

Now we utilize $\mathbb{A}$ as the approximate gradient for training the CK model given the training set $\mathcal{D}$.

## 3.2 ANALYSIS OF FEATURES AFTER ONE-STEP GD

We now study the feature space induced by the conjugate kernel after one step of gradient descent (GD). We first analyze $\sigma(\tilde{X}W) = F(\tilde{X}W_1) = F(\tilde{X}W_0 + \eta\tilde{X}G)$ by an approximation using Hermite polynomials. Denote $\beta \triangleq \frac{1}{\mathbf{n}\sqrt{\mathbf{N}}}X^\top y_1$ and let $\alpha = a_1 - a_2$. By Proposition 3.1 and results D, E in Lemma G.1, we generalize this to Lemma 3.2. For the proof, see Appendix G.

**Lemma 3.2** (Monomial Approximation of Data-Gradient). *For any $k \in \mathbb{N}$, sufficiently large $\mathbf{n}$, and w.p. 1 - o(1),*
$$
||(\tilde{X}G^\top)^{\circ k} - c_1^k(\tilde{X}\beta)^{\circ k}(\alpha^{\circ k})^\top|| \le C^k \mathbf{n}^{-\frac{k}{2}\log^{2k} \mathbf{n}}. \tag{5}
$$

Finally, we constructed the Data-Gradient form in our classification setup, satisfying the assumptions same to those in Theorem 3.2 of Moniri et al. (2024). We now decompose $F$ into a feasible form.

**Lemma 3.3** (Decomposition of Trained Features). *Let $F_0 = \sigma(\tilde{X}W_0^\top)$. With probability $1 - o(1)$,*
$$
F = F_l + \Delta,
$$
*where $F_l = F_0 + \sum_{k=1}^l c_1^k c_k \eta^k(\tilde{X}\beta)^{\circ k}(\alpha^{\circ k})^\top$ and $l$ is defined in section 2. Moreover, $||\Delta|| = o(\sqrt{\mathbf{n}})$, $||F_0|| = \Theta_{\mathbb{P}}(\sqrt{\mathbf{n}})$, and $||c_1^k c_k \eta^k(\tilde{X}\beta)^{\circ k}(\alpha^{\circ k})^\top||$ has an order larger than $o(\sqrt{\mathbf{n}})$.*

Based on these results, we analyze the feature representation using the approximation $F_l$, which dominates the residual term $||\Delta|| = o(\sqrt{\mathbf{n}})$ with probability $1 - o(1)$.

## 4 EMERGENCE OF ALIGNMENT AND LOCAL ELASTICITY

In this section we provide theorems indicating **train**($\beta$)-**unseen**($\mu$) similarity $|\beta^\top \mu|$ and $\beta^\top \Sigma \beta$ is governing the **Alignment** and the **Elasticity score**. Given separable Sub-Gaussian training data $\mathcal{D}$, we compute the approximate feature $F_l$ after a single gradient step in the above section.

**Condition 4.1** (Condition statement for Theorem 4.2 and Theorem 4.3). *Let $x, x' \sim \mathcal{N}(\mu, \Sigma)$ be i.i.d random vectors drawn from the arbitrary class conditional distribution given $\mu, \Sigma$. With assumption in section 2 and following from Lemma 3.3, remark the approximated initialized/trained neural network feature extractor as $F_0(x) = \sigma(W_0^\top x)$, $F_l(x) = F_0(x) + \sum_{k=1}^l c_1^k c_k \eta^k (x^\top \beta)^{\circ k} (\alpha^{\circ k})^\top$ where $\beta \triangleq \frac{1}{n\sqrt{N}} X^\top y_1$ with training datasaet $(X, y)$.*

**Theorem 4.2** (Alignment). *Following condition 4.1, denote $\mathcal{T}_k \triangleq c_1^k c_k \eta^k \mathbb{E}_x[(\beta^\top x)^k]$. Then, the average Inner Product between two approximated one-step trained features is as follows:*

$$\mathbb{E}_{x,x',\theta}[F_l(x)^\top F_l(x')] = \mathbb{E}_{x,\theta}\|F_0(x)\|^2 + 2\langle \mathbb{E}_{x,\theta} F_0(x), \sum_{k=1}^l \mathcal{T}_k \mathbb{E}_\theta[\alpha^{\circ k}]\rangle + \sum_{k=1,j=1}^l \mathcal{T}_k \mathcal{T}_j \mathbb{E}_\theta \langle \alpha^{\circ j}, \alpha^{\circ k}\rangle$$

(6)

*The first term $\mathbb{E}_{x,\theta}\|F_0(x)\|^2$ only depends on unseen distribution parameter $\mu, \Sigma$ without train distribution. The second term $2\langle \mathbb{E}_{x,\theta} F_0(x), \sum_{k=1}^l \mathcal{T}_k \mathbb{E}_\theta[\alpha^{\circ k}]\rangle$ depends on $\mu, \Sigma$, $|\beta^\top \mu|$ and $\beta^\top \Sigma \beta$. The last term $\sum_{k=1,j=1}^l \mathcal{T}_k \mathcal{T}_j \mathbb{E}_\theta \langle \alpha^{\circ j}, \alpha^{\circ k}\rangle$ depends on $|\beta^\top \mu|$ and $\beta^\top \Sigma \beta$. Therefore, the alignment measure grows as $|\beta^\top \mu|, \beta^\top \Sigma \beta$ increases.*

*Proof.* Proof is in Appendix I □

**Theorem 4.3** (Local Elasticity). *Following Condition 4.1, Then, the average $L^2$ distance between the initialized features $F_0(x)$ and the approximated one step trained features $F_l(x)$ is as follows:*

$$\mathbb{E}_{x,\theta}\|F_l(x) - F_0(x)\|^2 = \sum_{k=1}^l \sum_{m=1}^l \sum_{i=0}^{k+m} \kappa_{LE} |\beta^\top \mu|^{k+m-i} (\beta^\top \Sigma \beta)^{\frac{i}{2}} \mathbf{1}_{\mathrm{k+m}\ and\ i\ is\ even}.$$

(7)

*$\kappa_{LE}$ depends only on $k, m, i, \mathbf{N}, c_1, \eta$, and is independent of the data distribution parameters. The local elasticity measure grows as $|\beta^\top \mu|, \beta^\top \Sigma \beta$ increases.*

*Proof.* Proof is included in Appendix J □

**Note 4.4** (Interpretation of sign of $\beta$). *If the two given classes have a zero-centered symmetric structure, a symmetric representation should be learned regardless of the sign of $\beta$. This can be observed in our results and observations as well. We defined $\beta = \frac{1}{n\sqrt{N}} X^\top y_1$ in subsection 3.2. When the sign of $\alpha = a_1 - a_2$ is flipped, $\beta$ can also be defined as $\frac{1}{\mathbf{n}\sqrt{\mathbf{N}}} X^\top y_2$. With alternative definition, the same result is obtained for Theorem 4.2 and Theorem 4.3, where the scores are represented as polynomials of $|\beta^\top \mu|$ and non-negative $\beta^\top \Sigma \beta$.*

**Note 4.5** (Relationship between $l$ and learning rate $\eta$). *Our learning rate assumption is that it is determined by the parameter $l \in \mathbb{N}$, which determines the maximum Hermite expansion degree of the Alignment and LE scores as polynomials of $|\beta^\top \mu|$. This behavior aligns with the intuition that larger learning rates correspond to more aggressive updates of the features, causing them to shift and align more during the optimization process.*

## 5 EXPERIMENTS

*Remark 5.1.* Recall@1$= \mathbb{E}\mathbf{1}_{\mathrm{y_i=\hat{y}_{i,1\text{-}NN}}} \hat{y}_{i,1\text{-NN}}$ is the class of closest feature to $x_i$.

In our experiments, we examine the relationships between **train-unseen similarity** ( i.e. $|\beta^\top \mu|$, **Alignment**), **Elasticity**, and Recall@1. The experimental setups range from synthetic datasets trained with two-layer networks (*Setup 1, 2*) to real-world datasets, including CARS196 (Krause et al., 2013) and CUB200 (Wah et al., 2011), trained with multi-layer networks such as ResNet18 (*Setup 3*) and ResNet50 (He et al., 2015) (*Setup 4*).

**Setup 1, 2**  To evaluate the theory, we follow the configurations described in section 2. We use three different non-centered Sub-Gaussian distributions as training datasets: (i) a uniform distribution over a radius-$\sqrt{\mathbf{d}}$ ball (Data 1); (ii) a multi-dimensional element-wise truncated Gaussian distribution (Data 2); and (iii) a uniform distribution over a radius-$\sqrt{\mathbf{d}}$ sphere (Data 3).[2] We set $\mathbf{d} = \mathbf{n} = \mathbf{N} = 2^{11}$, and $\eta = \mathbf{n}^{0.25}$ in accordance with the assumptions. The means of Data 1 and 3 are $v$ and $-v$, respectively, where $v \triangleq 5r^2 \cdot \mathbf{u}$, with $\mathbf{u} \sim \text{Unif}\left(\mathbb{S}^{\mathbf{d}-1}\right)$. For Data 2, one class has support on $[1, \infty)$ across all dimensions, while the other class has support on $(-\infty, -1]$. We define $v \triangleq (1, 1, \cdots, 1)^\top$ for Data 2 used in Evaluation data generation.

For the evaluation data, we introduce unseen samples $x_{unseen}$, which are projected Gaussian distributed and defined as $x_{unseen} \triangleq z - {(z^\top \nu \nu)}/{\|\nu\|^4} + \nu$, where $z \sim \mathcal{N}(0, I)$, and $\nu \triangleq ev$ for **Setup 1** and $\nu \triangleq Rv$ for **Setup 2**, with $e \in (-1, 1)$, $R \in SO(d)$. We use this data for measuring Alignment, Elasticity, Recall@1. By adjusting $e$ and $R$, one can control the **train-unseen similarity** $\beta^\top \mu$, where $\mu \triangleq \mathbb{E}[x_{unseen}]$. Please refer to Figure 2 for illustrations of these setups.

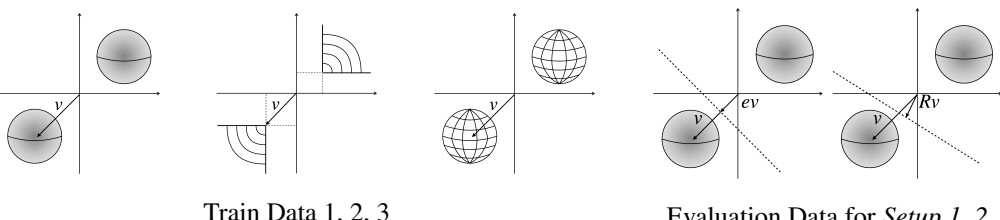

Train Data 1, 2, 3                          Evaluation Data for *Setup 1, 2*

Figure 2: Examples of training datasets (Data 1, 2, 3) and evaluation data used in *Setup 1, 2*.

**Setup 3, 4**  We also conduct the experiment with practical settings i.e. the multi-layer networks and the real-world data. In **Setup 3**, we designate either the CAR or CUB dataset and randomly select two classes as the training set. Then we sample five classes from each evaluation set of CAR and CUB as our new evaluation set. We set $\mathbf{d} = \mathbf{N} = 2^{11}, \mathbf{n} = 96$. The whole model consists of ResNet18 whose output dimension is $\mathbf{d}$, a single nonlinear layer $F(x) = \sigma(W^\top x)$, and classifier $a_1, a_2$. We measure $\beta$ and $\mu$ from the representations after ResNet18 architecture. Then they are passed through $F(x)$ and final classifier $a_1, a_2$. Note that we randomly initialize ResNet18 and do not freeze its layers during training. The **Setup 4**, conducted on the CARS196 and CUB200 datasets, and its every configuration follows the approach outlined in Zhai & Wu (2019), which represents a baseline in metric learning. We employ the *normsoft* metric learning loss function Zhai & Wu (2019). This setup is particularly relevant to our focus on unseen distribution, as it conducts the metric learning task with use of unseen data. The detailed configuration of two experiments is in Table 2.

**Alignment and Elasticity Observations**  With **Setup 1,2**, we analyze the behavior of Alignment, Elasticity as $|\beta^\top \mu|$ varies with $e$ and $R$. Following Thm. 4.2 and 4.3, as $|\beta^\top \mu|$ increases, we expect to observe a positive tendency in Alignment and Elasticity score defined in Definition 1.1, 1.2. In this experiment, the variable $e$ span from -0.9 to 0.9, and the 300 random rotation matrix is generated using a process in subsection K.4 for $R$. We repeat the experiment 30 times with different initializations of the neural network parameters and include the results along with the mean and standard deviation as in Figure 3, K.1, K.2. It demonstrates that Alignment and Elasticity score occur strongly as $e$ or $\beta^\top \mu$ increase. This phenomenon corresponds to the results of our theoretical findings, which suggest that the features from distributions closer to the training data emerge the stronger Alignment and Local Elasticity.

In **Setup 3**, we validate the theoretical results by adapting the network and data to a practical setting for a binary classification problem. After each training epoch, we evaluate $|\beta^\top \mu|$, Alignment, and Elasticity across evaluation datasets CAR and CUB. We calculate the ranks for each of these metrics—$|\beta^\top \mu|$, Alignment, and Elasticity—using the values measured across five classes in each dataset. These rankings are then compared across all metrics to see if they maintained consistent ranking orders using Kendall's $W$ (Kendall & Smith, 1939) ranking correlation. A $W$ value of 1

---

[2]The Sub-Gaussian property is proven for Data 1 and 3 in Vershynin (2018), and for Data 2 in Lemma E.1.

indicates complete agreement in rank order, while a value of 0 indicates no agreement. As a result, we found that, as the theory suggests, there is a rank correlation between Elasticity, Alignment, and $|\beta^\top \mu|$ on average across four different seeds. Numerically, during the middle stages of training, before the model converges, we observed that the model trained on CAR showed a rank correlation of at least 0.7 across all datasets, while the model trained on CUB exhibited a rank correlation of at least 0.5 across all datasets. See Figure 4. Additionally, on top of the strict order requirement of Kendall's $W$ statistic, we directly observe that $|\beta^\top \mu|$, Alignment, and Elasticity simultaneously increased or decreased without aggregation, as shown in Appendix N.

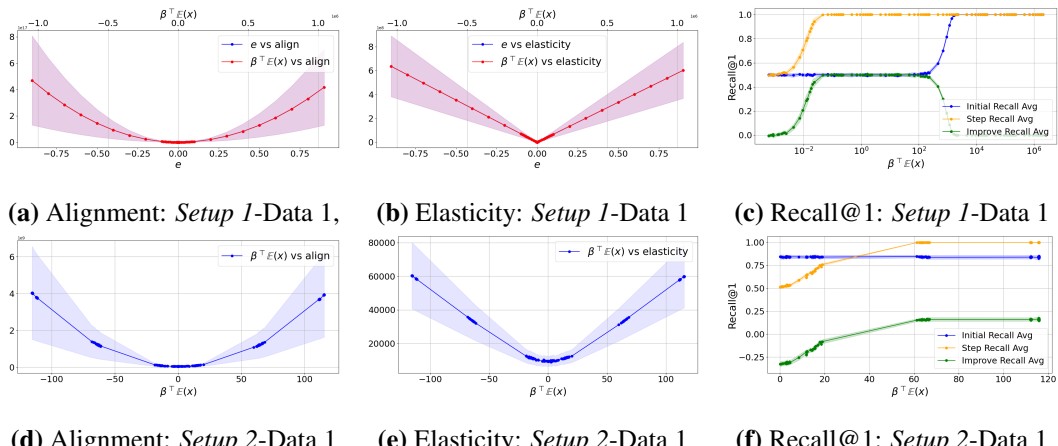

**(a)** Alignment: *Setup 1*-Data 1,    **(b)** Elasticity: *Setup 1*-Data 1    **(c)** Recall@1: *Setup 1*-Data 1

**(d)** Alignment: *Setup 2*-Data 1    **(e)** Elasticity: *Setup 2*-Data 1    **(f)** Recall@1: *Setup 2*-Data 1

Figure 3: Observation of Alignment **(a, d)** and Elasticity **(b, e)** Recall@1 **(c, f)**. Figure **(a, b)** are plotted across different $e$ (lower x-axis, exactly overlapped) and $\beta^\top \mu$ (upper x-axis) values. Figure **(c, d, e, f)** are plotted across different $\beta^\top \mu$ (x-axis) values. For figure **(c, f)** the blue line represents the clustering performance measured using the features in their initialized state, the orange line reflects the performance after one step of training, and the green line indicates the improvement, i.e., the difference between the two.

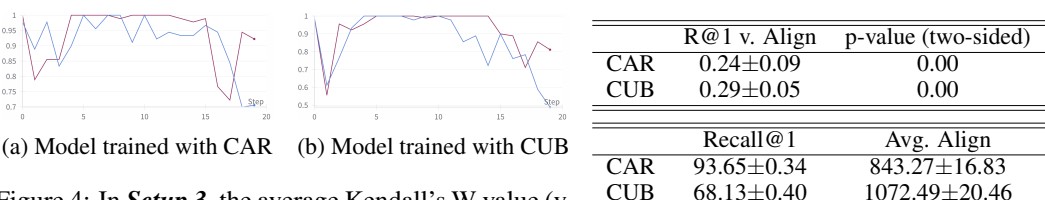

(a) Model trained with CAR    (b) Model trained with CUB

|  | R@1 v. Align | p-value (two-sided) |
|---|---|---|
| CAR | 0.24±0.09 | 0.00 |
| CUB | 0.29±0.05 | 0.00 |

|  | Recall@1 | Avg. Align |
|---|---|---|
| CAR | 93.65±0.34 | 843.27±16.83 |
| CUB | 68.13±0.40 | 1072.49±20.46 |

Figure 4: In *Setup 3*, the average Kendall's W value (y-axis) over step (x-axis) for a model trained on the (a) CAR and (b) CUB dataset with 4 different seeds. The magenta line represents the Kendall's W value for the CAR dataset, the blue line for the CUB dataset.

Table 1: In *Setup 4*, **(top)** The average correlation of Recall@1 and Alignment with p test. **(bottom)** The final R@1 and Align.

**Connections between $|\beta^\top \mu|$, Alignment and Recall@1**   In this section, we analyze the relationship between the train-unseen similarity, i.e. , $|\beta^\top \mu|$ and Recall@1 performance as well as Alignment score. Based on the theoretical finding that neural networks produce features with high alignment and elasticity for unseen classes close to the training data, we hypothesize that data from unseen data distribution similar to train classes undergoes greater shifts during learning, resulting in better alignment and cluster formation with superior Recall@1 performance in the feature space. To validate this hypothesis, we observe whether the higher train-unseen similarity leads to improved Recall@1 performance in *Setup 1, 2*.

Through *Setup 1, 2*, we measure Recall@1 with two classes using cosine similarity. See **(c, f)** in Figure 3 and K.4. One class is instantiated according to the original definition as *Setup*, while the other is constructed by inverting signs across all axes in data space. After a single learning step, we observe Recall@1 performance increases when the $|\beta^\top \mu|$ is higher across all neural networks (orange line in **(c, f)** at Figure 3).

Additionally, in *Setup 1*, we confirm that as $e(\beta^\top \mu)$ increases, the recall@1 measure at initialization also increases (blue line in **(c)** at Figure 3). This is a natural phenomenon because, by the definition of the dataset, as $e$ increases, the $L2$ distance between the mean of two evaluation classes increases. Further discussion of the observations of *Setup 1* is given in subsection K.5. In the setting of *Setup 2*, the value of $\beta^\top \mu$ changes due to variations in rotation; however, the distance between the two classes remains unchanged. Consequently, we observe that the initial Recall@1 does not vary (blue line in **(f)** at Figure 3). Moreover, we observed that when $\beta^\top \mu$ is too small, recall performance decreases after a single step of training (orange line in **(f)** at Figure 3). This suggests that unseen datasets, which are not too related to the domain of the train dataset, fail to generate meaningful representations.

In *Setup 4*, we extend our experiments from a two-class problem to practical multi-class scenarios within a baseline metric learning setting where the direct computation of $\beta$ is not feasible. To test the conjecture that strong train-unseen similarity leads to better alignment and improved Recall@1 performance, we analyze the correlation between Recall@1 and Alignment scores. At each step of training, we measure class-wise Alignment and class-wise Recall@1 with unseen classes. After, we compute the correlation between the Recall@1 and Alignment for each unseen class. Table 1 demonstrates the consistent tendencies of a positive correlation between Recall@1 and Alignment with a near zero p-value. This matches with our empirical results from a two-class synthetic dataset, where we observed a tendency for higher alignment to be associated with better Recall@1 performance. We use Pearson Correlation to measure the strength and direction of the linear relationship between Recall@1 and Alignment. For the p-value, we use two-sided test. We use Fisher's Combined Probability Test to combine the p-values. We provide unaggregated seed-wise results in Appendix O.

## 6 CONCLUSION

In this paper, we explored the emergence of Alignment and Local Elasticity in two-layer neural networks, focusing on their behavior when trained in the proportional regime. Our theoretical analysis extends the Conjugate Kernel (CK) framework to classification tasks, providing insights into how neural networks learn feature representations, particularly under Sub-Gaussian data distributions. We demonstrate that both Alignment and Local Elasticity arise simultaneously after just one step of training, especially in cases where data distributions closely resemble the training data. This phenomenon not only helps explain the clustering of representations but also sheds light on why neural networks trained on similar domains serve as effective feature extractors for tasks like metric learning. Furthermore, we validated our theoretical findings through experiments across various setups. These experiments confirmed the theoretical predictions, showing that neural networks exhibit stronger Alignment and Local Elasticity when evaluated on data distributions closer to the training set. Additionally, we identified a possible relationship between Recall@1, one of the generalization performance metrics for unseen distributions, and Alignment. Our work provides a unified framework for understanding feature learning in neural networks and opens avenues for further research in metric learning, transfer learning, and other task domains where neural networks are applied as feature extractors for unseen distributions. We believe this work offers valuable insights into the dynamics of neural networks, contributing to the broader understanding of deep learning theory.

**Reproducibility Statement**    In section 5, Appendix K, Appendix L, and Appendix O the dataset generation methods and hyperparameters for experimental reproduction are documented. The code used for data generation and the experimental from this research can be downloaded `https://anonymous.4open.science/r/emk-2E61`. Also, We derived all the proofs line by line.

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

## A    LIMITATIONS AND FUTURE WORKS

In this study, we have focused on non-centered sub-Gaussian training data, but this framework could be extended to more complex distributions, such as Gaussian mixtures. Exploring these broader classes of data distributions would enrich our understanding of the model's generalization capabilities. By the way, we have found that both Alignment and Local Elasticity are more strongly emerged by train-unseen similarity. However, it is necessary to explore how these two phenomena occur simultaneously. Furthermore, replacing the MSE loss with softmax cross-entropy could link this work more directly to Neural Collapse research (Ji et al., 2022), providing new insights into the geometric structures emerging during training. Additionally, studying scenarios where the parameters diverge further from their initialization after the first step of training could offer a long-term perspective on the learning dynamics. Moreover, There seems to be a connection between neural network alignment and the contraction of the Riemannian metric (Zavatone-Veth et al., 2023). Further research into this relationship could unveil deeper insights into the geometry of neural networks. Finally, in this study, the average of the Alignment and Elasticity scores was analyzed, and through multiple experiments, the validity of the analysis was supported. Theoretically, this can be extended to concentration as in Loureiro et al. (2021) and Mignacco et al. (2020), and analyzing the conditions under which the Alignment and Elasticity scores concentrate around the mean is one of the important research directions.

## B    EXTEND CLASSIFICATION SETTINGS TO REGRESSION SETTINGS

We chose a binary classification setup to analyze network learning, ensuring that it aligns with the settings proposed by He & Su (2019). However, our analysis is not limited to classification tasks alone. Inspired by works like Ba et al. (2022), we will incorporate a setting that reflects an regression form to demonstrate that our proof techniques can straightforwardly extend to scenarios involving regression setting. This straightforward adaptability is possible because our analysis applies to any loss or model that satisfies the condition of Proposition 3.1 and Lemma 3.2 in the main text. We argue that this is a key aspect showcasing the extensibility of our study.

Under all the assumptions stated in our paper, we define a new random variable $a \sim \mathcal{N}(0, \frac{1}{N}I)$, and modify the assumptions from Ba et al. (2022) for regression by replacing the centered Gaussian assumption with a non-centered sub-Gaussian assumption, leading to the following problem setup:

$$x_i \sim \text{SG s.t. } \mathbb{E}[x_i] \neq 0, \quad y_i = f^*(x_i) + \epsilon_i, \quad \epsilon_i \sim \mathcal{N}(0, \sigma_\epsilon^2),$$
$$f^* \text{ is a Lipschitz function, and } \sqrt{\mathbb{E}_x[f^{*2}]} = \Theta(1). \tag{8}$$

In the above setup, we define the loss as follows:

$$\theta = \{W, a\}, L(X, y; \theta) = \frac{1}{2n}||y - \sigma(XW)a||^2$$

$$G = -\frac{\partial L}{\partial W} = -\frac{1}{n}\left[X^\top\left[\left(\frac{1}{\sqrt{N}}(\frac{1}{\sqrt{N}}\sigma(XW)a - y)a^\top\right) \odot \sigma'(XW)\right]\right]$$

In this case, if we define $\alpha = a$ as opposed to the main text, $\alpha$ becomes a Gaussian with zero mean and variance halved, so it follows the same bound structure.

Specifically, based on the sub-Gaussian bound results in Appendix E, $\mathbb{A}$ can be bounded using the second equation of Lemma 14(i) from Ba et al. (2022) and the fact that $\mathbb{A}$ is rank-1, so $||A|| = ||A||_F$. For $\mathbb{B}$, we can remove $||a_2||_\infty$ from equation 20 in our Lemma F.1 proof. For $\mathbb{C}$, by removing $a_2 a_2^T$ from equation 30, we obtain the same bounds as the previous results.

In conclusion, these three bounds satisfy our Proposition 3.1 under the same conditions, and the same conclusion holds even outside of classification tasks. Note that $\beta$ is defined as in the previous setting.

## C    ADDITIONAL RELATED WORK

ADDITIONAL RELATED WORKS ABOUT NEURAL COLLAPSE    Additionally, investigations into the features of neural networks have led to observations suggesting that Neural Collapse does not

actually take place internally (Yang et al., 2023), and claims that it does not contribute to understanding generalization (Hui et al., 2022; Ma et al., 2023; Galanti et al., 2022).

REGARDING FEATURE-GRADIENT ALIGNMENT, Ziyin et al. (2024) argues that the alignment between features, weights, and gradients naturally facilitates the learning of compact representations. Beaglehole et al. (2024) investigate the Alignment of feature matrices by examining the correlation between feature matrices and the outer product of gradients. Furthermore, He & Su (2019) claim that gradients influence feature structures and Szolnoky et al. (2022); Chatterjee (2020) unveil the relation between the gradients of similar datasets.

RIEMANNIAN GEOMETRY PERSPECTIVE There is research from a Riemannian geometry perspective related to our result that the closer the data is to training data, i.e. , the larger $|\beta^\top \mu|$, the greater the occurrence of alignment and LE. Zavatone-Veth et al. (2023) find out the decrease of determinant of Riemannian metrics in the space i.e. volume decrease around training data. This is related to the strong tendency of the Local Elasticity and Alignment at the point close to the training samples.

NEURAL NETWORK THEORIES BASED ON THE TWO-LAYER ASSUMPTION Several prior studies have effectively utilized the feature extractor assumption same to our, to interpret phenomena observed in practical neural networks. For example, Damian et al. (2022) analyzed the efficient generalization and transfer performance of neural networks, while Tripuraneni et al. (2021) used this framework as a tool to study robustness to input distribution shifts. Similarly, Lee et al. (2023) employed it to analyze out-of-distribution inputs, and Bombari et al. (2023) utilized it to investigate adversarial robustness. These studies focused on understanding phenomena of neural network representations, particularly the hidden representations allowing them to model and explain behaviors observed in practical deep learning scenarios. Based on this body of work, we argue that the assumption of a two-layer network capable of learning hidden representations is a reasonable and effective framework for analyzing neural networks without significant loss of generality.

# D ADDITIONAL NOTATIONS

$\|\cdot\|_F$ is the Frobenius norm. $\|\cdot\|_\infty$ is the infinity norm. $\|\cdot\|_{\psi_2}$ is orlicz-2 norm $e^{(i)}$ Standard basis vector with 1 at position $i$.

ADDITIONAL INFORMATION OF HERMITE POLYNOMIALS Hermite polynomials can be represented as the following explicit form:

$$H_n(x) = (-1)^n e^{\frac{x^2}{2}} \frac{d^n}{dx^n} e^{-\frac{x^2}{2}}.$$

for $n \in \mathbb{N}_0$. Lastly, there are another expression:

$$H_n(x) = n! \sum_{m=0}^{\lfloor \frac{n}{2} \rfloor} \frac{(-1)^m}{m!(n-2m)!} \frac{x^{n-2m}}{2^m}$$

The probabilist's Hermite polynomials form an orthogonal set with respect to the standard normal weight function $w(x) = \frac{1}{\sqrt{2\pi}} e^{-\frac{x^2}{2}}$ on the interval $(-\infty, \infty)$. Their orthogonality condition is given by:

$$\int_{-\infty}^{\infty} H_m(x) H_n(x) \frac{1}{\sqrt{2\pi}} e^{-\frac{x^2}{2}} \, dx = \delta_{mn} n!,$$

where $\delta_{mn}$ is the Kronecker delta, and $n!$ is the factorial of $n$.

# E GENERALIZATION OF CENTERED SUB-GAUSSIAN RESULTS TOWARD NON-CENTERED

For more detailed explanation and well known results of Sub-Gaussian we used, please refer to Vershynin (2018; 2010).

**Lemma E.1.** *Truncated Gaussian distribution which have support on $(a, b)$ s.t. $a, b \in (-\infty, \infty)$ is Sub-Gaussian.*

*Proof.* Denote $\mathcal{N}_{(a,b)}(0, \sigma^2)$ is Truncated Gaussian distribution which have support on $(a, b)$ s.t. $a, b \in (-\infty, \infty)$. support $(\mathcal{N}_{(a,b)}(0, \sigma^2)) \subset \mathbb{R}^d$. Therefore, $\mathbb{P}(|X| \geq t)$ s.t. $X \sim \mathcal{N}_{(a,b)}(0, \sigma^2)$ have same tail behavior with Gaussian and Gaussian is Sub-Gaussian. $\square$

**Lemma E.2.** *Sum of non-centered Sub-Gaussian random variable is Sub-Gaussian.*

*Proof.* If the Orlicz 2 norm is bounded $||X||_{\psi_2} < \infty$, then X is Sub-Gaussian. Also, $||\mathbb{E}X||_{\psi_2} \leq C||X||_{\psi_2}$, and Sum of centered Sub-Gaussian random variable is Sub-Gaussian. We show $||\sum X_i||_{\psi_2} < \infty$, s.t. X is non-centered Sub-Gaussian.

$$
\begin{aligned}
||\sum X_i||_{\psi_2} &\leq ||\sum (X_i - \mathbb{E}X_i)||_{\psi_2} + ||\sum \mathbb{E}X_i||_{\psi_2} \\
&\leq ||\sum (X_i - \mathbb{E}X_i)||_{\psi_2} + \sum ||\mathbb{E}X_i||_{\psi_2} \\
&\leq ||\sum (X_i - \mathbb{E}X_i)||_{\psi_2} + C\sum ||X_i||_{\psi_2} < \infty
\end{aligned}
\tag{9}
$$

$\square$

**Lemma E.3.** *(Operator norm bound for non-centered Sub-Gaussian matrix, generalization of 4.4.5 in Vershynin (2018)) let $A \in \mathbb{R}^{m \times n}$, $A[i][j]$ is independent, non-centered Sub-Gaussian. $\forall t > 0$,*

$$
\begin{aligned}
||A|| &\leq CK(\sqrt{m} + \sqrt{n} + t) \text{ w.p. } 1 - \exp(-t^2) \\
\text{Alternatively, } ||A|| &\leq CK(\sqrt{m+n} + t) \text{ w.p. } 1 - \exp(-t^2)
\end{aligned}
\tag{10}
$$

$K = \max_{i,j} ||A[i][j]||_{\psi_2}$

**Lemma E.4.** *(Expectation of operator norm for non-centered Sub-Gaussian matrix generalization of 4.4.6 in Vershynin (2018))*

$$
\begin{aligned}
\mathbb{E}||A|| &\leq CK(\sqrt{m} + \sqrt{n}) \\
\text{Alternatively, } \mathbb{E}||A|| &\leq CK(\sqrt{m+n}), \quad \text{and, } \mathbb{E}||A||^2 \leq C(m+n)
\end{aligned}
\tag{11}
$$

*Proof of Lemma E.3 and Lemma E.4.* Based on the result of Lemma E.2, one can follow the same proof process of Vershynin (2018) $\square$

# F    ADDITIONAL RESULTS OF SECTION 3.1

For the aforementinoed $\mathbb{A}$, $\mathbb{B}$, and $\mathbb{C}$, we obtain bounds for each operator norm as follows

**Lemma F.1.**

$$
\begin{aligned}
&\mathbb{P}\left(||\mathbb{A}|| \leq C\left(\frac{1}{\sqrt{\mathbf{N}}} - C\frac{\sqrt{\mathbf{d}}}{\sqrt{\mathbf{nN}}}\right)\right) \leq 2\left(e^{-c\mathbf{N}} + e^{-c\mathbf{n}}\right) \\
&\mathbb{P}\left(||\mathbb{B}|| \geq \frac{C}{\mathbf{n}\sqrt{\mathbf{Nd}}}(\sqrt{\mathbf{n}} + \sqrt{\mathbf{d}})(\sqrt{\mathbf{n}} + \sqrt{\mathbf{N}})\log \mathbf{N}\right) \leq C\left(e^{-c\mathbf{N}} + e^{-c\mathbf{d}} + \mathbf{N}e^{-c\log^2 \mathbf{n}} + e^{-(\sqrt{\mathbf{n}}+\sqrt{\mathbf{d}})^2}\right) \\
&\mathbb{P}\left(||\mathbb{C}|| \geq \frac{C}{\sqrt{\mathbf{nN}}}(2\sqrt{\mathbf{d}} + \sqrt{\mathbf{n}})\log \mathbf{n} \log \mathbf{N}\right) \leq 2\left(\mathbf{n}e^{-c\mathbf{d}} + \mathbf{n}e^{-c\log^2 \mathbf{n}} + \mathbf{N}e^{-c\log^2 \mathbf{n}}\right).
\end{aligned}
\tag{12}
$$

*Proof of Lemma F.1 ($\mathbb{A}$).* Let us first define $\alpha = a_1 - a_2$. Then, we obtain

$$
\mathbb{A} = \frac{c_1}{\mathbf{n}\sqrt{\mathbf{N}}} X^\top \left(y_1 a_1^\top + y_2 a_2^\top\right).
\tag{13}
$$

Then, we can find an explicit notation of the norm as

$$
\begin{aligned}
||\mathbb{A}|| &= \frac{c_1}{\mathbf{n}\sqrt{\mathbf{N}}}||X^\top(y_1 a_1^\top + y_2 a_2^\top)|| = \frac{c_1}{\mathbf{n}\sqrt{\mathbf{N}}}||X^\top y_1(a_1^\top - a_2^\top)||_{op} \\
&= \frac{c_1}{\mathbf{n}\sqrt{\mathbf{N}}}||X^\top y_1||_2 ||(a_1 - a_2)||_2 = \frac{c_1}{\mathbf{n}\sqrt{\mathbf{N}}}\left(y_1^\top X X^\top y_1\right)^{1/2}||\alpha||_2
\end{aligned}
\tag{14}
$$

$\|\alpha\|_2$ **study** By definition, $\alpha \sim \mathbf{N}(0, \frac{2}{\mathbf{N}})$, so $\frac{\sqrt{\mathbf{N}}}{2}\alpha[i]$ is a sub-Gaussian. Use Thm 3.3.1 in Vershynin (2018),

$$\mathbb{P}\left(\left|\|\frac{\sqrt{\mathbf{N}}}{2}\alpha\| - \sqrt{\mathbf{N}}\right| \geq t\right) \leq 2e^{-ct^2} \quad \text{let } t = \sqrt{\mathbf{N}}/2 \tag{15}$$

$$\mathbb{P}(\|\alpha\|_2 \leq 1) \leq 2e^{-c\mathbf{N}}$$

$(y_1^\top X X^\top y_1)^{1/2}$ **study** Note that the $U, V$ matrices resulting from the SVD belong to the $O$-group, so there is no length transformation.

$$y_1^\top X X^\top y_1 = \|X^\top y_1\|_2^2 = \|U\Sigma V^\top y_1\|_2^2 = \|\Sigma V^\top y_1\|$$
$$= \sum_i \sigma_i^2 |V^\top y[i]|^2 \geq \sigma_{\min}^2 \sum_i |V^\top y[i]|^2 = \sigma_{\min}^2 \|y\|_2^2 = \mathbf{n}\sigma_{\min}^2 \tag{16}$$

We get $(y_1^\top X X^\top y_1)^{1/2} \geq \sqrt{\mathbf{n}}\sigma_{\min}$. $\sigma_{\min}$ is singular value of $X$ which is a anistropic sub-Gaussian matrix. With the result of Remark 1.2 in Liaw et al. (2016),

$$\mathbb{P}\sigma_{\min} \leq (\sqrt{\mathbf{n}} - c\sqrt{\mathbf{d}})) \leq e^{-\mathbf{n}}. \tag{17}$$

Therefore, $\mathbb{P}(\|\mathbb{A}\| \leq C(\frac{1}{\sqrt{\mathbf{N}}} - C\frac{\sqrt{d}}{\sqrt{\mathbf{n}\mathbf{N}}})) \leq 2(e^{-c\mathbf{N}} + e^{-c\mathbf{n}})$. $\qquad\square$

**Fact F.2** (from Ba et al. (2022)). *For $m \in \mathbb{R}^m, n \in \mathbb{R}^n, M \in \mathbb{R}^{m \times n}$,*

$$mn^\top \odot M = diag(m)M diag(n)$$
$$\|mn^\top \odot M\| \leq \|diag(m)\| \, \|M\| \, \|diag(n)\| = \|m\|_\infty \|M\|\|n\|_\infty n \tag{18}$$

**Lemma F.3.** *For Sub-Gaussian R.V. $a$,*

$$\mathbb{P}(\|a\|_\infty \leq t/\sqrt{\mathbf{N}}) \geq 1 - 2\mathbf{N}e^{-ct^2}$$

*Proof.* We use the Hoeffding inequality such that

$$\mathbb{P}(\|a\|_\infty \geq \frac{t}{\sqrt{\mathbf{N}}}) = \mathbb{P}\left(\max_i |a_i| \geq \frac{t}{\sqrt{\mathbf{N}}}\right) \leq \mathbb{P}\left(\bigcup_i\{|a_i| \geq \frac{t}{\sqrt{\mathbf{N}}}\}\right) \leq \sum_i \mathbb{P}\left(|a_i| \geq \frac{t}{\sqrt{\mathbf{N}}}\right)$$

$$\overset{\text{i.i.d.}}{=} \mathbf{N}\mathbb{P}\left(|a_i| \geq \frac{t}{\sqrt{\mathbf{N}}}\right) = \mathbb{P}(|\sqrt{\mathbf{N}}a_i| \geq t) \leq 2\mathbf{N}\exp(-ct^2)$$

$$\tag{19}$$
$$\square$$

**Fact F.4.** *Let a sub-Gaussian random variable $v$ s.t. $\|v\|_{\psi_2} \leq k$, and bounded function $\sigma$, then $\sigma(v)$ is Sub-Gaussian, i.e. $\|\sigma(v)\|_{\psi_2} \leq \|\lambda\|_{\psi_2} < \infty$.*

*Proof of Lemma F.1 ($\mathbb{B}$).*

$$\mathbb{B} = \frac{1}{\mathbf{n}\sqrt{\mathbf{N}}}X^\top(y_1 a_1^\top + y_2 a^\top) \odot \sigma_\perp'(XW_0) \tag{20}$$

$$\|\mathbb{B}\| \leq \frac{1}{\mathbf{n}\sqrt{\mathbf{N}}}\|X\| \, \|y_1 a_1^\top + y_2 a_2^\top \odot \sigma_\perp'(XW_0)\|$$

$$\leq \frac{1}{\mathbf{n}\sqrt{\mathbf{N}}}\|X\|\left(\|y_1 a_1^\top \odot \sigma_\perp'(XW_0)\| + \|y_2 a_2^\top \odot \sigma_\perp'(XW_0)\|\right)$$

$$\leq \frac{1}{\mathbf{n}\sqrt{\mathbf{N}}}\|X\|\left(\|y_1\|_\infty \, \|\sigma_\perp'(XW_0)\| \, \|a_1\|_\infty + \|y_2\|_\infty \, \|\sigma_\perp'(XW_0)\| \, \|a_2\|_\infty\right) \tag{21}$$

$$= \frac{1}{\mathbf{n}\sqrt{\mathbf{N}}}\|X\| \, \|\sigma_\perp'(XW_0)\|(\|a_1\|_\infty + \|a_2\|_\infty)$$

$\|\sigma'_\perp(XW_0)\|$ **study** Use the result of D.4 in Fan & Wang (2020), which is hold for orthogonal columns. $X$ is sampled from continuous support distribution $c_1, c_2$. The first vector is linearly independent with probability 1 due to the continuous support of its distribution. For the second vector, which is drawn independently, the probability that it lies in the span of the first vector is 0, as it also has a continuous density. This reasoning extends to n vectors, implying that, with high probability, they are orthogonal or nearly orthogonal because no vector falls into the span of the others. Thus, $\forall \mathbb{B} > 0$ following is hold.

$$\mathbb{P}(\{\|\sigma'_\perp\| \geq C(\sqrt{\mathbf{n}} + \sqrt{\mathbf{N}})\lambda_\sigma \mathbb{B}\}, \mathscr{A}_\mathbb{B}) \leq 2e^{-c\mathbf{N}}$$

$$\mathscr{A}_\mathbb{B} = \{\{\|W_0\| \leq \mathbb{B}\}, \{\sum_{i=1}^{\mathbf{N}}(\|W[i]\|^2 - 1)^2 \leq \mathbb{B}^2\}\}. \tag{22}$$

Therefore,

$$\mathbb{P}(\|\sigma'_\perp\| \geq C(\sqrt{\mathbf{n}} + \sqrt{\mathbf{N}})\lambda_\sigma \mathbb{B}) \leq 2e^{-c\mathbf{N}} + \mathbb{P}(\mathscr{A}_\mathbb{B}^c) \tag{23}$$

$\mathbb{P}(\mathscr{A}_\mathbb{B})$ **study** We choose $t = C\sqrt{\frac{\mathbf{d}}{\mathbf{N}}}$, $B = C\sqrt{\frac{\mathbf{d}}{\mathbf{N}}}$.

CASE OF $\|W_0\| \leq B$ By Lemma E.3,

$$\mathbb{P}(\|\sqrt{\mathbf{N}}W_0\| \geq 2\sqrt{\mathbf{N}} + \sqrt{\mathbf{d}}) \leq 2e^{-c\mathbf{N}} \Rightarrow \mathbb{P}(\|W_0\| \geq C\sqrt{\frac{\mathbf{d}}{\mathbf{N}}}) \leq 2e^{-c\mathbf{N}} \tag{24}$$

Therefore, $\|W_0\| \leq \mathbb{B}$ at least w.p. $1 - 2e^{-c\mathbf{N}}$

CASE OF $\sum_{i=1}^{N}(\|W[i]\|^2 - 1)^2 \leq \mathbb{B}^2$ By definition, $\|W_0[i]\|^2 = 1$, so $0 \leq \mathbb{B}^2$ w.p. 1.

We know $\mathbb{P}(\mathscr{A}_\mathbb{B}^c) \leq 2e^{-cN}$.

$$\mathbb{P}(\|\sigma'_\perp\| \geq C(\sqrt{\mathbf{n}} + \sqrt{\mathbf{N}})\sqrt{\frac{\mathbf{d}}{\mathbf{N}}}) \leq 2e^{-c\mathbf{N}} \tag{25}$$

Use Lemma F.3, and E.3,

$$\|\sigma'_\perp\| \leq C\left(\sqrt{\frac{\mathbf{nN}}{\mathbf{d}}} + \sqrt{\frac{\mathbf{N}^2}{\mathbf{d}}}\right) \qquad \text{w.p. } 1 - C(e^{-c\mathbf{N}} + e^{-c\mathbf{d}}) \tag{26}$$

$$\|a\|_\infty \leq \frac{t}{\sqrt{\mathbf{N}}} \qquad \text{w.p. } 1 - 2\mathbf{N}e^{-ct^2} \tag{27}$$

$$\|X\| \leq \sqrt{\mathbf{n}} + \sqrt{\mathbf{d}} + t' \qquad \text{w.p. } 1 - 2e^{-ct'^2}. \tag{28}$$

In summary, we get

$$\|\mathbb{B}\| \leq \frac{C}{\mathbf{n}\sqrt{\mathbf{N}}}(\sqrt{\mathbf{n}} + \sqrt{\mathbf{d}} + t')\left(\sqrt{\frac{\mathbf{nN}}{\mathbf{d}}} + \sqrt{\frac{\mathbf{N}^2}{\mathbf{d}}}\right)\frac{t}{\sqrt{\mathbf{N}}}$$

let $t = \log \mathbf{n}$, $t' = \sqrt{\mathbf{n}} + \sqrt{\mathbf{d}}$

$$\mathbb{P}(\|\mathbb{B}\| \geq \frac{C}{\mathbf{n}\sqrt{\mathbf{Nd}}}(\sqrt{\mathbf{n}} + \sqrt{\mathbf{d}})(\sqrt{\mathbf{n}} + \sqrt{\mathbf{N}})\log N) \leq C\left(e^{-c\mathbf{N}} + e^{-c\mathbf{d}} + \mathbf{N}e^{-c\log^2 \mathbf{n}} + e^{-(\sqrt{\mathbf{n}} + \sqrt{\mathbf{d}})^2}\right).$$

$$\tag{29}$$

This compelete the proof. $\square$

*Proof of Lemma F.1 ($\mathbb{C}$).* We know that $\sigma'$ is bounded, so $\|\sigma'\|_F \leq \lambda_\sigma \sqrt{\mathbf{nN}}$

$$\mathbb{C} = -\frac{1}{\mathbf{nN}}X^\top \sigma(XW_0)(a_1 a_1^\top + a_2 a_2^\top) \odot \sigma'(XW_0), \tag{30}$$

ans we can bound the norm as follows

$$\|\mathbb{C}\| \leq \frac{1}{\mathbf{n}\mathbf{N}} \|X\| (\|\sigma a_1 a_1^\top \odot \sigma'\| + \|\sigma a_2 a_2^\top \odot \sigma'\|)$$

$$\leq \frac{1}{\mathbf{n}\mathbf{N}} \|X\| (\|\sigma a_1\|_\infty \|a_1\|_\infty \|\sigma'\|_F + \|\sigma a_2\|_\infty \|a_2\|_\infty \|\sigma'\|_F) \tag{31}$$

$$\leq \frac{\lambda_\sigma}{\sqrt{\mathbf{n}\mathbf{N}}} \|X\| (\|\sigma a_1\|_\infty \|a_1\|_\infty + \|\sigma a_2\|_\infty \|a_2\|_\infty)$$

**Control of** $\|\sigma a\|_\infty$    Let $t = \sqrt{\mathbf{d}}$. Given $X$ s.t. $\mathbb{P}(|X[i] - \sqrt{\mathbf{d}}| \geq \sqrt{\mathbf{d}}) \leq 2e^{-ct^2}$, consider one element $\sigma(X[j]^\top W_0)a = \sum_i^{\mathbf{N}} a_i \sigma(x[j]^\top W_0[i])$.

We know $a[i], \sqrt{\mathbf{n}} W_0[i]$ is an independent centered sub-Gaussian, and use Fact F.4, then $\sigma\left(\frac{X[j]^\top}{\sqrt{\mathbf{N}}}\sqrt{\mathbf{N}}W_0\right)a$ is sub-exponential and mean is zero, since $\|a_i \sigma(x[j]^\top W_0[i])\|_{\psi_1} \leq \|a_i\|_{\psi_2}\|\sigma(x[j]^\top W_0[i])\|_{\psi_2} < \infty$. Apply the Bernstein inequality for the sub-exponential,

$$\mathbb{P}(|\sigma(X[j]^\top a)| \geq \log \mathbf{n} \text{ given } \{|X[i] - \sqrt{\mathbf{d}}| \geq \sqrt{\mathbf{d}}\}) \leq 2e^{-c\log^2 \mathbf{n}}. \tag{32}$$

For every element $\|\sigma(XW_0)a\|_\infty \leq \log \mathbf{n}$ w.p. $1 - [2\mathbf{n}e^{-c\log^2 \mathbf{n} + 2\mathbf{n}e^{-c\mathbf{d}}}]$

By Lemma F.3 $\mathbb{P}(\|a\|_\infty \leq t/\sqrt{\mathbf{N}}) \geq 1 - 2\mathbf{N}e^{-ct^2}$, and Lemma E.3 with $t = \sqrt{\mathbf{d}}$

$$\mathbb{P}\left(\|\mathbb{C}\| \geq \frac{C}{\sqrt{\mathbf{n}\mathbf{N}}}(2\sqrt{\mathbf{d}} + \sqrt{\mathbf{n}})\log \mathbf{n} \log \mathbf{N}\right) \leq 2(\mathbf{n}e^{-c\mathbf{d}} + ne^{-c\log^2 \mathbf{n}} + \mathbf{N}e^{-c\log^2 \mathbf{n}}). \tag{33}$$

$\square$

*Remark* F.5. In the proportional regime, as $\mathbf{n}, \mathbf{d}, \mathbf{N} \to \infty$, these quantities can be interchanged to a constant. Thus, Lemma F.1 is reformulated as follows

$$\mathbb{P}(\|\mathbb{A}\| \leq \kappa/\sqrt{\mathbf{n}}) \leq Ce^{-c\mathbf{n}})$$

$$\mathbb{P}\left(\|\mathbb{B}\| \geq \frac{C\log \mathbf{N}}{\mathbf{n}}\right) \leq C(e^{-c\mathbf{n}} + \mathbf{n}e^{-c\log^2 \mathbf{n}}) \tag{34}$$

$$\mathbb{P}\left(\|\mathbb{C}\| \geq \frac{C\log^2 \mathbf{N}}{\mathbf{n}}\right) \leq C(\mathbf{n}e^{-c\mathbf{n}} + \mathbf{n}e^{-c\log^2 \mathbf{n}})$$

*Proof of Proposition 3.1.* Using $\|G_0 - \mathbb{A}\| = \|\mathbb{B} + \mathbb{C}\| \leq \|\mathbb{B}\| + \|\mathbb{C}\|$ and Lemma F.5

$$\mathbb{P}\left(\|G_0 - \mathbb{A}\| \geq C\frac{\log^2 \mathbf{n}}{\mathbf{n}}\right) \leq \mathbb{P}\left(\|G_0 - \mathbb{A}\| \geq C\left(\frac{\log n}{n} + \frac{\log^2 \mathbf{n}}{\mathbf{n}}\right)\right) \leq Cne^{-c\log^2 \mathbf{n}}. \tag{35}$$

Therefore, almost surely, in the proportional limit,

$$\|G_0 - \mathbb{A}\| \leq C\frac{\log^2 \mathbf{n}}{\mathbf{n}} = \frac{\kappa}{\sqrt{\mathbf{n}}}\frac{C}{\kappa}\frac{\log^2 \mathbf{n}}{\sqrt{\mathbf{n}}} \leq \|\mathbb{A}\|\frac{C}{\kappa}\frac{\log^2 \mathbf{n}}{\sqrt{\mathbf{n}}} \leq \kappa'\frac{\log^2 \mathbf{n}}{\sqrt{\mathbf{n}}}(\|G_0\| + \|G_0 - \mathbb{A}\|). \tag{36}$$

$\square$

# G    ADDITIONAL RESULTS OF SECTION 3.2

**Lemma G.1.** *Given dataset $\mathcal{D}, \tilde{\mathcal{D}}$*

A. $M_a \triangleq \max_{1 \leq i \leq \mathbf{N}} |a_i| \leq \frac{C\log^{1/2}\mathbf{n}}{\sqrt{\mathbf{n}}}$ *w.p* $1 - 2ne^{-c\log \mathbf{n}}$

B. $M_b \triangleq \max_{1 \leq i \leq \mathbf{n}} |<\tilde{X}[i], \beta>| \leq C\log^{1/2}\mathbf{n}$, *w.p.* $1 - 2\mathbf{n}e^{-c\log \mathbf{n}}$

C. $M_{W_0} \triangleq \sup_{k \geq 1} \|(W_0 W_0^\top)^{\circ k}\| \leq C$

D. $\|\tilde{X}\| \leq C\sqrt{\mathbf{n}}$

E. $\sqrt{\mathbf{N}}||G|| = O_{\mathbb{P}}(1)$

F. $||A^{\circ k}|| \leq ||A||^k$

*Proof.* For A, B, C, and D, we employ proof techniques adapted from Moniri et al. (2024).

For A, B, by hoeffding inequality $\mathscr{P}(|X_i| \geq t) \leq 2e^{-ct^2}$ for $t = \log^{1/2} \mathbf{n}$, and use $a_i, \langle \tilde{X}[i], \beta \rangle$ is Sub-Gaussian.

For C, refer Moniri et al. (2024).

For D, by Lemma E.3 and the proportional regime.

For E, by Lemma F.1 $||G|| \leq ||\mathbb{A}|| + ||\mathbb{B}|| + ||\mathbb{C}|| = O_{\mathbb{P}}(\frac{1}{\sqrt{\mathbf{n}}} + \frac{\log \mathbf{n}}{\mathbf{n}} + \frac{\log^2 \mathbf{n}}{\mathbf{n}}) = O_{\mathbb{P}}(\frac{1}{\sqrt{\mathbf{n}}})$

For F, refer Bai & Silverstein (2010) Corollary A.21. □

**Corollary G.2.** *By Proposition 3.1 and D, E in Lemma G.1, we have w.p. $1 - o(1)$.*

$$||\tilde{X}G^\top - \mu_1 \tilde{X}\beta\alpha^\top|| = O(\frac{\log^2 \mathbf{n}}{\sqrt{\mathbf{n}}}) \tag{37}$$

*Remark* G.3. Remark $W_1 = W_0 + \eta\sqrt{\mathbf{n}}G$, so $\tilde{X}W_1 = \tilde{X}W_0 + \eta\sqrt{\mathbf{n}}\tilde{X}G$.

*Proof of Lemma 3.2.* $k = 1$ is trivial with above statements. We follow Moniri et al. (2024) for $k \geq 2$. We need to show $\exists C > 0$, w.p. 1-o(1)

$$||(\tilde{X}G^\top)^{\circ k} - c_1^k(\tilde{X}\beta)^{\circ k}(\alpha^{\circ k})^\top|| \leq C^k \mathbf{n}^{-\frac{k}{2}} \log^{2k} \mathbf{n} \tag{38}$$

$$(\tilde{X}G^\top)^{\circ k} = (\tilde{X}G^\top - c_1\tilde{X}\beta\alpha^\top + c_1\tilde{X}\beta\alpha^\top)^{\circ k}$$

$$= \sum_{j=1}^{k} \binom{k}{j} c_1^{k-j} \text{diag}(\tilde{X}\beta)^{\circ(k-j)}(\tilde{X}G^\top - c_1\tilde{X}\beta\alpha)^{\circ j} \text{diag}(\alpha)^{\circ(k-j)} \quad \text{by binomial theorem}$$

$$+ c_1^k \text{diag}(\tilde{X}\beta)^{\circ k} \text{diag}(\alpha)^{\circ k} \tag{39}$$

Thus, $(\tilde{X}G^\top)^{\circ k} - c_1^k(\tilde{X}\beta)^{\circ k}(\alpha^{\circ k})^\top = \sum_{j=1}^{k} \binom{k}{j} c_1^{k-j} \text{diag}(\tilde{X}\beta)^{\circ(k-j)}(\tilde{X}G^\top - c_1\tilde{X}\beta\alpha)^{\circ j} \text{diag}(\alpha)^{\circ(k-j)}$

$$\tag{40}$$

We have to norm bound RHS of equation 40

$$||\text{diag}(\tilde{X}\beta)^{\circ(k-j)}(\tilde{X}G^\top - c_1\tilde{X}\beta\alpha)^{\circ j}\text{diag}(\alpha)^{\circ(k-j)}||$$

$$\leq ||\text{diag}(\tilde{X}\beta)^{\circ(k-j)}|| \, ||(\tilde{X}G^\top - c_1\tilde{X}\beta\alpha)^{\circ j}|| \, ||\text{diag}(\alpha)^{\circ(k-j)}|| \tag{41}$$

$$\leq (M_a M_b)^{k-j}||(\tilde{X}G^\top - c_1\tilde{X}\beta\alpha)||^j \qquad \text{by Lemma G.1 A, B, F}$$

In summary, w.p. $1 - o(1)$

$$||(\tilde{X}G^\top)^{\circ k} - c_1^k(\tilde{X}\beta)^{\circ k}(\alpha^{\circ k})^\top|| \leq C \sum_{j=1}^{k} \left(\frac{\log^{1/2} \mathbf{n}}{\sqrt{\mathbf{n}}}\right)^{k-j} \left(\frac{\log^2 \mathbf{n}}{\sqrt{\mathbf{n}}}\right)^j \tag{42}$$

□

*Remark* G.4. The definition of gradient $G$ and the size of the norm are different between Moniri et al. (2024) and our paper, but both produce the same results as above, up to scaling factor $\frac{1}{\sqrt{\mathbf{N}}}$.

# H STUDY OF EXPECTATION OF HERMITE POLYNOMIAL

The inner product between a random Gaussian vector $x$ and a vector $u, v$, where $u, v$ corresponds to a column of the weight matrix $W$ or $\beta$, is substituted into the variable of a Hermite polynomial and its expectation is derived.

We have analyzed various macroscopic results regarding the feature space of a neural network using Hermite polynomials and different activation functions. We have cited previously known facts, while our derived results are presented without explicitly marking them as new. We believe these findings will strengthen our paper and aid in the analysis of dynamics across different feature spaces.

## H.1 EXPECTATION OF A PRODUCT OF TWO HERMITE POLYNOMIALS

Here is the result of the expectation of the product of two Hermite polynomials, utilizing the orthogonality of Hermite polynomials.

**Lemma H.1** (Orthogonality of Hermite polynomials from Lemma C.1 Moniri et al. (2024)). *See also derivation in Chapter 11.2 O'Donnell (2021).*

*Let $(Z_1, Z_2)$ be jointly Gaussian with $\mathbb{E}[Z_1] = \mathbb{E}[Z_2] = 0$, $\mathbb{E}[Z_1^2] = \mathbb{E}[Z_2^2] = 1$, and $\mathbb{E}[Z_1 Z_2] = \rho$. Then for any $k_1, k_2 \in \{0, 1, \cdots, \}$*

$$\mathbb{E}[H_{k_1}(Z_1) H_{k_2}(Z_2)] = k_1! \rho^{k_1} \mathbf{1}_{k_1 = k_2}$$

*In the other form, for $d \in \mathbb{N}$, $Z \sim \mathcal{N}(0, I_d)$, $a, b \in \mathbb{S}^{d-1}$,*

$$\mathbb{E}[H_{k_1}(Z^\top a) H_{k_2}(Z^\top b)] = k_1! (a^\top b)^{k_1} \mathbf{1}_{k_1 = k_2}$$

**Fact H.2.** *Let $W \in \mathbb{R}^{d \times N}$ s.t. $\forall i\ W[i] \in \mathbb{S}^{d-1}$. For $Z \sim \mathcal{N}(0, I)$,*

$$\mathbb{E}_{Z \sim n(0,1)}[H_j(W^\top Z) H_k(W^\top Z)^\top] = k!(W^\top W)^{\circ j} \mathbf{1}_{j=k} \tag{43}$$

$$\mathbb{E}_{Z \sim n(0,1)}[H_j(W^\top Z)^\top H_k(W^\top Z)] = k! \sum ||W[i]||^{2j} \mathbf{1}_{j=k} = k! N \mathbf{1}_{j=k} \tag{44}$$

*Proof.* We apply $H_j$ element-wise. By Lemma H.1, we can acquire the above result. $\qquad\square$

The following remark presents a modified condition of Lemma H.1 for the case where $a, b \notin \mathbb{S}^{d-1}$ in Lemma H.1. In this case, the variances of $Z^\top a$ and $Z^\top b$ are not equal to 1, and the covariance may exceed the bounds $[-1, 1]$. Under this condition, we will compute the expectation of the product of two Hermite polynomials as in Lemma H.1.

*Remark* H.3 (the modified condition of Lemma H.1). For $d \in \mathbb{N}$, $u, v \in \mathbb{R}^d$, $Z \sim \mathcal{N}(0, I_d)$,

$Z_1 = \langle u, Z \rangle \sim \mathcal{N}(0, ||u||_2^2)$, $Z_2 = \langle v, Z \rangle \sim \mathcal{N}(0, ||v||_2^2)$.

Then, $Z_1, Z_2$ is $\rho =^\triangleq \langle \frac{u}{||u||}, \frac{v}{||v||} \rangle$ - correlated

$$
\begin{aligned}
corr(Z_1, Z_2) &= \frac{\mathbb{E}[Z_1 Z_2]}{\sqrt{V(Z_1)}\sqrt{V(Z_2)}} &&= \frac{\mathbb{E}_Z \langle u, Z \rangle \langle v, Z \rangle}{||u||\,||v||} \\
&= \frac{\mathbb{E}_g \sum_i \sum_j u_i v_j Z_i Z_j}{||u||\,||v||} &&= \frac{\sum_i \sum_j u_i v_j \mathbb{E}_Z[Z_i Z_j]}{||u||\,||v||} \\
&= \frac{\langle u, v \rangle}{||u||\,||v||}
\end{aligned} \tag{45}
$$

Additionally,

$$\begin{pmatrix} Z_1 \\ Z_2 \end{pmatrix} \sim n\left( \begin{pmatrix} 0 \\ 0 \end{pmatrix}, \begin{pmatrix} ||u||^2 & \langle u, v \rangle \\ \langle v, u \rangle & ||v||^2 \end{pmatrix} \right) \tag{46}$$

We first introduce Isserlis' theorem, which is essential for the proof. This theorem allows the expectation of the product of centered Gaussian random variables to be expressed as a product of covariances, making the computation feasible.

**Theorem H.4** (Isserlis' Theorem (Isserlis, 1918; Vignat, 2011)). *Let $X = (X_1, \cdots, X_d)$ Gaussian random vector s.t. $\mathbb{E}[X] = 0$ , and let $A = \{\alpha_1, \cdots, \alpha_N\}$ be set of integers s.t. $1 \le \alpha_i \le d$, $\forall i$. Denote $X_A = \prod_{\alpha_i \in A} X_{\alpha_i}$, and $X_\emptyset = 1$. Let $\prod(A)$ denote partitions of $A$ into disjoint pairs and $\sigma \in \prod(A)$ is pair.*

$$\mathbb{E}[X_A] = \sum_{\sigma \in \prod(A)} \prod_{(i,j) \in \sigma} \mathbb{E}[X_{\alpha_i} X_{\alpha_j}] \mathbf{1}_{\text{d is even}}. \tag{47}$$

Now, we generalize the assumptions from the previous works so that Lemma H.1 holds for arbitrary vectors as Remark H.3. This could allow the weights of the networks to become analyzable when they go beyond the assumption of lying on the unit spheres.

**Theorem H.5** (Generalization of Lemma H.1 for centered Gaussian distribution). *For $d \in \mathbb{N}$, $u, v \in \mathbb{R}^d$, $g \sim \mathcal{N}(0, I_d)$, $\langle u, g \rangle \sim \mathcal{N}(0, ||u||_2^2)$, $\langle v, g \rangle \sim \mathcal{N}(0, ||v||_2^2)$.*

$$
\begin{aligned}
&\mathbb{E}_g[H_j(u^\top g)H_k(v^\top g)] \\
&= \frac{j!\langle u, v \rangle^j}{||u||^2 ||v||^2} \mathbf{1}_{j=k} - \frac{(||u||^2 - 1)(||v||^2 - 1)}{||u||^2 ||v||^2} \mathbb{E}_g[(v^\top g)^k (u^\top g)^j] \\
&\quad + \frac{(||v||^2 - 1)}{||v||^2} \mathbb{E}_g[H_j(u^\top g)(v^\top g)^k] + \frac{(||u||^2 - 1)}{||u||^2} \mathbb{E}_g[H_k(v^\top g)(u^\top g)^j]
\end{aligned}
\tag{48}
$$

*Remark* H.6. The same results can be derived as in Lemma H.1 when the variance is 1 in Thm. H.5.

*Proof of Theorem H.5.* (Generalize Chapter 11.2 O'Donnell (2021)'s derivation to non unit variance)

$\mathbb{E}_{z \sim n(0,\sigma^2)}[e^{tz}]$ **study**

First, we study about $\mathbb{E}_{g \sim n(0,\sigma^2)}[e^{tg}]$ in order to analysis non unit variance case.

$$
\begin{aligned}
\mathbb{E}_{g \sim n(0,\sigma^2)}[e^{tg}] &= \frac{1}{\sqrt{2\pi}\sigma} \int e^{tg} e^{-\frac{g^2}{2\sigma^2}} dg \\
&= \frac{1}{\sqrt{2\pi}\sigma} e^{\frac{1}{2}t^2} \int \exp\left(-\frac{(g - \sigma^2 t)^2}{2\sigma^2}\right) \quad \text{complete square} \\
&= e^{\frac{1}{2}t^2}
\end{aligned}
\tag{49}
$$

$\mathbb{E}_{Z,Z'}[\exp(sZ + tZ')]$ **study**

Studying $\mathbb{E}_{Z,Z'}[\exp(sZ + tZ')]$, we can derive what we need to show.

$$
\begin{aligned}
\mathbb{E}_{Z,Z'}[\exp(sZ + tZ')] &= \mathbb{E}_{g \sim n(0,I)}[\exp(s\langle u, g \rangle) + \exp(t\langle v, g \rangle)] \\
&= \prod_i \mathbb{E}_{g \sim n(0,1)}[\exp((su_i + tv_i)g_i)] \quad\quad\quad \text{Use equation 49} \\
&= \prod_i \exp\left(\frac{1}{2}(su_i + tv_i)^2\right) = \prod_i \exp\left(\frac{1}{2}s^2||u||^2 + \langle u, v \rangle st + \frac{1}{2}t^2||v||^2\right)
\end{aligned}
\tag{50}
$$

Therefore,

$$\exp(\langle u, v \rangle st) = \mathbb{E}_g[\exp(su^\top g - \frac{1}{2}s^2||u||^2)\exp(tv^\top g - \frac{1}{2}t^2||v||^2)].$$

**Facts for proof** : one can verify below propositions with simple calculations.

Let $P_j(z) + z^j = H_j(z)$, $C_u = ||u||^2 - 1$, $a > 0$

Let $f(s) = \exp(sz - \frac{1}{2}s^2)$, $\bar{f}(s) = \exp(sz - \frac{1}{2}as^2)$

7.3.A. By Taylor expansion, $\exp(\langle u, v \rangle st) = \sum_{j=0}^{\infty} \frac{1}{j!}\langle u, v \rangle^j s^j t^j$.

7.3.B. By Taylor expansion, $\bar{f}(s) = \sum_{j=0}^{\infty} \frac{1}{j!} \bar{f}^{(n)}(0)s^j$

7.3.C. $\bar{f}^{(n)}(0) = H_n(z) + C_u P_n(z)$

By using the fact that $\exp(\langle u, v\rangle st) = \mathbb{E}_g[\exp(su^\top g - \frac{1}{2}s^2\|u\|^2)\exp(tv^\top g - \frac{1}{2}t^2\|v\|^2)]$, we can eliminate the different orders of $s\,t$ by a Taylor expansion and equating all monomials of the resulting polynomials.

$$
\begin{aligned}
j!\langle u, v\rangle^j \mathbf{1}_{\mathrm{j=k}} &= \mathbb{E}_g\Big[(H_j(u^\top g) + P_j(u^\top g)C_u)(H_j(v^\top g) + P_j(v^\top g)C_v)\Big] \\
&= \mathbb{E}_g\Big[(H_j(u^\top g) + (H_j(u^\top g) - (u^\top g)^j)C_u)(H_j(v^\top g) + (H_j(v^\top g) - (v^\top g)^j)C_v)\Big] \\
&= \|u\|^2\|v\|^2\mathbb{E}_g[H_j(u^\top g)H_j(v^\top g)] + (\|u\|^2 - 1)(\|v\|^2 - 1)\mathbb{E}_g[(v^\top g)^j(u^\top g)^j] \\
&\quad - \|u\|^2(\|v\|^2 - 1)\mathbb{E}_g[H_j(u^\top g)(v^\top g)^j] - \|v\|^2(\|u\|^2 - 1)\mathbb{E}_g[H_j(v^\top g)(u^\top g)^j]
\end{aligned}
\tag{51}
$$

Therefore,

$$
\begin{aligned}
&\mathbb{E}_g[H_j(u^\top g)H_j(v^\top g)] \\
&= \frac{j!\langle u, v\rangle^j}{\|u\|^2\|v\|^2}\mathbf{1}_{\mathrm{j=k}} - \frac{(\|u\|^2 - 1)(\|v\|^2 - 1)}{\|u\|^2\|v\|^2}\mathbb{E}_g[(v^\top g)^j(u^\top g)^j] \\
&\quad + \frac{(\|v\|^2 - 1)}{\|v\|^2}\mathbb{E}_g[H_j(u^\top g)(v^\top g)^j] + \frac{(\|u\|^2 - 1)}{\|u\|^2}\mathbb{E}_g[H_j(v^\top g)(u^\top g)^j]
\end{aligned}
\tag{52}
$$

Note that the result of Lemma H.7 can be applied for concrete calculation, and conclude the proof.

$\square$

**Lemma H.7.** *For $d \in \mathbb{N}$, $u, v \in \mathbb{R}^d$, $g \sim \mathcal{N}(0, I_d)$, $\bar{Z}_1 = \langle u, g\rangle$, $\bar{Z}_2 = \langle v, g\rangle$.*

$$
\begin{pmatrix} \bar{Z}_1 \\ \bar{Z}_2 \end{pmatrix} \sim n\left(\begin{pmatrix} 0 \\ 0 \end{pmatrix}, \begin{pmatrix} \|u\|^2 & \langle u, v\rangle \\ \langle v, u\rangle & \|v\|^2 \end{pmatrix}\right)
\tag{53}
$$

*$X_{\alpha_i}$ is defined at Thm. H.4*

$$
\begin{aligned}
\mathbb{E}_{\bar{Z}_1, \bar{Z}_2}[H_j(\bar{Z}_1)\bar{Z}_2^k] &= j!\sum_{m=0}^{\lfloor\frac{j}{2}\rfloor} \frac{(-1)^m}{m!(j-2m)!2^m} \sum_{\sigma\in\prod(\{\{\bar{Z}_1\}\times j-2m\}\cup\{\{\bar{Z}_2\}\times k\}\})} \prod_{(p,q)\in\sigma} \mathbb{E}[X_{\alpha_p}X_{\alpha_q}]\mathbf{1}_{\mathrm{j+k-2m\ is\ even}} \\
\mathbb{E}_{\bar{Z}_1, \bar{Z}_2}[\bar{Z}_1^j\bar{Z}_2^k] &= \sum_{\sigma\in\prod(\{\{\bar{Z}_1\}\times j\}\cup\{\{\bar{Z}_2\}\times k\}\})} \prod_{(p,q)\in\sigma} \mathbb{E}[X_{\alpha_p}X_{\alpha_q}]\mathbf{1}_{\mathrm{j+k\ is\ even}}
\end{aligned}
\tag{54}
$$

*Proof.* By explicit formula of Hermite polynomials

$$
\mathbb{E}_{\bar{Z}_1, \bar{Z}_2}[H_j(\bar{Z}_1)(\bar{Z}_2)^k] = j!\sum_{m=0}^{\lfloor\frac{j}{2}\rfloor} \frac{(-1)^m}{m!(j-2m)!2^m}\mathbb{E}_{\bar{Z}_1, \bar{Z}_2}[\bar{Z}_1^{j-2m}\bar{Z}_2^k]
\tag{55}
$$

Therefore, we need to figure out $\mathbb{E}_{\bar{Z}_1, \bar{Z}_2}[\bar{Z}_1^p\bar{Z}_2^q]$. We know $\bar{Z}_1, \bar{Z}_2$ is mean zero Gaussian, so we can apply Thm. H.4 with $A = \{\{\bar{Z}_1\}\times p\}\cup\{\{\bar{Z}_2\}\times q\}\}$, $\mathbb{E}[\bar{Z}_1^p\bar{Z}_2^q] = \sum_{\sigma\in\prod(A)}\prod_{(\tau,v)\in\sigma}\mathbb{E}[X_{\alpha_\tau}X_{\alpha_v}].\mathbf{1}_{\mathrm{p+q\ is\ even}}$

$\square$

**Corollary H.8** (Corollary of Lemma H.7). *Remark $Z_1 \sim \mathcal{N}(0, \|u\|^2)$ For the case $k = 0$,*

$$
\mathbb{E}_{\bar{Z}_1}[\bar{Z}_1^j] = \|u\|^j(j-1)!!\mathbf{1}_{\mathrm{j\ is\ even}}
\tag{56}
$$

*Proof.*

$$\mathbb{E}_{\bar{Z}_1, \bar{Z}_2}[\bar{Z}_1^j \bar{Z}_2^k] = \mathbb{E}_{\bar{Z}_1}[\bar{Z}_1^j] = \sum_{\sigma \in \prod(\{\bar{Z}_1\} \times j)} \prod_{(p,q) \in \sigma} \mathbb{E}[X_{\alpha_p} X_{\alpha_q}] \mathbf{1}_{\text{j is even}}$$

$$= \sum_{\sigma \in \prod(\{\bar{Z}_1\} \times j)} \prod_{(p,q) \in \sigma} \|u\|^2 \mathbf{1}_{\text{j is even}} = \sum_{\sigma \in \prod(\{\bar{Z}_1\} \times j)} \|u\|^j \mathbf{1}_{\text{j is even}} = (j-1)!! \|u\|^j \mathbf{1}_{\text{j is even}}$$

(57)

$\square$

### H.2 Expectation of a product of two Hermite polynomials—Generalization toward non-centered Gaussian

We will change Theorem H.5 and Lemma H.7 to adopt a generalized Gaussian assumption with a mean of zero.

**Lemma H.9** (Taylor expansion of Hermite polynomials from Lemma C.2 Moniri et al. (2024)). *For any $k_1, k_2 \in \{0, 1, \cdots, \}$ and $x, y \in \mathbb{R}$,*

$$H_k(x + y) = \sum_{j=0}^{k} \binom{k}{j} x^j H_{k-j}(y). \tag{58}$$

**Theorem H.10** (Generalization of Thm. H.5 for any Gaussian distribution). *For $d \in \mathbb{N}$, $u, v \in \mathbb{R}^d$, $\xi \sim \mathcal{N}(0,1)$, $g \sim \mathcal{N}(\mu, \Sigma)$, $Z_1 = \langle u, g \rangle \sim \mathcal{N}(\mu^\top u, u^\top \Sigma u)$, $Z_2 = \langle v, g \rangle \sim \mathcal{N}(\mu^\top v, v^\top \Sigma v)$.*

$$\mathbb{E}_g[H_j(Z_1) H_k(Z_2)]$$

$$= \sum_{\alpha=0}^{j} \sum_{\beta=0}^{k} \binom{j}{\alpha} \binom{k}{\beta} (u^\top \mu)^\alpha (v^\top \mu)^\beta$$

$$\times \left[ \frac{(j-\alpha)!(u^\top \Sigma v)^{j-\alpha}}{u^\top \Sigma u v^\top \Sigma v} \mathbf{1}_{\text{j}-\alpha=\text{k}-\beta} - \frac{(u^\top \Sigma u - 1)(v^\top \Sigma v - 1)}{u^\top \Sigma u v^\top \Sigma v} \mathbb{E}_g[(\sqrt{u^\top \Sigma u} \xi)^{j-\alpha} (\sqrt{v^\top \Sigma v} \xi)^{k-\beta}] \right.$$

$$\left. + \frac{(v^\top \Sigma v - 1)}{v^\top \Sigma v} \mathbb{E}_g[H_{j-\alpha}(\sqrt{u^\top \Sigma u} \xi)(\sqrt{v^\top \Sigma v} \xi)^{k-\beta}] + \frac{(u^\top \Sigma u - 1)}{u^\top \Sigma u} \mathbb{E}_g[(\sqrt{u^\top \Sigma u} \xi)^{j-\alpha} H_{k-\beta}(\sqrt{v^\top \Sigma v} \xi)] \right]$$

(59)

*Proof of Theorem H.10.* By reparametrization i.e. $Z_1 = \sqrt{u^\top \Sigma u} \xi + u^\top \mu$, $Z_2 = \sqrt{v^\top \Sigma v} \xi + v^\top \mu$, and Lemma H.9,

$$H_j(\sqrt{u^\top \Sigma u} \xi + u^\top \mu) = \sum_{\alpha=0}^{j} \binom{j}{\alpha} (u^\top \mu)^\alpha H_{j-\alpha}(\sqrt{\mu^\top \Sigma u} \xi). \tag{60}$$

$$\mathbb{E}_g[H_j(u^\top g) H_k(v^\top g)] = \mathbb{E}_\xi[H_j(\sqrt{u^\top \Sigma u} \xi + u^\top \mu) H_k(\sqrt{v^\top \Sigma v} \xi + v^\top \mu)]$$

$$= \mathbb{E}_\xi\Big[ \sum_{\alpha=0}^{j} \binom{j}{\alpha} (u^\top \mu)^\alpha H_{j-\alpha}(\sqrt{\mu^\top \Sigma u} \xi) \Big] \Big[ \sum_{\beta=0}^{k} \binom{k}{\beta} (v^\top \mu)^\beta H_{k-\beta}(\sqrt{\mu^\top \Sigma v} \xi) \Big]$$

$$= \sum_{\alpha=0}^{j} \sum_{\beta=0}^{k} \binom{j}{\alpha} \binom{k}{\beta} (u^\top \mu)^\alpha (v^\top \mu)^\beta \mathbb{E}_\xi[H_{j-\alpha}(\sqrt{\mu^\top \Sigma u} \xi) H_{k-\beta}(\sqrt{\mu^\top \Sigma v} \xi)]$$

(61)

Use same proof technique Thm. H.5, with $\left( \begin{smallmatrix} \sqrt{u^\top \Sigma u} \xi \\ \sqrt{v^\top \Sigma v} \xi \end{smallmatrix} \right) \sim \mathcal{N}\left( \begin{pmatrix} 0 \\ 0 \end{pmatrix}, \begin{pmatrix} u^\top \Sigma u & u^\top \Sigma v \\ v^\top \Sigma u & v^\top \Sigma v \end{pmatrix} \right)$

$$\mathbb{E}_\xi[H_{j-\alpha}(\sqrt{u^\top \Sigma u}\xi)H_{k-\beta}(\sqrt{v^\top \Sigma v}\xi)]$$

$$= \frac{(j-\alpha)!(u^\top \Sigma v)^{j-\alpha}}{u^\top \Sigma u v^\top \Sigma v}\mathbf{1}_{j-\alpha=k-\beta} - \frac{(u^\top \Sigma u - 1)(v^\top \Sigma v - 1)}{u^\top \Sigma u v^\top \Sigma v}\mathbb{E}_g[(\sqrt{u^\top \Sigma u}\xi)^{j-\alpha}(\sqrt{v^\top \Sigma v}\xi)^{k-\beta}]$$

$$+ \frac{(v^\top \Sigma v - 1)}{v^\top \Sigma v}\mathbb{E}_g[H_{j-\alpha}(\sqrt{u^\top \Sigma u}\xi)(\sqrt{v^\top \Sigma v}\xi)^{k-\beta}] + \frac{(u^\top \Sigma u - 1)}{u^\top \Sigma u}\mathbb{E}_g[(\sqrt{u^\top \Sigma u}\xi)^{j-\alpha}H_{k-\beta}(\sqrt{v^\top \Sigma v}\xi)]$$

(62)

In summery,

$$\mathbb{E}_g[H_j(u^\top g)H_k(v^\top g)]$$

$$= \sum_{\alpha=0}^{j}\sum_{\beta=0}^{k}\binom{j}{\alpha}\binom{k}{\beta}(u^\top \mu)^\alpha(v^\top \mu)^\beta$$

$$\times \left[\frac{(j-\alpha)!(u^\top \Sigma v)^{j-\alpha}}{u^\top \Sigma u v^\top \Sigma v}\mathbf{1}_{j-\alpha=k-\beta} - \frac{(u^\top \Sigma u - 1)(v^\top \Sigma v - 1)}{u^\top \Sigma u v^\top \Sigma v}\mathbb{E}_\xi[(\sqrt{u^\top \Sigma u}\xi)^{j-\alpha}(\sqrt{v^\top \Sigma v}\xi)^{k-\beta}]\right.$$

$$\left. + \frac{(v^\top \Sigma v - 1)}{v^\top \Sigma v}\mathbb{E}_\xi[H_{j-\alpha}(\sqrt{u^\top \Sigma u}\xi)(\sqrt{v^\top \Sigma v}\xi)^{k-\beta}] + \frac{(u^\top \Sigma u - 1)}{u^\top \Sigma u}\mathbb{E}_\xi[(\sqrt{u^\top \Sigma u}\xi)^{j-\alpha}H_{k-\beta}(\sqrt{v^\top \Sigma v}\xi)]\right]$$

(63)

$\square$

The following Corollary which calculates the Expectation of the Power of a Gaussian Random Variable can be derived using the binomial expansion with the reparametrization technique and Corollary H.8. It corresponds to the case $k = 0$ in Lemma H.7.

**Corollary H.11** (Corollary of Lemma H.7). *Given $\beta$, let Gaussian Random Variable $Z \sim \mathcal{N}(\mu^\top \beta, \beta^\top \Sigma \beta)$, then expectation of power of $Z$ is*

$$\mathbb{E}_Z(Z)^k = \sum_{t=0}^{k}\binom{k}{t}(\mu^\top \beta)^{k-t}\mathbb{E}_{\bar{Z}\sim\mathcal{N}(0,\beta^\top \Sigma \beta)}[\bar{Z}^t]$$

$$= \sum_{t=0}^{k}\binom{k}{t}(\mu^\top \beta)^{k-t}(t-1)!! \cdot (\beta^\top \Sigma \beta)^{\frac{t}{2}}\mathbf{1}_{t \text{ is even}}.$$

(64)

The following corollary, which computes the Gaussian expectation of Hermite polynomials, is derived from the explicit form of Hermite polynomials and Corollary H.11. It corresponds to the case $k = 0$ in Theorem H.10.

**Corollary H.12** (Corollary of Theorem H.10). *For $d \in \mathbb{N}$, given $w \in \mathbb{R}^d$, $x \sim \mathcal{N}(\mu, \Sigma)$,*

$$\mathbb{E}_x[H_n(w^\top x)] = \sum_{m=0}^{\lfloor \frac{n}{2} \rfloor}\sum_{i=0}^{n-2m}\frac{(-1)^m(i-1)!! \, n!}{2^m \, m!(n-2m)!}\binom{n-2m}{i}(w^\top \mu)^{n-2m-i}(w^\top \Sigma w)^{\frac{i}{2}}\mathbf{1}_{i \text{ is even}}$$

(65)

# I  DETAIL OF ALIGNMENT ANALYSIS

*Proof of Theorem 4.2.* Let $\mathfrak{d}_k \triangleq c_1^k c_k \eta^k$.

When $x \sim \mathcal{n}(\mu, \Sigma)$, we approximate $F$ to dominant term $F_l$, then

$$\mathbb{E}_{x,x',\theta}[F_l(x)^\top F_l(x')]$$

$$= \mathbb{E}_{x,x',\theta}[(F_0(x) + \sum_{k=1}^{l} \daleth_k (\beta^\top x)^k \alpha^{\circ k})^\top (F_0(x') + \sum_{k=1}^{l} \daleth_k (\beta^\top x')^k \alpha^{\circ k})^\top]$$

$$= \mathbb{E}_{x,x',\theta}\Big[ \langle F_0(x), F_0(x') \rangle$$

$$+ \langle F_0(x), \sum_{k=1}^{l} \daleth_k (\beta^\top x')^k \alpha^{\circ k} \rangle + \langle F_0(x'), \sum_{k=1}^{l} \daleth_k (\beta^\top x)^k \alpha^{\circ k} \rangle + \sum_{k=1,j=1}^{l} \daleth_k \daleth_j (\beta^\top x)^k (\beta^\top x')^j \sum_r \alpha[r]^{j+k} \Big]$$

$$\triangleq \mathscr{A} + \mathscr{B} + \mathscr{C}$$

$$(66)$$

## I.1 $\quad \mathscr{A}: \mathbb{E}_\theta\left[ \mathbb{E}_x[\sigma(W_0^\top x)]^\top \mathbb{E}_{x'}[\sigma(W_0^\top x')] \right]$ STUDY

Let $\daleth_{n,m,i} \triangleq \frac{(-1)^m (i-1)!! \, n!}{2^m \, m!(n-2m)!} \binom{n-2m}{i}$.

By Corollary H.12,

$$\mathscr{A} = \mathbb{E}_\theta\Big[ \sum_{a=1}^{\mathbf{N}} \sum_{n=1}^{\infty} \sum_{m=0}^{\lfloor \frac{n}{2} \rfloor} \sum_{i=0}^{n-2m} \sum_{o=1}^{\infty} \sum_{p=0}^{\lfloor \frac{o}{2} \rfloor} \sum_{q=0}^{o-2p} c_n c_o \daleth_{n,m,i} \daleth_{o,p,q}$$

$$\times (W_0[a]^\top \mu)^{n-2m-i} (W_0[a]^\top \Sigma W_0[a])^{i/2} \mathbf{1}_{\text{i is even}}$$

$$\times (W_0[a]^\top \mu)^{o-2p-q} (W_0[a]^\top \Sigma W_0[a])^{q/2} \mathbf{1}_{\text{q is even}} \Big]$$

$$(67)$$

We know, $W_0 \sim \mathbb{R}^{\mathbf{d} \times \mathbf{N}}$, $W_0[i] \sim \text{Unif}(\mathbb{S}^{\mathbf{d}-1})$.

Denote $w \triangleq W_0[a]$, $t = n + o - 2m - 2p - i - q$, $k = \frac{i+q}{2}$.

$$\mathbb{E}\left[ (W_0[a]^\top \mu)^{n-2m-i} (W_0[a]^\top \Sigma W_0[a])^{i/2} (W_0[a]^\top \mu)^{o-2p-q} (W_0[a]^\top \Sigma W_0[a])^{q/2} \right]$$

$$= \mathbb{E}[(w^\top \mu)^t (w^\top \Sigma w)^k]$$

$$(68)$$

Use covariance matrix property, which is diagonalizable i.e. $\Sigma = Q\Lambda Q$. $w^\top \Sigma w = w^\top Q \Lambda Q^\top w$. Let $z = Q^\top w$ and $\tilde{\mu} = Q^\top \mu$ then $w^\top \Sigma w = z^\top \Lambda z = \sum_i \lambda_i z_i^2$. By symmetry, $z \sim \text{Unif}(\mathbb{S}^{\mathbf{d}-1})$

Therefore, using multi-index notation, where $|\alpha| = \sum_{i=1}^{n} \alpha_i$, $\alpha_i \geq 0$, and $\binom{t}{\alpha} = \frac{t!}{\alpha_1! \alpha_2! \cdots \alpha_n!}$

$$\mathbb{E}_w[(w^\top \mu)^t (w^\top \Sigma w)^k] = \mathbb{E}_z[(z^\top \tilde{\mu})^t (\sum_i \lambda_i z_i^2)^k]$$

$$= \mathbb{E}_z[(\sum_{|\alpha|=t} \binom{t}{\alpha} \prod_{i=1}^{d} (z_i \tilde{\mu}_i)^{\alpha_i})(\sum_{|\beta|=k} \binom{k}{\beta} \prod_{j=1}^{d} (z_j^2 \lambda_j)^{\beta_j})$$

$$= \sum_{|\alpha|=t} \sum_{|\beta|=k} \binom{t}{\alpha} \binom{k}{\beta} \prod_{i=1}^{d} \prod_{j=1}^{d} \tilde{\mu}_i^{\alpha_i} \lambda_j^{\beta_j} \mathbb{E}_z[z_i^{\alpha_i} z_j^{2\beta_j}]$$

$$(69)$$

The term related to $\mu$ of $\mathscr{A}$ is associated with the random value $W[a]$. Therefore, taking expectation on network parameters, $\mathscr{A}$ only depends on unseen distribution parameter $\mu, \Sigma$ without train distribution.

In summery, let $S(r,s,i,j) = \mathbb{E}_z[z_i^r z_j^s]$,

$R(n,m,i,o,p,q,\alpha,\beta,c_n,c_o,\mathbf{N}) = \mathbf{N} c_n c_o \daleth_{n,m,i} \daleth_{o,p,q} \mathbf{1}_{\text{i,q are even}} \binom{n+o-2m-2p-i-q}{\alpha} \binom{\frac{i+q}{2}}{\beta}$, which are deterministic function, then

$$\mathscr{A} = \sum_{n=1}^{\infty} \sum_{m=0}^{\lfloor \frac{n}{2} \rfloor} \sum_{i=0}^{n-2m} \sum_{o=1}^{\infty} \sum_{p=0}^{\lfloor \frac{o}{2} \rfloor} \sum_{q=0}^{o-2p} \sum_{|\alpha|=n+o-2m-2p-i-q} \sum_{|\beta|=\frac{i+q}{2}}$$

$$R(n,m,i,o,p,q,\alpha,\beta,c_n,c_o,\mathbf{N}) \prod_{l=1}^{d} \prod_{j=1}^{d} \tilde{\mu}_l^{\alpha_l} \lambda_j^{\beta_j} S(\alpha_l,\beta_j,l,j) \tag{70}$$

## I.2  $\mathscr{B} : 2\sum_{k=1}^{l} \mathfrak{d}_k \langle \mathscr{B}_k, \mathbb{E}_\alpha[a^{\circ k}] \rangle$ STUDY

$\mathscr{B}_k$: $\mathbb{E}_{x,x'}[\sigma(W_0^\top x)(\beta^\top x')^k]$ **study**
Let $Z_1 = \langle W_0[a], x \rangle$, $Z_2 = \langle \beta, x' \rangle$,

then $Z_1|W_0[a] \sim \mathcal{N}(W_0[a]^\top \mu, W_0[a]^\top \Sigma W_0[a])$, $Z_2 \sim \mathcal{N}(\beta^\top \mu, \beta^\top \Sigma \beta)$.

Therefore, by Corollary H.11 and H.12

$$\mathscr{B}_k[a] = \mathbb{E}_\theta \sum_{j=1}^{\infty} c_j \mathbb{E}_{Z_1} H_j(Z_1) \mathbb{E}_{Z_2}(Z_2)^k$$

$$= \sum_{j=1}^{\infty} \sum_{m=0}^{\lfloor \frac{j}{2} \rfloor} \sum_{i=0}^{j-2m} \sum_{t=0}^{k} \binom{j-2m}{i} \binom{k}{t} \frac{(-1)^m c_j (i-1)!!\, j!\, (k-t-1)!}{2^m\, m!(j-2m)!} \mathbf{1}_{k-t \text{ is even}} \mathbf{1}_{i \text{ is even}}$$

$$\mathbb{E}_\theta \left[ (W_0[a]^\top \mu)^{j-2m-i} (W_0[a]^\top \Sigma W_0[a])^{\frac{i}{2}} \right] (\mu^\top \beta)^t \cdot (\beta^\top \Sigma \beta)^{\frac{k-t}{2}}$$

$$= \sum_{j=1}^{\infty} \sum_{m=0}^{\lfloor \frac{j}{2} \rfloor} \sum_{i=0}^{j-2m} \sum_{t=0}^{k} \sum_{|\alpha|=j-2m-i} \sum_{|\beta|=\frac{i}{2}} \binom{j-2m}{i} \binom{k}{t} \binom{j-2m-i}{\alpha} \binom{\frac{i}{2}}{\beta} \frac{(-1)^m c_j (i-1)!!\, j!\, (k-t-1)!}{2^m\, m!(j-2m)!}$$

$$\prod_{u=1}^{d} \prod_{v=1}^{d} \tilde{\mu}_u^{\alpha_u} \lambda_v^{\beta_v} S(\alpha_u,\alpha_v,u,v)\, (\mu^\top \beta)^t \cdot (\beta^\top \Sigma \beta)^{\frac{k-t}{2}} \mathbf{1}_{k-t \text{ is even}} \mathbf{1}_{i \text{ is even}} \tag{71}$$

The term related to $\mu$ of $\mathscr{B}_k[a]$ is associated with the random value $W[a]$. Therefore, $\forall a$, $\mathscr{B}_k[a]$ depends on unseen distribution parameter $\mu, \Sigma$ and $\beta^\top \mu$ with same value.

$\mathbb{E}_\alpha[a^{\circ k}]$ **study**  We know $a_1[i], a_2[i] \sim \mathcal{N}(0, \frac{1}{\mathbf{N}})$, so $\alpha[i] \triangleq (a_1 - a_2)[i] \sim \mathcal{N}(0, \frac{2}{\mathbf{N}})$. Therefore, by centered gaussian moments,

$$\mathbb{E}_\alpha[\alpha[r]^k] = \frac{(k)!}{w^{\frac{k}{2}} (\frac{k}{2})!} (\frac{2}{N})^{\frac{k}{2}} \mathbf{1}_{k \text{ is even}} \tag{72}$$

Since $\mathbb{E}_\alpha[\alpha[r]^k]$ is nonzero only when $k$ is even, and even condition of $k-t$ is exist in $\mathscr{B}_k[a]$, taking the absolute value of $\beta^T \mu$ within $\mathscr{B}_k$ produce equivalent results.

Therefore, in $\mathscr{B} = 2\sum_{k=1}^{l} \mathfrak{d}_k \langle \mathscr{B}_k, \mathbb{E}_\alpha[a^{\circ k}] \rangle$ is depends on $\mu, \Sigma, |\beta^\top \mu|$ and $\beta^\top \Sigma \beta$

## I.3  $\mathscr{C}$: $\sum_{k=1,j=1}^{l} \mathfrak{d}_k \mathfrak{d}_j \mathbb{E}_\alpha[\sum_r \alpha[r]^{j+k}] \mathscr{C}_{j,k}$ STUDY

$\mathscr{C}_{j,k}$: $\mathbb{E}_{x,x'}[(\beta^\top x)^k (\beta^\top x')^j]$ **study**
Let $Z_1 \triangleq \beta^\top x \sim \mathcal{N}(\beta^\top \mu, \beta^\top \Sigma \beta)$, same as $Z_2'$. Using Corollary H.11,

$$\mathbb{E} Z_1^j Z_2^k = \sum_{s=0}^{j} \sum_{t=0}^{k} \binom{j}{s} \binom{k}{t} (j-s-1)!!(k-t-1)!!(\mu^\top \beta)^{s+t} (\beta^\top \Sigma \beta)^{\frac{j-s+k-t}{2}} \mathbf{1}_{j-s,\, k-t \text{ are even}} \tag{73}$$

Therefore, the term related to $\mu$ in $\mathscr{C}_{j,k}$ is only dominated by the discriminative data $\beta$, independent of the randomly initialized parameters.

$\sum_r \mathbb{E}_\alpha[\alpha[r]^{j+k}]$ **study**

$$\sum_r \mathbb{E}_\alpha[\alpha[r]^{j+k}] = \frac{\mathbf{N}(j+k)!}{2^{\frac{j+k}{2}}(\frac{j+k}{2})!}\left(\frac{2}{\mathbf{N}}\right)^{\frac{j+k}{2}} \mathbf{1}_{\text{j+k and i is even}} \tag{74}$$

Since $\mathbb{E}_\alpha[\alpha[r]^{j+k}]$ is nonzero only when $j+k$ is even, and even condition of $j-s$ and $k-t$ are exist in $\mathscr{C}_k[a]$, so $s+t$ is even in this conditions, taking the absolute value of $\beta^T \mu$ within $\mathscr{C}_k$ produce equivalent results.

Therefore, in $\mathscr{C} = \sum_{k=1,j=1}^l \natural_k \natural_j \mathbb{E}_\alpha[\sum_r \alpha[r]^{j+k}]\mathscr{C}_{j,k}$ is depends on $|\beta^\top \mu|$ and $\beta^\top \Sigma \beta$

$\square$

## J    DETAIL OF LOCAL ELASTICITY ANALYSIS

*Proof of Theorem 4.3.*

$$\mathbb{E}_{x,\theta}||F_l(x) - F_0(x)||^2 = \mathbb{E}_{x,\theta}[\sum_{k=1}^l c_1^k c_k \eta^k (\beta^\top x)^k (\alpha^{\circ k})]^\top [\sum_{m=1}^l c_1^m c_m \eta^m (\beta^\top x)^m (\alpha^{\circ m})]. \tag{75}$$

For $\mathbb{E}[(x^\top \beta)^{k+m}]$, by Corollary H.11

$$\begin{aligned}
\mathbb{E}_x[(x^\top \beta)^{k+m}] &= \mathbb{E}_{z\sim n(0,1)}[(\beta^\top \mu + \sqrt{\beta^\top \Sigma \beta}z)^{k+m}] \\
&= \sum_{i=0}^{k+m}\binom{k+m}{i}(\beta^\top \mu)^{k+m-i}(\beta^\top \Sigma \beta)^{\frac{i}{2}}\mathbb{E}[z^i] \\
&= \sum_{i=0}^{k+m}\binom{k+m}{i}(\beta^\top \mu)^{k+m-i}(\beta^\top \Sigma \beta)^{\frac{i}{2}}(i-1)!! \, \mathbf{1}_{\text{i is even}}
\end{aligned} \tag{76}$$

Remark $\natural_k \triangleq c_1^k c_k \eta^k$. Finally,

$\mathbb{E}_{x,\theta}||F_l(x) - F_0(x)||^2$

$$= \sum_{k=1}^l \sum_{m=1}^l c_1^{k+m} c_k c_m \eta^{k+m} \sum_{i=0}^{k+m}\binom{k+m}{i}(\beta^\top \mu)^{k+m-i}(\beta^\top \Sigma \beta)^{\frac{i}{2}}(i-1)!! \, \mathbf{1}_{\text{i is even}}\mathbb{E}_\alpha[\alpha^{\circ k\top}\alpha^{\circ m}]$$

$$= \sum_{k=1}^l \sum_{m=1}^l \sum_{i=0,\text{ even}}^{k+m} \natural_k \natural_m \binom{k+m}{i}(\beta^\top \mu)^{k+m-i}(\beta^\top \Sigma \beta)^{\frac{i}{2}}(i-1)!! \sum_r \mathbb{E}_\alpha[\alpha[r]^{k+m}]. \tag{77}$$

Taking Expectation over Network parameters, one can acquire

$$= \sum_{k=1}^l \sum_{m=1}^l \sum_{i=0}^{k+m} \natural_k \natural_m \binom{k+m}{i}(\beta^\top \mu)^{k+m-i}(\beta^\top \Sigma \beta)^{\frac{i}{2}}(i-1)!! \frac{\mathbf{N}(k+m)!}{2^{\frac{k+m}{2}}(\frac{k+m}{2})!}\left(\frac{2}{\mathbf{N}}\right)^{\frac{k+m}{2}}\mathbf{1}_{\text{k+m and i is even}} \tag{78}$$

Therefore, $k+m-i$ is even. For clarity, we use absolute values,

$$= \sum_{k=1}^l \sum_{m=1}^l \sum_{i=0}^{k+m} \natural_k \natural_m \binom{k+m}{i}\frac{\mathbf{N}(k+m)!}{2^{\frac{k+m}{2}}(\frac{k+m}{2})!}\left(\frac{2}{\mathbf{N}}\right)^{\frac{k+m}{2}}(i-1)!! \, |\beta^\top \mu|^{k+m-i}(\beta^\top \Sigma \beta)^{\frac{i}{2}} \, \mathbf{1}_{\text{k+m and i is even}} \tag{79}$$

For clearity, we define constant

$$\kappa_{LE}(k,m,i,\mathbf{N},c_1,c_k,c_m,\eta) \triangleq \natural_k \natural_m \binom{k+m}{i}\frac{\mathbf{N}(k+m)!}{2^{\frac{k+m}{2}}(\frac{k+m}{2})!}\left(\frac{2}{\mathbf{N}}\right)^{\frac{k+m}{2}}(i-1)!!$$

which depends on constant $k, m, i, \mathbf{N}, c_1, c_k, c_m, \eta$.

$$= \sum_{k=1}^{l} \sum_{m=1}^{l} \sum_{i=0}^{k+m} \kappa_{LE} \ |\beta^{\top}\mu|^{k+m-i} (\beta^{\top}\Sigma\beta)^{\frac{i}{2}} \ \mathbf{1}_{\text{k+m and i is even}} \tag{80}$$

$\kappa_{LE}$ depends only on the constants $k, m, i, \mathbf{N}, c_1, c_k, c_m, \eta$, and is independent of the parameters of the data distribution.

$\square$

## K  ADDITIONAL INFORMATION OF EXPERIMENT 1, 2

### K.1  ADDITIONAL RESULTS FOR ALIGNMENT AND ELASTICITY

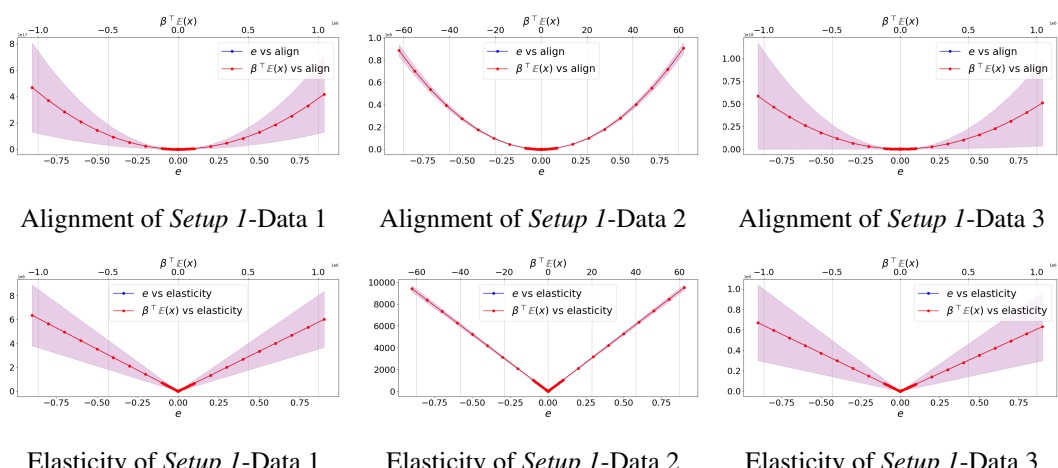

Figure K.1: **Setup 1** Observation of Alignment and Elasticity (y-axis) derived from the LHS of Thm. 1.1, 1.2 across different $e$ (blue, lower x-axis, exactly overlaped) and $\beta^{\top}\mu$ (red, upper x-axis) values.

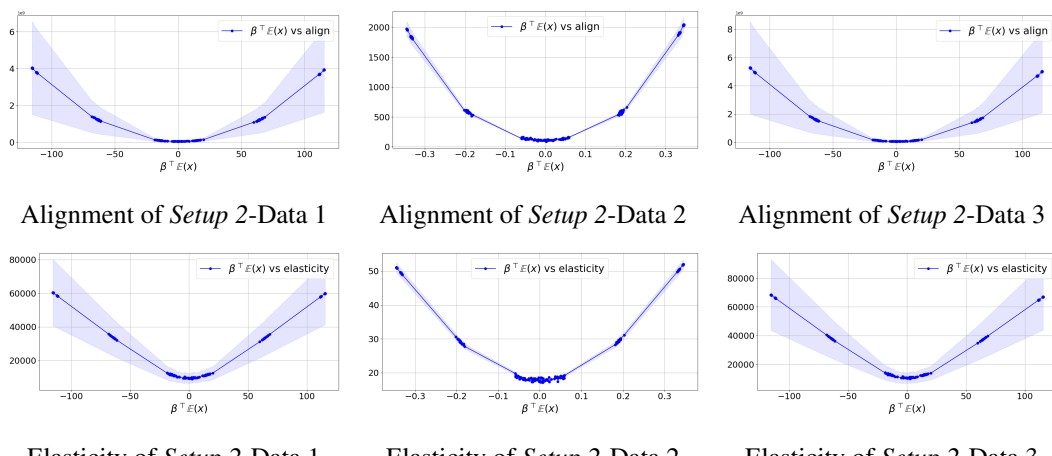

Figure K.2: **Experiment 2** The observation of Alignment and Elasticity (y-axis), derived from the LHS of Thms. 1.1 and 1.2, across different values of $\beta^{\top}\mu$ (x-axis) with varying $R$.

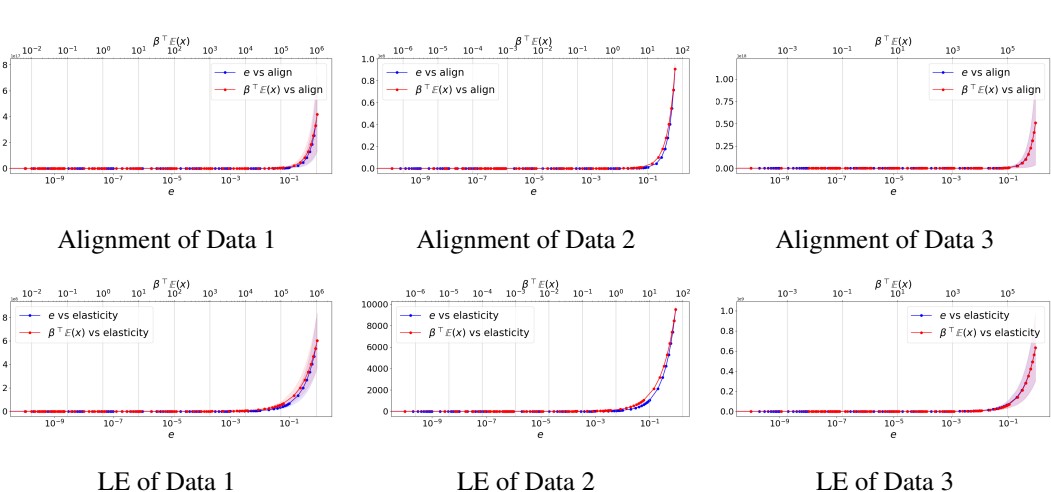

Figure K.3: **Experiment 1** The x-axis is displayed on a logarithmic scale. Observation of Alignment and LE (y-axis) derived from the LHS of Thm. 4.2, 4.3 across different $e$ (blue, lower x-axis) and $\beta^\top \mu$ (red, upper x-axis) values.

## K.2 ADDITIONAL RESULTS FOR RECALL@1

We present the cosine similarity Recall@1 experiment for the remaining datasets not included in the main text in this section Figure K.4.

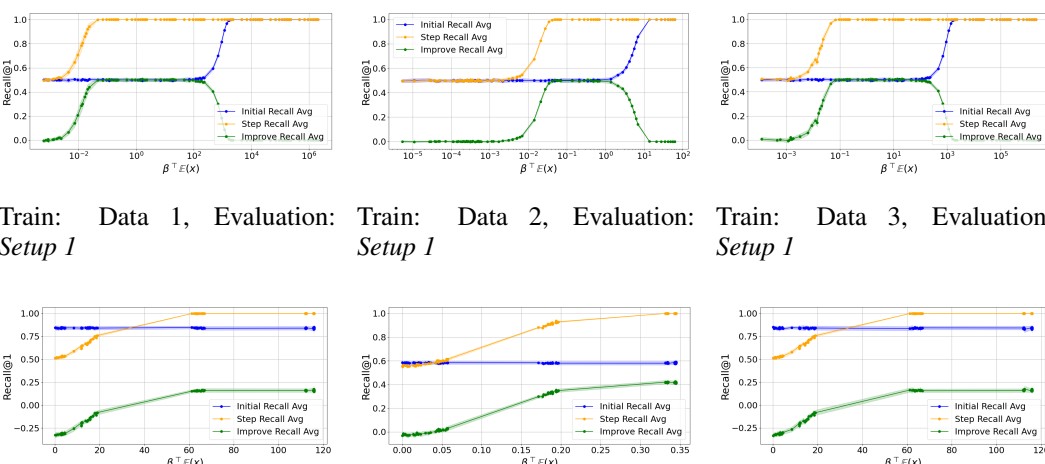

Train: Data 1, Evaluation: *Setup 1*          Train: Data 2, Evaluation: *Setup 1*          Train: Data 3, Evaluation: *Setup 1*

Train: Data 1, Evaluation: *Setup 2*          Train: Data 2, Evaluation: *Setup 2*          Train: Data 3, Evaluation: *Setup 2*

Figure K.4: Recall@1 (y-axis) measurement of **Exper 1, 2** of features across different $\beta^\top \mu$ values (x-axis). The blue line represents the clustering performance measured using the features in their initialized state, the orange line reflects the performance after one step of training, and the green line indicates the improvement, i.e., the difference between the two. For *Setup 1* (top), the x-axis is on a logarithmic scale, whereas for *Setup 2* (bottom), the x-axis is on a linear scale.

### K.2.1 INNER PRODUCT RECALL@1 OF EXPERIMENT 1

In this experiment we use Recall@1 with Inner Product similarity. Figure K.5. Similar trends are observed in the Recall@1 of the Inner Product similarity as in the Cosine similarity. The Recall@1 of the Inner Product similarity is also maximized when the alignment is high.

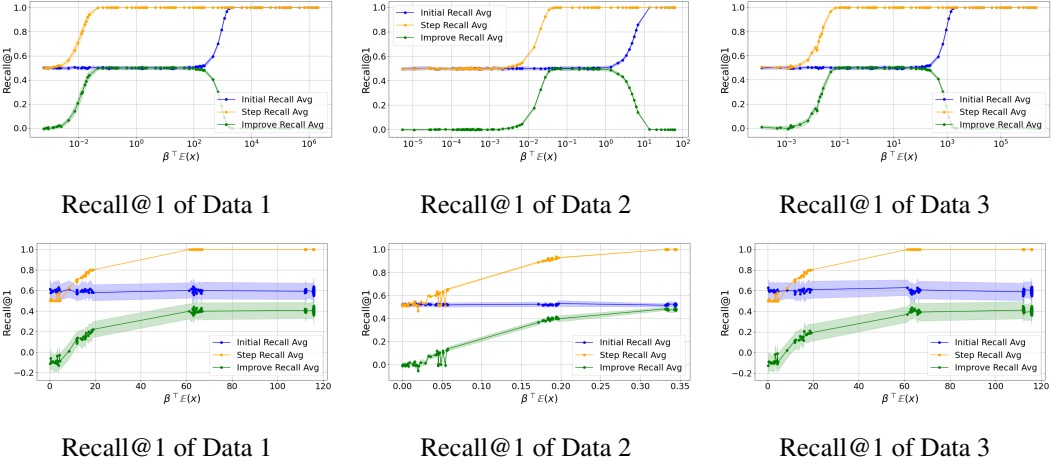

Recall@1 of Data 1          Recall@1 of Data 2          Recall@1 of Data 3

Recall@1 of Data 1          Recall@1 of Data 2          Recall@1 of Data 3

Figure K.5: Recall@1 measurement using Inner Product similarity

### K.3 Empirical Validation of the Linear Relationship between Generated Data Parameter $e$ and $\beta^\top \mu$ in Experiment 1

As shown in Figure K.6, we observe a positive, linear relationship between $e$ and $\beta^\top \mu$ as $e$ is varied. This confirms the validity of our test data generation method based on $e$.

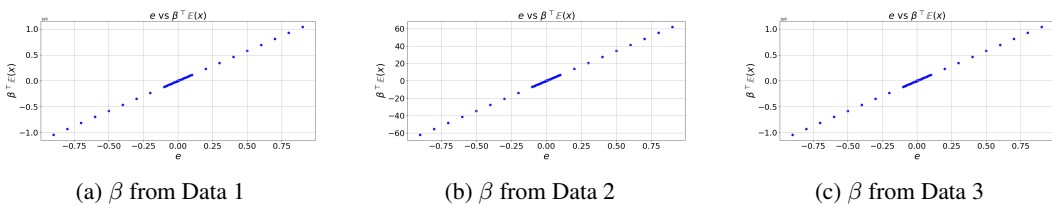

(a) $\beta$ from Data 1    (b) $\beta$ from Data 2    (c) $\beta$ from Data 3

Figure K.6: We calculated $\beta$ from Training Datasets 1, 2, and 3, and then computed $\beta^\top \mu$ by adjusting $e$ to determine $\mu$ in the test data. The x-axis represents $e$, and the y-axis shows the values of $\beta^\top \mu$.

### K.4 Rotation Matrix Generation Process of *Setup 2*

To generate a set of rotation matrices with diverse magnitudes of rotation, we constructed an algorithm that samples $k = 300$ random matrices, each formed by adding i.i.d. Gaussian noise matrix of varying variance to the identity matrix $I$. The process ensures the generation of rotation matrices with varying extents of rotation, from slight to more substantial deviations from the identity matrix.

The rotation matrices are generated as follows:

1. A matrix is initialized as $I + \epsilon \cdot M$, where $M$ is a i.i.d. standard random Gaussian matrix.

2. Using the QR decomposition, we orthogonalize this matrix to ensure it forms a valid rotation matrix.

3. Finally, if the determinant of the resulting matrix is negative, we flip the sign of the first column to maintain a determinant of $+1$, ensuring it is a valid rotation.

In summary, this method provides a collection of matrices that progressively deviate from $I$, allowing us to observe and sample rotations of increasing magnitude.

### K.5 Additional Discussion of Recall@1 evaluation for Expr 1

The Recall@1 results of Expr 1 setting indicate three phases in Recall@1 outcomes.

**The first phase**: The learning process fails to improve performance either because the training and evaluation data are too distant, as predicted by our theory, or because e is too small for the feature extractor to achieve separation. We interpret that either of these factors contributes to the lack of performance improvement. **The second phase**: Performance improves as the similarity between training and evaluation data becomes appropriate, allowing better Recall@1 after training. It is noteworthy that the improvement also increases along with larger $e$. This indicates not only the increased $e$ leading to greater distances between evaluation features but also the Recall@1 improvement with $e$ as our theory. **The third phase**: Effective feature separation has already occurred; thus, even with sufficiently close training data, the learning process does not enhance Recall@1 performance.

We conclude that the first and third phases represent unsuitable configurations for retrieval tasks, while the second phase provides a dataset which effectively supports training for retrieval tasks and is explainable by our theory.

## L Additional Settings for Experiment 3, 4

In Table 2, we provide the detailed parameters for the experiments.

| | Experiment 3 | Experiment 4 |
|---|---|---|
| **Model** | ResNet18 | ResNet50 |
| **Learning Task** | Binary Classification | Multi-class Classification |
| **Loss** | Mean Squared Error | Norm Softmax (Zhai & Wu, 2019) |
| **Epoch** | 20 | 30 |
| **Batch Size** | 96 (Full GD) | 75 (use 3 classes with 25 samples) |
| **Optimizer** | Adam | SGD |
| **Learning Rate** | 0.001 | CARS: 0.01 / CUB: 0.001 |

Table 2: Comparison between Experiment 3 and Experiment 4

## M    ADDITIONAL INFORMATION FOR SETUP 3

For gradient stability and fair evaluation, all classes are truncated to include only 48 images. The batch size is set to 96 for Full gradient descent. Remark that, to align the experimental setup with our theoretical setting, two classifier heads and sign flipped label $1, -1$ is used.

## N    ADDITIONAL RESULTS OF EXPERIMENT 3

The performance of the two classifier heads during training is shown in Figure N.1. The results of the empirical validation without Kendall's W aggregation are presented in subsection N.1 for the model trained with the CARS196 dataset and in subsection N.2 for the model trained with the CUB200 dataset. Consistent with Kendall's $W$ calculations and theoretical analyses, in most cases, we observe that LE, alignment, and $|\beta^\top \mu|$ individually rise and fall in similar trends during training. The gray line represents metrics calculated on the entire dataset, while the colored lines denote individual test classes. Since classes were randomly sampled per seed, the same color represents the same class only within a single seed.

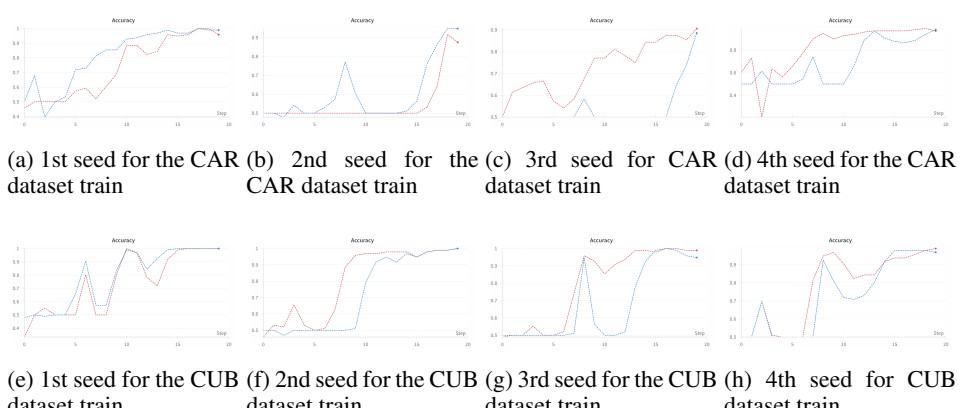

(a) 1st seed for the CAR dataset train
(b) 2nd seed for the CAR dataset train
(c) 3rd seed for CAR dataset train
(d) 4th seed for the CAR dataset train

(e) 1st seed for the CUB dataset train
(f) 2nd seed for the CUB dataset train
(g) 3rd seed for the CUB dataset train
(h) 4th seed for CUB dataset train

Figure N.1: Classification accuracy measured with training data. As two classifier heads were used in the theoretical setup, two accuracy values are plotted for each setting.

## N.1 TRAIN MODEL WITH CARS196

### N.1.1 1ST SEED

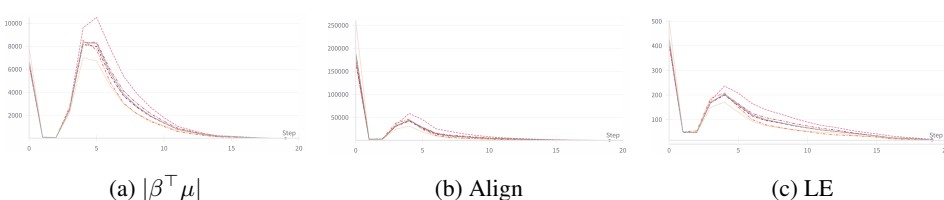

(a) $|\beta^\top \mu|$      (b) Align      (c) LE

Figure N.2: The 1st seed of the CAR dataset training. Results were computed using the features of five randomly selected classes from the CAR dataset's test set.

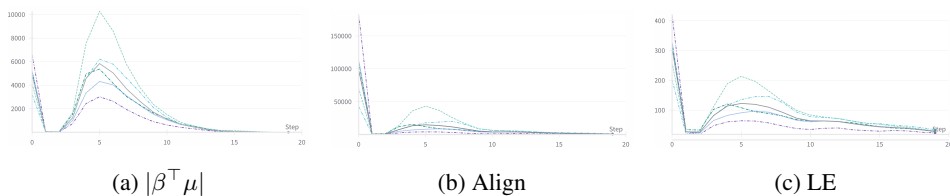

(a) $|\beta^\top \mu|$      (b) Align      (c) LE

Figure N.3: The 1st seed of the CAR dataset training. Results were computed using the features of five randomly selected classes from the CUB dataset's test set.

### N.1.2 2ND SEED

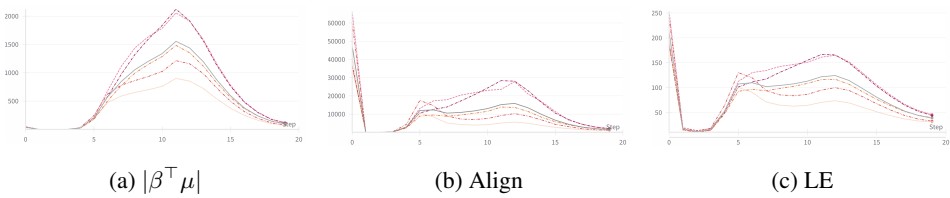

(a) $|\beta^\top \mu|$      (b) Align      (c) LE

Figure N.4: The 2nd seed of the CAR dataset training. Results were computed using the features of five randomly selected classes from the CAR dataset's test set.

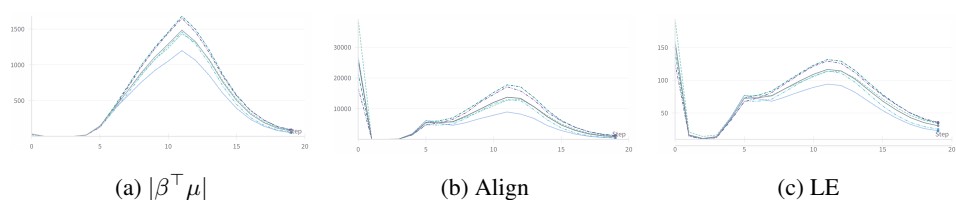

(a) $|\beta^\top \mu|$      (b) Align      (c) LE

Figure N.5: The 2nd seed of the CAR dataset training. Results were computed using the features of five randomly selected classes from the CUB dataset's test set.

### N.1.3 3RD SEED

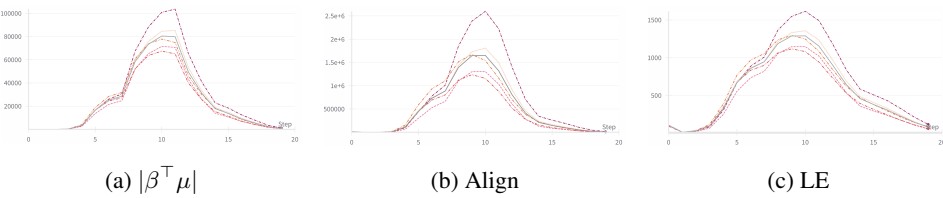

(a) $|\beta^\top \mu|$          (b) Align          (c) LE

Figure N.6: The 3rd seed of the CAR dataset training. Results were computed using the features of five randomly selected classes from the CAR dataset's test set.

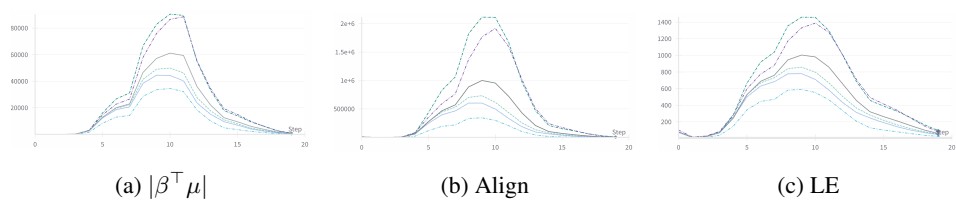

(a) $|\beta^\top \mu|$          (b) Align          (c) LE

Figure N.7: The 3rd seed of CAR dataset training. Results were computed using the features of five randomly selected classes from the CUB dataset's test set.

### N.1.4 4TH SEED

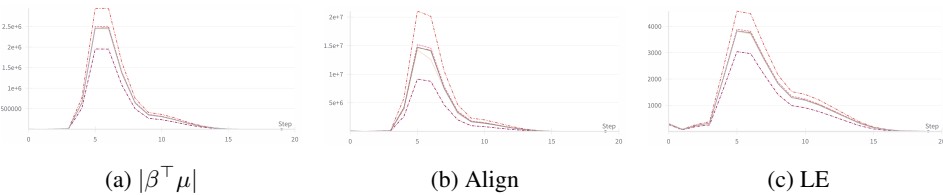

(a) $|\beta^\top \mu|$          (b) Align          (c) LE

Figure N.8: The 4th seed of the CAR dataset training. Results were computed using the features of five randomly selected classes from the CAR dataset's test set.

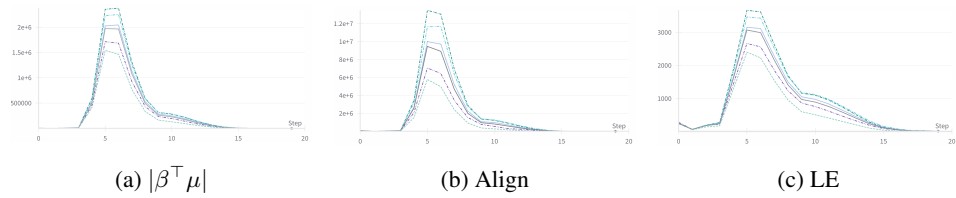

(a) $|\beta^\top \mu|$          (b) Align          (c) LE

Figure N.9: The 4th seed of the CAR dataset training. Results were computed using the features of five randomly selected classes from the CUB dataset's test set.

## N.2 TRAIN MODEL WITH CUB200

### N.2.1 1ST SEED

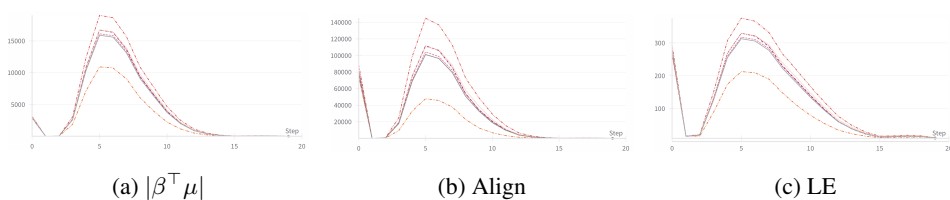

(a) $|\beta^\top \mu|$         (b) Align         (c) LE

Figure N.10: The 1st seed of the CUB dataset training. Results were computed using the features of five randomly selected classes from the CAR dataset's test set.

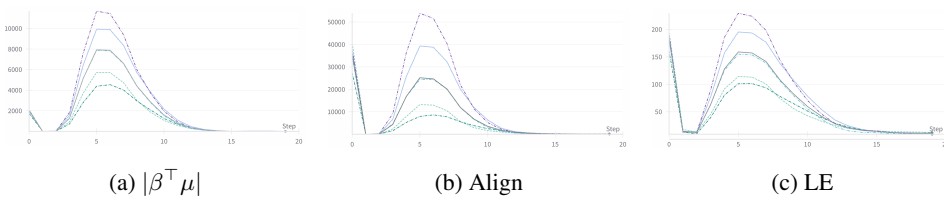

(a) $|\beta^\top \mu|$         (b) Align         (c) LE

Figure N.11: The 1st seed of the CUB dataset training. Results were computed using the features of five randomly selected classes from the CUB dataset's test set.

### N.2.2 2ND SEED

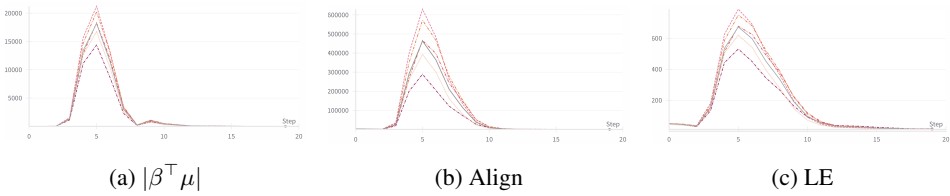

(a) $|\beta^\top \mu|$         (b) Align         (c) LE

Figure N.12: The 2nd seed of the CUB dataset training. Results were computed using the features of five randomly selected classes from the CAR dataset's test set.

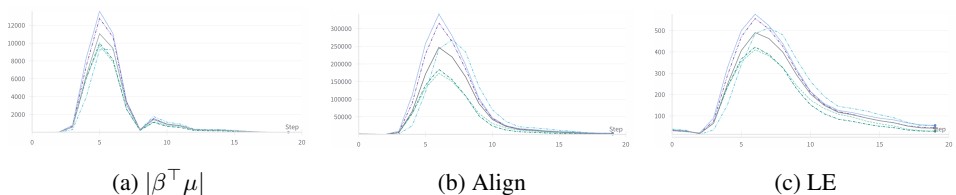

(a) $|\beta^\top \mu|$         (b) Align         (c) LE

Figure N.13: The 2nd seed of the CUB dataset training. Results were computed using the features of five randomly selected classes from the CUB dataset's test set.

### N.2.3  3RD SEED

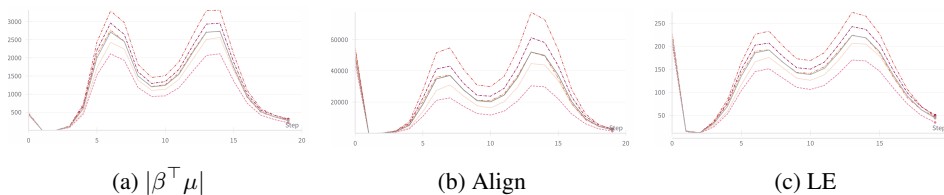

(a) $|\beta^\top \mu|$          (b) Align          (c) LE

Figure N.14: The 3rd seed of the CUB dataset training. Results were computed using the features of five randomly selected classes from the CAR dataset's test set.

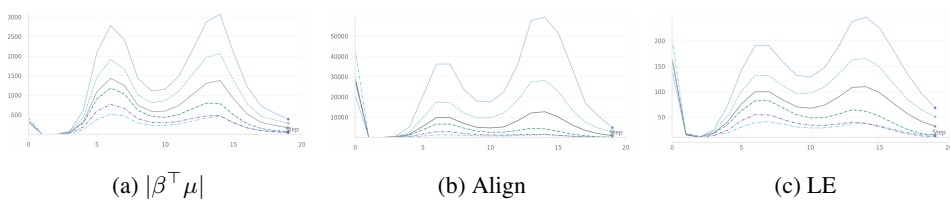

(a) $|\beta^\top \mu|$          (b) Align          (c) LE

Figure N.15: The 3rd seed of the CUB dataset training. Results were computed using the features of five randomly selected classes from the CUB dataset's test set.

### N.2.4  4TH SEED

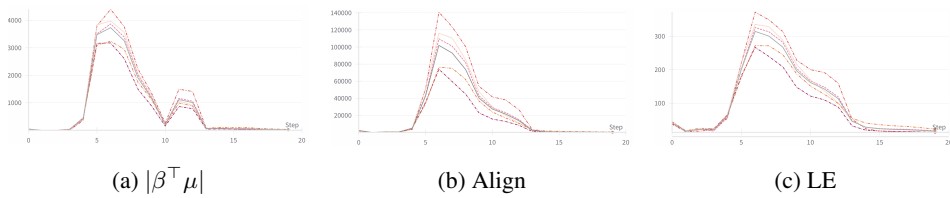

(a) $|\beta^\top \mu|$          (b) Align          (c) LE

Figure N.16: The 4th seed of the CUB dataset training. Results were computed using the features of five randomly selected classes from the CAR dataset's test set.

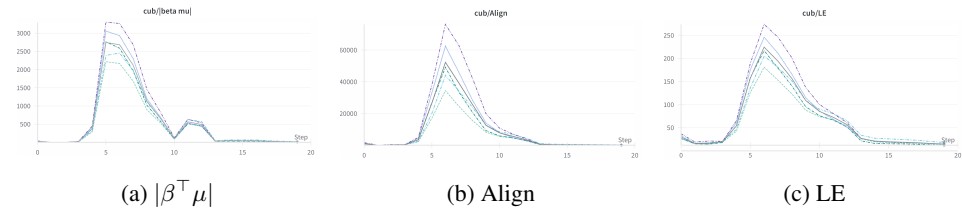

(a) $|\beta^\top \mu|$          (b) Align          (c) LE

Figure N.17: The 4th seed of the CUB dataset training. Results were computed using the features of five randomly selected classes from the the CUB dataset's test set.

## O   ADDITIONAL RESULTS OF EXPERIMENT 4

We provide non aggregated data for the experiment 4 in this section. The data is presented in the form of tables. The tables are as follows:

| R@1 v. Align | p-value | Recall@1 | Recall@2 | Recall@4 | Recall@8 | final Avg. Align |
|---|---|---|---|---|---|---|
| 0.3195 | 0.0013 | 93.4694 | 96.6056 | 97.9707 | 98.7947 | 829.4791 |
| 0.2386 | 0.0180 | 93.7523 | 96.4949 | 97.9461 | 98.7332 | 858.0315 |
| 0.1052 | 0.3025 | 94.0844 | 96.7409 | 98.1798 | 98.9792 | 857.6394 |
| 0.2864 | 0.0043 | 93.3096 | 96.4457 | 98.0445 | 98.8439 | 827.9364 |
| 0.2374±0.0942 | 0.0000 | 93.6539±0.3404 | 96.5718±0.1311 | 98.0353±0.1050 | 98.8378±0.1046 | 843.2716±16.8294 |

Table 3: Measurement from CARS196 trained model. R@1 v. Align is Pearson correlation between Recall@1 and final Avg. Align. Recall@k and final average Alignment is measured after training.

| R@1 v. Align | p-value | Recall@1 | Recall@2 | Recall@4 | Recall@8 | final Avg. Align |
|---|---|---|---|---|---|---|
| 0.2925 | 0.0032 | 68.0621 | 78.7643 | 86.5294 | 91.6948 | 1060.0284 |
| 0.2308 | 0.0209 | 68.6867 | 79.4564 | 87.6097 | 92.2687 | 1088.8525 |
| 0.3498 | 0.0004 | 67.9946 | 79.2539 | 86.7995 | 92.4038 | 1050.2296 |
| 0.2769 | 0.0053 | 67.7583 | 78.8150 | 87.0695 | 92.2181 | 1090.8598 |
| 0.2875±0.0491 | 0.0000 | 68.1254±0.3962 | 79.0724±0.3374 | 87.0020±0.4613 | 92.1464±0.3111$ | 1072.4926±20.4613 |

Table 4: Measurement from CUB200 trained model. R@1 v. Align is Pearson correlation between Recall@1 and final Avg. Align. Recall@k and final average Alignment is measured after training.

