# OpenReview forum: "Emergence of Alignment and Local Elasticity in Two-Layer Neural Networks"
_ICLR.cc/2025/Conference — Submitted to ICLR 2025_

### Official Review · Reviewer_YL7P · 2024-11-03

**Soundness:** 2
**Presentation:** 1
**Contribution:** 1
**Rating:** 3
**Confidence:** 5

**Summary:**

This paper studies two feature learning measurements: alignment and local elasticity of a two-layer neural network. They specifically show alignments between the neurons get improved within classes, and local elasticity emerges after a large gradient descent step. They consider the setting when the number of samples, the number of features and the input dimension go to infinity. They further implement numerical experiments to verify the phenomena.

**Strengths:**

1. Feature learning is an important topic in the theoretical foundations of deep learning. The alignment phenomenon observed in this paper seems to be interesting and could be useful to explain the success of neural networks.
2. The setup considered in this paper is a nice one to study feature learning of neural networks.

**Weaknesses:**

1. I do not see the motivation to study the alignment and local elasticity. They do seem to have some relation to feature learning, but the paper fails to show why such feature learning is important for better generalization. I think either designing experiments to explicitly show that improving alignment leads to better generalization or analytically building connections between generalization and alignment can be useful.

2. The machinery used in this paper is mainly developed by Ba et al. (2022). I do not see interesting new techniques. Please correct me if I miss anything.

3.  The writing of the paper needs improvement. I had a difficult time understanding the paper.
- For example, definitions 1.1 and 1.2 do not seem to be sufficiently formal.
- Also, why do theorem 1.3 and 1.4 hold with probability? Where does the randomness come from?
- How to see the improvement of alignment through the arguments in Theorem 5.3? The same in Theorem 5.4.

There are many typos and missing periods. I will list a few below, but there are many more.
- Line 050 11unseen
- Line 168 extra period after $Hn(x)$
- Line 202 datum
- Line 202 the first y is a column vector but the second y is a row vector.
- Line 226 missing period
- Line 298 no space between let and $\alpha$
- Line 309, 311,315 missing period
- …

**Questions:**

Please see the weaknesses.

---

> ### Author Response · Authors · 2024-11-22
>
> Thank you for your review. We respond to the weaknesses you pointed out as follows.
>
> W1 -> We support our motivation through experiments measuring the clustering performance (recall@1), which implies the prediction on unseen data is available through $\beta^\top \mu$.
>
> We observe that the alignment increases as $\beta^\top \mu$ increases in two-layer networks, and multi-class networks also demonstrate a positive correlation between alignment and recall@1 with low p-value.
>
> Explanations on intuition including margin-based method is also added in Introduction section.
>
> W2 -> The assumption over data distribution makes our proof unique to previous works.
>
> Existing studies analyzing two-layer networks [Ba et al. (2022), Moniri et al. (2024)] assume standard gaussian distribution over their data distributions, with their analysis scope on regression problems.
>
> On the other hand, we generalize such assumption into non centered Sub-Gaussian distribution, which is applicable to classification problems thanks to its finite supports representable as separable class distributions.
>
> At proof technique level, we apply approximation methods by  Ba et al. (2022) and Moniri et al. (2024) but with new assumptions applicable to a classification problem. Especially, we expand Gaussian distribution assumption on data into Sub-Gaussian and substitute Lipschitz teacher function into binary class labels. Such differences on conditions derive the general statements to previous works, but the definitions and the proof between statements require novel delineations. For instance, the definition of $\beta$ exploits the opposite signs of $y_1$ and $y_2$ to compose different gradient signals in the two-label classification, and we utilize SVD to acquire the norm bound of arbitrary labels without Lipschitz assumption.
>
> W3 -> For Theorems 5.3 and 5.4, we modify their symbols and clarify the assumptions, which are now readable without appendices.
>
> Additionally, we cleared the randomness and made Theorem 5.3 and 5.4 deterministic statements, so as informal statements Thm 1.3, 1.4.
>
> Through Sections 5.3 and 5.4, we observe that as the similarity \( \beta^\top \mu \) between the train data and unseen data increases, the metrics of alignment and LE also increase. To clarify this further, we have added a discussion section.
>
> Additional experiments also verify the influence of $\beta^\top \mu$ to alignment and LE in both two-layer and multi-layer networks.

---

### Official Review · Reviewer_aHJs · 2024-11-03

**Soundness:** 2
**Presentation:** 1
**Contribution:** 1
**Rating:** 3
**Confidence:** 4

**Summary:**

This paper studies feature learning in the conjugate kernel model with a focus on two phenomena called "alignment" and "local elasticity". The main observation is that, after one step of training, a distribution similar to the training data shows improved feature alignment and "stronger" elasticity.

**Strengths:**

Understanding scaling limits of feature learning is clearly an important theoretical direction so the topic is timely and worthwhile. The model setting (conjugate kernel) is an interesting one for questions of feature learning and remains under explored, in my opinion. The paper also converts the regression setting of Ba et al to a classification task, which could highlight important differences between the two the problems.

**Weaknesses:**

- Throughout, the assumptions, theoretical setting, and analysis are presented in extremely unclearly. Notation frequently appears without introduction, for example, in Sec. 1.1, no assumptions are stated about the task, training data, objective, network used, etc. The distributions $c_3$ and $c_4$ appear without introduction (though they appear in Fig. 1) and why the subscripts 3 and 4 are used is never explained. Theorems 1.3 and 1.4 are stated without any real introduction and appear more to be more like approximate asymptotic calculations than anything close to a theorem.
- The lack of clarity extends to the writing and discussion of the related works. The alignment and local elasticity section of the related work is especially difficult to parse. What does it mean: "it is theoretically known that the Alignment structure corresponds to the situation where features from training data are aligned around labels"? Statements like this appear throughout the section and do not help to contextualize the results presented in the paper.
- I was not able to identify what the author(s) consider to be the main contribution. The informal theorems essentially state that the features align with the feature vector that results from optimization using unregularized least squares regression.
- The formal theorem statements are arguably even less precise than the informal ones. For example, in 5.3, essentially nothing is directly defined and all the variable names are deferred to the appendix. At this point, it's not even useful to include it because it's simply uninterpretable.
- The most important weakness is the lack of rigor. Assumptions are not stated, even in the more detailed theoretical results. For example, in Theorem 5.3, which is essentially just a calculation, none of the notation is even introduced in the main text, and the details given in Appendix H have odd formatting and typos.

**Questions:**

- What is the main theoretical contribution?
- Can the authors state the results in a way that more clearly delineates the essential assumptions?

---

> ### Author Response · Authors · 2024-11-22
>
> We have improved the paper based on your comments. We would appreciate it if you could refer to the revised version. Additionally, below are the contributions and meaning of our informal theorem we are asserting.
>
> -> The main contribution is that we can predict the feature structure after training, expressed through metrics such as alignment and LE, based on information from the data before training. We will clarify this in the introduction.
>
> The key idea of the informal theorem is that the alignment and LE of the feature vectors learned using MSE can be expressed in terms of the statistics of the training and test data, namely \( \mu \) and \( \beta \). We have rewritten this more clearly. Apologies for any confusion caused.

---

### Official Review · Reviewer_phYQ · 2024-11-03

**Soundness:** 2
**Presentation:** 1
**Contribution:** 2
**Rating:** 5
**Confidence:** 3

**Summary:**

This work studies the change in the feature of a two layer network after one step of gradient descent. They showed that both alignment and local-elasticity emerges after one step of training

**Strengths:**

Both alignment and local elasticity are interesting concepts in deep learning and this work studies how they emerge during training.

The math is solid and the theoretical results are sufficiently clean that they can be understood rather easily

**Weaknesses:**

I think there are a few weaknesses.

1. The theorems are not sufficiently clear. For example, the result seems to only deal with one step of gradient descent after initialization, and neither theorem statements made this clear

2. The problem setting only deals with classification and this should be made clear in the abstract

3. The work claims to discover the relationship/connection between alignment and local elasticity in many places, but I do not see or understand what exactly is this connection. The conclusion I can draw from the paper is that they both occur after 1 step of GD, but co-occurrence does not feel like a strong enough connection

4. The problem setting is also very unclear. Judging from the problem setting 3.2, is only the first layer being trained? Perhaps it is, because training the second layer or not is irrelevant if the model is only trained for one step

5. After all, I think the problem setting is too restricted (two layer net, classification task, one step of GD) and the message is not sufficiently strong or interesting

Minor:
The figures are not quite visible. The fonts should be a lot larger

**Questions:**

Can the alignment part of the result be related to the alignment phenomena discussed in https://arxiv.org/abs/2410.03006？

---

> ### Author Response · Authors · 2024-11-22
>
> Thank you for the helpful review that contributed to improving the paper. We are now addressing your questions.
>
> We have clarified the theorem, particularly providing a clearer that “we provide analysis of the features after one step of training”. Additionally, we explicitly stated that this study focuses solely on classification problems. Apologies for any confusion caused.
>
> We have claimed that after one step of GD, alignment and LE are increased by the same variable, \( \beta^\top \mu \). To express this more precisely, we will replace the term "connection" with the statement "both phenomena occur after one step." Furthermore, in the final "further works" section, we plan to mention the need to explore the connection of these two phenomena as events that occur after one step.
>
> We have made it clear in the problem setup that only the first layer is trained, while the initialized second layer affects the gradient. Reflecting this, we revised the overall description of the problem setup and did not depict the second layer before and after training in Fig. 1, thus enhancing clarity.
>
> Incorporating your comment, we experimented with the CARS196 and CUB200 datasets using a multi-layer network after sufficient gradient steps(20 epoch), and confirmed that \( \beta^\top \mu \), alignment, and LE are strongly rank correlated with Kendall’s W statistic and move together.
>
>
> Q. Can the alignment part of the result be related to the alignment phenomena discussed in https://arxiv.org/abs/2410.03006？
> -> This is a paper that we have not yet reviewed. It has provided us with a great deal of inspiration, and we are grateful for it. The paper above argues that alignment between latent representations, weights, and gradients during training enables the neural network to naturally learn compact representations, supporting this claim through covariance and correlation analysis. In contrast, we focus on the alignment between representations within the same class (intra-class representations) and have discovered conditions under which alignment occurs more strongly in a two-layer neural network.
>
> We believe there is a strong connection between these two papers when considering recent trends in Neural Collapse (NC) research. NC suggests that in a classifier, variability collapse—alignment between representations—occurs, as well as convergence to self-duality, meaning alignment between representations and weights. Our paper addresses the alignment related to variability collapse in a two-layer structure that can learn features, whereas the above paper extends the analysis by including gradients in addition to the alignment between representations and weights, as seen in NC. This distinction is significant, and we will incorporate this correlation in our paper.

---

> > ### Comment · Reviewer_phYQ · 2024-11-25
> > **Reply and thoughts**
> >
> > Thanks for the rebuttal. I thought about it for while but I decided to keep my initial rating.
> >
> > I cannot help but feel that the significance is not quite strong -- this is a personal opinion of course. The problem is still what I mentioned in the first place: two layer net, classification task, one step of GD. Even if the authors proved a very strong result in this setting, it is still limited by these problems. On top of these limitations, I think the problem is also that the result reached is not a strong one...
> >
> > I do not mind accepting this paper, but I think someone else would have to vouch for its significance. Also, I hope that the authors are not disencouraged by this review. I think the authors has found something very curious but its significance at this point does not feel so convincing

---

> ### Author Response · Authors · 2024-11-26
>
> We deeply appreciate your warm encouragement and thoughtful feedback. To address your concerns regarding the number of layers, steps, and tasks, we have conducted extensive validations of our theory using ResNet architectures with various depths and a wide range of steps. As detailed in the revised manuscript, we have confirmed that our theoretical predictions continue to hold for neural networks trained on real-world data.
>
> That said, we find your concern about limitations regarding the number of layers, steps, and tasks still exist. To provide additional clarity and to better convey our intentions, we provide additional background information and a concise theoretical analysis.
>
> We hope these additions address your concerns and help you view the manuscript more favorably. If there are any other aspects you believe require further clarification or additional analysis, please do not hesitate to let us know.
>
> Additionally, we have been informed by ICLR that the discussion period has been extended, and we look forward to engaging in meaningful discussions to address any further points you may have. Thank you once again for your time and constructive evaluation.
>
> ## Response to Concerns Regarding Two-Layer Networks: Neural Network Theories Based on the Two-Layer Assumption
>
> Several prior studies have effectively utilized the feature extractor assumption same to our, to interpret phenomena observed in practical neural networks. For example, Damian et al., 2022 analyzed the efficient generalization and transfer performance of neural networks, while [1] used this framework as a tool to study robustness to input distribution shifts. Similarly, [2] employed it to analyze out-of-distribution inputs, and [3] utilized it to investigate adversarial robustness. These studies focused on understanding phenomena of neural network representations, particularly the hidden representations allowing them to model and explain behaviors observed in practical deep learning scenarios. Based on this body of work, we argue that the assumption of a two-layer network capable of learning hidden representations is a reasonable and effective framework for analyzing neural networks without significant loss of generality.
>
> [1] Nilesh Tripuraneni, Ben Adlam, and Jeffrey Pennington. Overparameterization improves robustness
> to covariate shift in high dimensions. In A. Beygelzimer, Y. Dauphin, P. Liang, and J. Wortman
> Vaughan (eds.), Advances in Neural Information Processing Systems, 2021. URL https://
> [openreview.net/forum?id=PxMfDdPnTfV](http://openreview.net/forum?id=PxMfDdPnTfV).
>
> [2] Donghwan Lee, Behrad Moniri, Xinmeng Huang, Edgar Dobriban, and Hamed Hassani. Demysti-
> fying disagreement-on-the-line in high dimensions, 2023. URL https://arxiv.org/abs/
> 2301.13371.
>
> [3] Simone Bombari, Shayan Kiyani, and Marco Mondelli. Beyond the universal law of robustness:
> Sharper laws for random features and neural tangent kernels, 2023. URL [https://arxiv](https://arxiv/).
> org/abs/2302.01629.

---

> ### Author Response · Authors · 2024-11-26
>
> ## Concerns Regarding One-Step Learning: Theoretical Justification of the One-Step Learning Assumption and Discussion of the Extensibility of Our Analysis
>
> As you pointed out, the assumption that a neural network learns in a single step may seem restrictive. However, as mentioned by Ba et al., 2022, numerous studies on the early phase of training in deep learning ([4,5,6,7]) suggest that neural networks are significantly influenced by their initial learning stages. Moreover, it has been theoretically established that a sufficiently large single gradient step can sharply reduce the training loss ([8]), enable the learning of task-relevant features ([9,10]), and move the network away from its initialization ([11]). Based on these findings, we argue that analyzing feature learning under the one-step assumption provides meaningful insights into neural network behavior.
> At the same time, Dandi et al., 2023 analyzed the staircase property of neural networks in the context of multi-step two-layer neural networks with a large (non-constant) learning rate, similar to our setting. However, such an analysis requires delving into the dynamics of the gradual expansion of the feature subspace, which is beyond the scope of this paper due to volume constraints. We acknowledge the importance of this direction and plan to address it in future work.
>
> [4] Aditya Golatkar, Alessandro Achille, and Stefano Soatto. Time matters in regularizing deep net-
> works: Weight decay and data augmentation affect early learning dynamics, matter little near
> convergence, 2019. URL https://arxiv.org/abs/1905.13277.
>
> [5] Guillaume Leclerc and Aleksander Madry. The two regimes of deep network training, 2020. URL
> https://arxiv.org/abs/2002.10376.
>
> [6] Scott Pesme, Loucas Pillaud-Vivien, and Nicolas Flammarion. Implicit bias of sgd for diagonal
> linear networks: a provable benefit of stochasticity, 2021. URL https://arxiv.org/abs/
> 2106.09524.
>
> [7] Stanislav Fort, Gintare Karolina Dziugaite, Mansheej Paul, Sepideh Kharaghani, Daniel M. Roy,
> and Surya Ganguli. Deep learning versus kernel learning: an empirical study of loss landscape
> geometry and the time evolution of the neural tangent kernel, 2020. URL [https://arxiv](https://arxiv/).
> org/abs/2010.15110.
>
> [8] Niladri S. Chatterji, Philip M. Long, and Peter L. Bartlett. When does gradient descent with logistic
> loss find interpolating two-layer networks?, 2021. URL https://arxiv.org/abs/2012.
> 02409.
>
> [9] Amit Daniely and Eran Malach. Learning parities with neural networks, 2020. URL https:
> [//arxiv.org/abs/2002.07400](https://arxiv.org/abs/2002.07400).
>
> [10] Spencer Frei, Niladri S. Chatterji, and Peter L. Bartlett. Random feature amplification: Feature
> learning and generalization in neural networks, 2023. URL https://arxiv.org/abs/
> 2202.07626.
>
> [11] Karl Hajjar, L´ena¨ıc Chizat, and Christophe Giraud. Training integrable parameterizations of deep
> neural networks in the infinite-width limit, 2021. URL https://arxiv.org/abs/2110.
> 15596.

---

> ### Author Response · Authors · 2024-11-26
>
> ## Concerns Regarding Classification Tasks: Extend Classification Settings to Regression Settings.
>
> We chose a classification setup to analyze network learning in settings where classes are distinct. However, our analysis is not limited to classification tasks alone.
> To address your concerns, we have prepared additional clarification. Inspired by works like Ba et al., 2022, we will incorporate a setting that reflects an regression form to demonstrate that our proof techniques can straightforwardly extend to scenarios involving regression setting. This straightforward adaptability is possible because our analysis applies to any loss or model that satisfies the condition of Proposition 3.1 and Lemma 3.2 in the main text. We argue that this is a key aspect showcasing the extensibility of our study, and we will include this discussion in the Appendix.
>
> Under all the assumptions stated in our paper, we define a new random variable $a \sim \mathcal{N}(0, \frac{1}{N} I)$, and modify the assumptions from Ba et al. 2022 for regression by replacing the centered Gaussian assumption with a non-centered sub-Gaussian assumption, leading to the following problem setup:
>
> $$
> x_i \sim \text{SG} \text{ s.t. } \mathbb{E}[x_i] \neq 0, \quad y_i = f^*(x_i) + \epsilon_i, \quad \epsilon_i \sim \mathcal{N}(0, \sigma_\epsilon^2), \quad f^* \text{ is a Lipschitz function, and } \sqrt{\mathbb{E}_x[f^{*2}]} = \Theta(1).
> $$
>
> In the above setup, we define the loss as follows:
>
> $\theta = \{ W, a\}$, $L(X, y; \theta) = \frac{1}{2n} ||y - \sigma(XW)a||^2$
>
> $G = - \frac{\partial L}{\partial W} = -\frac{1}{ n} \Bigg[X^\top \bigg[ \big(\frac{1}{\sqrt{N}} (\frac{1}{\sqrt{N}} \sigma(XW) a - y) a^\top \big) \odot \sigma'(XW) \bigg] \Bigg]$
>
> In this case, if we define $\alpha = a$ as opposed to the main text, $\alpha$ becomes a Gaussian with zero mean and variance halved, so it follows the same bound structure.
>
> Specifically, based on the sub-Gaussian bound results in Section D, $\mathbb{A}$ can be bounded using the second equation of Lemma 14(i) from Ba et al. 2022 and the fact that  $\mathbb{A}$ is rank-1, so $\lVert A \rVert = \lVert A\rVert_F$.  For $\mathbb{B}$, we can remove $||a_2||_\infty$ from equation 20 in our Lemma E.1 proof. For $\mathbb{C}$, by removing $a_2 a_2^T$ from equation 29, we obtain the same bounds as the previous results.
>
> In conclusion, these three bounds satisfy our Proposition 3.1 under the same conditions, and the same conclusion holds even outside of classification tasks. In this case, $\beta$ is defined as in the previous setting.
>
> To explain in the context of previous research, the original paper on Local Elasticity (He & Su, 2019) also analyzes the difference between linear classifiers and neural networks in the setting of a "binary classifier." We were greatly motivated by He & Su, 2019 in formulating our analysis setting, and we would like to clarify that our classification setup assumption is simply the same as the original setting and not an arbitrary scope limitation.
>
> Furthermore, we would like to emphasize again that one of the key contributions of our work is demonstrating that, based on our assumptions, the class-input similarity in He & Su, 2019 clearly operates as $|\beta^\top \mu|$. This is also a significant insight, as it highlights a limitation in the work of Dan et al., 2023, who extended LE to regression tasks. While their paper shows that, under certain assumptions, the LE-based similarity metric proposed by He & Su, 2019 can be defined for regression problems, they did not theoretically analyze what factors contribute to increasing elasticity, as seen in Hypothesis 1.1(a).

---

### Official Review · Reviewer_j5DU · 2024-11-05

**Soundness:** 3
**Presentation:** 2
**Contribution:** 3
**Rating:** 6
**Confidence:** 3

**Summary:**

This paper studies the alignment and local elasticity in the feature learning process. Specifically, the authors analyze a two-layer neural network trained in the high dimensional limit and show the expression of alignment and local elasticity in this regime. Based on the theoretical analysis, the authors show that when the training data is closer to the data distribution, the class will have a better alignment and local elasticity.

**Strengths:**

The paper provides a rigorous theoretical foundation for alignment and local elasticity in a two-layer neural network with the explicit formula on both metrics. The analysis is solid and it provides a new perspective on feature learning in neural networks.

**Weaknesses:**

1. The presentation of the theoretical works needs to be improved, see Questions below. Specifically, many results are not well discussed in terms of their implications.

2. Experiments are limited to simulation data only. It would be very helpful to verify if the conclusion still holds when going beyond the simple two-layer neural network settings.

**Questions:**

1. Why do Theorem 5.3 and 5.4 both appear in high probability formulation? The equation for the expectation is deterministic on both sides, what is the exact randomness here?

2. Section 4 seems to be purely technical lemmas and has no direct implication on the main conclusion of the paper, therefore I suggest the authors defer it to the appendix.

3. It is unclear how to conclude Theorem 1.3 and 1.4 based on the formal version of Theorem 5.3 and 5.4. The authors should include more discussion on the relationship between these two results.

4. Is it essential to use two-layer neural networks to conclude the relationship between $\mu^T\beta$ and the alignment and local elasticity? The conclusion is rather simple so I suspect onc may reach the same conclusion even in the linear model.

---

> ### Author Response · Authors · 2024-11-22
>
> Thank you for your comments, we reply your questions and concerns
>
> 1. Experiments are limited to simulation data only. It would be very helpful to verify if the conclusion still holds when going beyond the simple two-layer neural network settings.
> -> We only substitute two-layer networks to pretrained ResNet18 from our theoretic configurations and verify the equivalent results in additional experiments.
>
> 2. Why do Theorem 5.3 and 5.4 both appear in high probability formulation? The equation for the expectation is deterministic on both sides, what is the exact randomness here?
>
> ->Apologies, we are already using the approximated \( F_l \) in this theorem, so it is not a probabilistic technique. We have removed it. Additionally, we have improved the theoretical techniques, so I would appreciate it if you could review it again.
>
>
> 3. Section 4 seems to be purely technical lemmas and has no direct implication on the main conclusion of the paper, therefore I suggest the authors defer it to the appendix.
> -> Except for the parts necessary to understand the final theorem, unnecessary part has been moved to the appendix.
>
> 4. Is it essential to use two-layer neural networks to conclude the relationship between $\beta^\top \mu$ and the alignment and local elasticity? The conclusion is rather simple so I suspect onc may reach the same conclusion even in the linear model.
>
>
> -> We studied on the occurrence of alignment and LE in the context of classification tasks using a two-layer network. This two-layer network is particularly relevant to the practice of using feature extractors in practical settings, especially when features are obtained from a network with a removed task head. While two-layer analysis is not essential, the intention behind this approach exists as modeling feature extraction, and since many previous studies have used the same Conjugate Kernel setting, we can also explain that the CK was used in terms of comparison and extension.

---

### Official Review · Reviewer_Cqo2 · 2024-11-05

**Soundness:** 1
**Presentation:** 1
**Contribution:** 2
**Rating:** 3
**Confidence:** 4

**Summary:**

The paper studies the behavior of two quantities, namely *alignment* and *local elasticity*, in the training of a two-layer neural network via GD. Such quantities are used to characterize the feature learning phenomenon in a classification task: under the training process, they allow the authors to show that features enhance the alignment of samples from evaluation distributions which are similar to the training dataset. The results are validated by a number of numerical experiments.

**Strengths:**

The paper deals with an interesting topic, namely the deformation of the input data structure induced by a neural network and how it correlates with the training dataset as a single-step GD training is performed. The setup is quite generic, as the authors considered a two-layer neural network admitting a possible non-convex loss. As a result, it is found that the 'strength' of the deformation depends on how much the evaluation dataset is similar to the training set. Although this might appear not entirely surprising (the network has indeed been trained specifically to learn the characterizing features of the training dataset, and search afterward for similar features in the input), the paper proposes a quantitative approach to the problem by studying two quantities of interest, namely the alignment and the local elasticity, and show how such quantities indeed depend on the correlation between training database and signal in the input.

**Weaknesses:**

Unfortunately, the quality of the text is very poor, both at the $\mathrm{\LaTeX}$ editing level and in writing. Quantities are often not specified, sentences are sometimes broken (e.g., see lines 329-330, where a sentence reads "Under the assumption that new data is sampled from a multivariate normal distribution, i.e. $x\sim\mathcal N(\mu,\Sigma)$." missing the main clause [??]), notation is not introduced properly and conventions appears to change from one section to another, making the reading very difficult (see below for just a few comments). This lack of clarity and exposition quality results in compromised readability, and is, in my opinion, a major obstacle to the publication of the manuscript. On top of this, the discussion of the main theoretical results is minimal, and at some point confusing: see the *Questions* section where I list a series of points both on the form and on the content.

**Questions:**

*Comments and questions*
- What does "11unseen" mean at line 050? At line 086 $F$ is introduced, but neither the dimension of $x$ nor the dimension of $W$ nor how $\sigma$ acts are clarified: I suppose the authors rely on the (at this point) standard notation in the discussed setup but all of this has to be specified, also because at line 107 $\beta$ appears and it is rescaled by $n\sqrt N$, both unspecified quantities. At some point in the text, the authors write "In case of any confusion of notation, please refer to Appendix C for clarification": although I understand that compactness is required in ICLR, I fear there is a limit to what can be deferred to the Appendix...
- The concept of *Evaluation sample* is referred to in Definition 1.1 without being introduced (its meaning is given at page 4). Similarly, the authors refer to $c_3$ and $c_4$ without giving context: this issue reappears in Definition 1.2. Therein, the definition of local elasticity is also confusing: do the local elasticity conditions depend on the chosen ordered pair $(c_3,c_4)$ as it looks? Despite being on page 2, it is not obvious what the authors are referring to, and their actual goal becomes more transparent only once the reading of the entire manuscript is complete...
- Although informal, statements of Theorems 1.3 and Theorem 1.4 hide, in my opinion, too many details: some of them (for example the range of some running indices, as in Theorem 1.3) can be inferred, but others have to be dug out from the Appendix subsequent pages, such as the (apparently absent, at first) role of the nonlinearity or the $x$ distribution.
- Is the orthogonal decomposition required for $\sigma(x)$ at line 167 the one explicitly stated soon after in terms of Hermite polynomials?
- In line 184, the authors refer to an extension that might open "new avenues for analysis of deep representation of classification". Are they referring to the sub-gaussian assumption? Is it not a very common assumption for asymptotic analysis in particular when a Gaussian equivalence principle is applied or proved? Incidentally, Fig 2's relation with the product similarity to $\beta$ is not clear and its presence here, in my opinion, only adds confusion on the considered setup: it might be worth moving it directly to section 6 where the experiments are discussed.
- At line 202, it seems that the label vector $y$ of each datapoint is a row or column vector depending on the class (?!). Also, according to what is written here, $Y\in\mathbb R^{n\times 2}$, whereas $X\in\mathbb R^{n\times d}$. On the other hand, using a different (!?) notation, $\beta=\frac{1}{n\sqrt N}X^\top y$. The dimension of $\beta$ is not clear. If the authors meant  $\beta=\frac{1}{n\sqrt N}X^\top Y$, then $\beta\in\mathbb R^{d\times 2}$, so that $\beta^\top\mu\in\mathbb R^2$, whereas it seems that $\beta\in\mathbb R^d$.
- The reference to the training algorithm is not clear at line 204 as the loss used has not yet been specified.
- If $X\in\mathbb R^{n\times d}$ as in line 201 and $W\in\mathbb R^{d\times N}$ as in line 224, the product $XW^\top$ in Eq 2 is inconsistent. Assuming it to be $XW$, then $\frac{1}{\sqrt N}\sigma(XW)a_c\in\mathbb R^d$: if $y[c]$ has the same as $y_1$, $y_2$ defined at line 203, then it should be an $n$-dimensional vector, which would create another inconsistency... In Eq 3 another notation is used once again, namely $y[:,c]$ in place of $y[c]$ (?) and possibly $Y[:,c]$?
- Unless I am wrong, in the decomposition of the gradient in Eq 4, in the first two lines $a$ should be replaced by $a_2$: please check.
- The "proof" of Proposition 4.1 might benefit from moving some of the arguments above inside the proof itself or removed, as it essentially says that the Proposition is true... Proof of Lemma 4.3 in the main text, is not a genuine proof (nor a sketch). What $\Theta_{\mathbb P}$ means?
- Isn't Lemma 5.1 a special case of Theorem 5.2 obtained by taking e.g. $k=0$?
- Theorems 5.3 and 5.4 should be the core of the contribution, yet the comment on them is minimal and the expression of the alignment is given in terms of quantities whose definition is postponed in the appendix. Eq 11 is also unintelligible without sorting to the Appendix. It also seems that the result actually depends on the specific instance of the training dataset through $\beta$, as, in my understanding, the authors only average on the evaluation point. In this sense, I would have expected the theory or experiments to be simplified by some concentration result, as the one obtained in the proportional regime for classification tasks, e.g., by [Mignacco et al. (2020)](https://proceedings.mlr.press/v119/mignacco20a.html) or [Loureiro et al. (2021)](https://proceedings.neurips.cc/paper/2021/hash/543e83748234f7cbab21aa0ade66565f-Abstract.html) for the case of convex losses. Could the authors comment on this? Does concentration occur? On a side note, I am not sure if using the letter [מ‎](https://en.wikipedia.org/wiki/Mem) in Theorem 5.3 improves the readability of the manuscript (as the use of ב in Appendix H). The authors might consider the use of Latin and Greek letters, but this is, of course, up to them.
- What do the authors mean by "Considering randomness, we assume that the sole term $\mathscr C_{j,k}$ represents the Alignment"?
- I might have missed it, but the notation introduced in line 377 is never used in the main text, and is perhaps inconsistent, as $F_0(x)$ is put equal to both $F_0$ and $F'_0$ (?!).
- The dataset construction described in the *Data generation* paragraph in Section 6 is not very clear, e.g., it is not clear if the hyperplane is orthogonal to $\mu$ and passes through $e\mu$. A few additional lines, e.g., the description in Appendix A, should clarify.
- The appendix is written very poorly and, in some points, looks more like a collection of sketchy notes, rich in typos and poor in comments. In some points, the text is just confusing. For example, in line 890 the authors write "$\mathcal N_{(a,b)}(\mu,\sigma)$ is truncated Gaussian s.t. support of distribution, $\mathrm{supp}(\mathcal N_{(a,b)}(\mu, \sigma^2))\in(a, b)$, which is also known as Sub-Gaussian familly". Aside from typos and inconsistent notation in just one sentence, the definition looks different from the usual one, and indeed a dedicated Lemma D.1 is later introduced to *prove* that truncated Gaussians are subGaussians.
- Labels in the experiment figures are extremely small and almost unreadable.
- What is the role played by the separability condition on the training dataset in the theoretical analysis?
- It is not clear to me if Fig 3 and Fig 4 represent the results of the experiments only or if any quantitative comparison with the theoretical prediction is given. Could the authors clarify this point?
- In the discussion of the Recall @ 1 performance, the authors say that "the second phase provides a dataset which effectively supports training for retrieval tasks and is explainable by our theory": could the author clarify this point? From the plots, it looks like the Recall @ 1 performance is not obviously correlated with the Alignment results.
- What is the role of $\psi_1$ and $\psi_2$? Do the authors run multiple instances of their setup to obtain their curves?
- As a final comment, I iterate that on multiple points, the paper is not clear, lacks precision, or presents typos. I recommend a re-reading to improve grammar and fix some evident typos or the bad rendering of formulas and citations (e.g., in line 100 "say" has to be replaced with "Say" and $E_{x_3}$ has to be changed to $\mathbb E_{x_3}$; at line 297 maybe "hence" is better than "where", a space is missing before the definition of $\alpha$, which incidentally has the same name of the learning rate exponent, etc — tracking all typos in the manuscript would be a demanding task). Moreover, the text seems to be written using directly a "Python-like" notation fpr matrices, that it might be made more standard.

---

> ### Author Response · Authors · 2024-11-22
>
> Thank you for the extensive comments, which have greatly improved the paper. Below are the revisions made based on your feedback. We would appreciate it if you could evaluate it once again.
>
> 1. Is the orthogonal decomposition required for  at line 167 the one explicitly stated soon after in terms of Hermite polynomials?
>
> -> Yes, that is correct. To minimize misunderstandings, we have consistently expressed all the expressions using Hermitian decomposition. Orthogonal decomposition can be considered as applying Hermitian decomposition only to the linear components, while treating the rest as a single function.
>
> 2. In line 184, the authors refer to an extension that might open "new avenues for analysis of deep representation of classification". Are they referring to the sub-gaussian assumption? Is it not a very common assumption for asymptotic analysis in particular when a Gaussian equivalence principle is applied or proved?
>
>
> -> The sub-Gaussian assumption on the data, to the best of our knowledge, is introduced here for the first time in the context of classification problems. Existing works analyzing two-layer models, such as Ba et al. (2022) and Moniri et al. (2024), are based on the standard Gaussian data assumption. We extend this by highlighting the importance of the sub-Gaussian class, which, owing to its finite support, can represent multiple separable data distributions.
>
>
> 3.Fig 2's relation with the product similarity to  is not clear and its presence here.
>
> -> I have taken your advice and moved the figure to Section 6. It has helped in separating the specific examples from the theoretical assumptions. Thank you for your valuable suggestion.
>
>
> 4. At line 202, it seems that the label vector y of each datapoint is a row or column vector depending on the class (?!)....
> -> It should be understood that $\beta$ is defined as X^Ty_1. I have made this correction in the main text as well. Additionally, in order to set up the neural network for classification with the MSE loss, we made $y$ as symmetric. Therefore, any form of $y$ is acceptable. We have added a discussion on this point after the main theorem.
>
>
> 5. The reference to the training algorithm is not clear at line 204 as the loss used has not yet been specified.
> -> I have clarified the loss function part. Apologies for the confusion. I have removed the term f_NN(x)and revised the loss to accept weight and train data.
>
> 6. Isn't Lemma 5.1 a special case of Theorem 5.2?
> You are correct in your observation. We have rephrased Lemma 5.1 as a corollary and moved both Lemma 5.1 and Theorem 5.2, which are only used in the intermediate steps of the theoretical proof, to the appendix in order to add experiments and discussions.
>
> 7. In this sense, I would have expected the theory or experiments to be simplified by some concentration result, as the one obtained in the proportional regime for classification tasks, e.g., by Mignacco et al. (2020) or Loureiro et al. (2021) for the case of convex losses. Could the authors comment on this?
>
> -> We do not analyze the concentration in this work, but will discuss it in further works.
> Besides, we averaged Alignment and LE with 30 different seeds in two-layer networks and every experiment show consistent result.
>
> 8. What do the authors mean by "Considering randomness, we assume that the sole term C_jk represents the Alignment"?
>
> -> The revised theorem, which now also take expectation over the parameter. Even in this revision, we demonstrates that $\beta^\top\mu$ is the only term dependent to $\beta$ and $\mu$ as original statement, deriving the equivalent result to our previous claim.
>
>
> 9. Questions about truncated Gaussian (The appendix is written very poorly and, in some points, looks more like a collection of sketchy notes, rich in typos and poor in comments. In some points, the text is just confusing. For example, in line 890 the authors write ... )
>
> -> We intended to emphasize that the truncated Gaussian dataset as a example satisfies our theoretic assumption on Sub-Gaussian condition, which only used for experiments through Lemma D.1. This dataset follows the truncated Gaussian in element-wise. We will details it in the revised main paper to clarify the confusion. Besides, we will greatly appreciate to the reviewer if they let us know the “usual definition” of truncated-Gaussians.
>
>
> 10. What is the role played by the separability condition on the training dataset in the theoretical analysis?
> -> We have confirmed that the assumption of separable sets is not necessary. We will remove it.
>
> 11. It is not clear to me if Fig 3 and Fig 4 represent the results of the experiments only or if any quantitative comparison with the theoretical prediction is given. Could the authors clarify this point?
>
> ->We will rewrite it more clearly. We have only presented the experimental results.

---

> ### Author Response · Authors · 2024-11-22
>
> 12. In the discussion of the Recall @ 1 performance, the authors say that "the second phase provides a dataset which effectively supports training for retrieval tasks and is explainable by our theory": could the author clarify this point? From the plots, it looks like the Recall @ 1 performance is not obviously correlated with the Alignment results.
>
> To avoid confusion, we added a new dataset, which does not change the L2 distance between the two training classes, which is different to original dataset. This allowed us to obtain experimental results that eliminate the need for phase analysis in the recall@1 discussion within the two-layer setup. We found that as $ |\beta^\top \mu| $ increases, the recall@1 performance after training improves, while the recall@1 performance based on the initialized features remains unchanged regardless of $ |\beta^\top \mu| $. Based on this, we revised the explanation in the main text. Additionally, we conducted practical metric learning experiments using ResNet50 as the backbone and confirmed the positive correlation between recall@1 and alignment.
>
>
> 13. what is the role of \psi_1 and \psi_2 ?
> -> This is a constant needed to define the proportional regime. The proportional regime is necessary to bound the gradient decomposition. The value itself is not important for the purposes of this paper.

---

> > ### Comment · Reviewer_Cqo2 · 2024-12-02
> >
> > I would like to thank the authors for going through their manuscript and providing a new version, which has been very largely rewritten with respect to the original submission, improving on the first, problematic version.
> >
> > Some important aspects of the submission still appear puzzling to me. For example, the authors do not discuss possible concentration phenomena (which are central in this kind of asymptotic analysis) both from the static (see the aforementioned works in my report) and in the dynamics (a recent reference, which I see has been pointed out by another referee as well, might be a work by [Cui et al. (2024)](https://arxiv.org/pdf/2402.04980) but also the cited [Dandi et al. (2023)](https://arxiv.org/abs/2305.18270): however, a number of works focused on the characterization of the dynamics of "order parameters" in one gradient step in two-layer neural networks). In these contributions, the tracking of quantities like the overlap between the signal and the estimator (e.g., what the authors call $\beta^\top\mu$) or the estimator and the noise (e.g., what is called here $\beta^\top\Sigma\beta$) are crucial. Also, the role of what is called here $\psi_1$ and $\psi_2$ and of the learning rate is very nontrivial. I think a discussion on these results cannot be avoided: concentration (or the lack of it) is one of the underlying features of such high-dimensional asymptotic analyses, that benefits from a low-dimensional description of high-dimensional models. I tend to keep therefore my score as previously assigned. As commented above, the manuscript might benefit from a deep revision that I understand is not feasible in the short time window of a rebuttal discussion.
> >
> > Incidentally, with respect to the subGaussian paragraph, I was not referring to possible "usual definitions" of truncated Gaussians, but instead to the fact that, from the text, *it looked like sub-Gaussian distributions are defined through truncated Gaussians*, which is not the usual definition but it was a problem of exposition, I believe.

---

### Official Review · Reviewer_VPAi · 2024-11-07

**Soundness:** 1
**Presentation:** 1
**Contribution:** 2
**Rating:** 3
**Confidence:** 3

**Summary:**

The manuscript analyzes feature alignment and local elasticity in two layer nretworks trained with a single gradient step in the proportional regime. The authors extend similar proof scheme by (Ba et al. 2022, Moniri et al. 2024) in the context of Sub-Gaussian data and classification problems with two separable classes. They consider data distribution that might differ with respect to the training one; they characterize the effect of the distribution shift on the resulting feature alignment and local elasticity properties. They provide numerical illustrations to support their claims.

**Strengths:**

The main strength of this paper stands in the nice research idea. There have been different works in theoretical machine learning dealing with learned representations in two-layer networks trained with one step of gradient; this manuscript extends findings in the literature introducing novel data distribution and learning setting.

**Weaknesses:**

The main weakness of this work is the presentation of the manuscript. In many places the authors try to give intuition behind their results; although this is certainly welcome, the resulting effect is a disordered exposition in the main text that needs to be enhanced. The paper heavily relies on techniques already proven in other works, this fact induces some concern on the clear elements of novelty of this submission. The numerical illustrations are far from being clear.

**Questions:**

The main concern on my side is the clarity of the exposition, below I will try to give some pointers for the authors.

- The authors, in the first part of the main text, try to provide intuition behind their results. However, the resulting effect is a higher confusion. Figure 1 is far from being clear, and I struggled to understand many parts of this plot. For example: the origin is present only in the left panel; (a,b,c,d) are training data from the legend but this is not clear from the picture; $(a_1,a_2)$ are not defined in the central panel; the sphere and their transormed mapping are not defined; what are the grey arrows supposed to mean in the right panel?

- In page 2 the math is not sound. One needs to provide a mathematical consistent definition; for example: $c_3, c_4$ are not clearly specified in Defs (1.1,1.2); the indices are not specified in the sum of Theorems (1.3,1.4).

- The concepts of Local Elasticity and Alignment structure are firstly defined in these definitions, and reported by citing related papers. There is never an explanation what these phenomena really are. The explanations associated are often unsatisfactory; for example: "Our analysis implicates that Alignment and Local Elasticity have strong relevance with how close ($\beta^T \mu$) the data distributions are to the training samples". It is not clear what is meant here.

- Similarly to the above, the explanation for the CK framework is never provided and the authors refer only to related works.

- A large part of the technical results should be moved to the appendix in order to provide more space for the intuition building. The results are largely extension of previous techniques to different settings, therefore the authors should focus more on the discussion of their results. All the lemmas between Page 6 and 8 could be moved to the appendix in my opinion for enhancing the discussion in the main text.

- The related works section should be expaned. I would advice to compact the short section in Page 3 with the introduction to provide a more broad view on the field.


Regarding the technical part of the contribution, below are few pointers.

- How crucial is the learning rate assumption scaling? [1] extended (Moniri et al. 2024) to the maximal learning rate scaling regime showing the presence of fully non-linear equivalent map. Also heuristic arguments would be welcome on this point.

[1] Asymptotics of feature learning in two-layer networks after one gradient-step, Cui et al. 2024.

- The numerical illustrations need to be changed. Figures (3,4,5) are not understandable as the font size is too small.

- It would be interesting to observe distribution shift that are not limited to $e\mu$ presented in this manuscript. Can the authors comment on this point?

- I do not understand the training procedure detailed in Page 5. What does it mean to have a subscript $c$ for the second layer $a_c$?

Other minor points below.
- Abstract: "recall@1"; Page 1: "Enable deep learning" -> "Enable deep learning methods"; Page 1: "1 1unseen"; Page 4: recall what is $\beta$ for the reader.

**Details Of Ethics Concerns:**

N/A.

---

> ### Author Response · Authors · 2024-11-22
>
> Thank you for your valuable comments. we reply your questions and concerns.
>
> The paper heavily relies on techniques already proven in other works, this fact induces some concern on the clear elements of novelty of this submission.
> -> Though our work depends on two-layer networks analysis suggested by  Ba et al. (2022), it still has a novelty on expanding the problem domain into classification and investigating the emergence condition of alignment and LE theoretically.
>
> At a proof technique level, we employ the approximation methods by  Ba et al. (2022) and Moniri et al. (2024) but with new assumptions applicable to a classification problem. Especially, we expand Gaussian distribution assumption on data into Sub-Gaussian and substitute Lipschitz teacher function into binary class labels. Such differences on conditions derive the general statements to previous works, but the definitions and the proof between statements require novel delineations. For instance, the definition of $\beta$ exploits the opposite signs of $y_1$ and $y_2$ to compose different gradient signals in the two-label classification, and we utilize SVD to acquire the norm bound of arbitrary labels without Lipschitz assumption.
>
>
> For Figure 1
>
> We reflect your comments by designating the origin on Figure 1 and add explanation about the train datasets and evaluation data “a, b, c, d.” The second layer weights are omitted from the Figure 1, which do not engage with the gradient analysis.
>
>
> For definition of CK
> We replace “CK” in Line 76 to “neural network feature” and specify “Conjugate Kernel” in Line 82 instead of its abbreviation.
>
> Organization of paper
> As you suggested, we supplement additional experiments to verify the theoretic relation between  $|\beta^\top \mu|$, Alignment, and LE in two-layer networks. We also conduct the analysis on practical neural networks such as ResNet with the mitigated assumption. The lemmas in Page 6 and 8 except essential ones to understand the main paper are moved to appendices.
>
> discussion of Question: How crucial is the learning rate assumption scaling?
>
> Our analysis follows the framework by Moniri et al. (2024) which determines the interval of learning rate by parameter $l$, and it is related to the degree of Hermite expansion in our work. Both Alignment and LE are expanded by polynomials of $|\beta^\top \mu|$, and the max degree of the polynomial is determined by the aforementioned $l$. Thus the learning rate grows as the $l$ grows by the framework, and the measurements of LE and Alignment is representable in the polynomials with the bigger degree. It matches the intuition that the loss is minimized more as the feature is learned with the larger learning rate, resulting features are updated and clustered strongly.
>
>
> The second layer weight $a_c$
> Unlike previous studies requiring a single output layer for regression, two classifier heads compose the two-class classifier networks in our work.
>
>
>
> Thank you

---

> > ### Comment · Reviewer_VPAi · 2024-11-26
> >
> > I warmly thank the authors for their efforts in addressing my concerns. After a careful read of the other reviewers' comments, I would like to keep my original score.

---

> ### Author Response · Authors · 2024-11-26
>
> Thank you so much for taking the time to carefully reconsider your evaluation and for your thoughtful feedback. We truly appreciate your engagement with the paper.
> In response to the concerns raised, we have prepared thorough answers tailored to each reviewer’s comments, and we are currently awaiting responses from the others.
> As noted by ICLR, we are fortunate to have an extended discussion period, which provides us with an excellent opportunity to further address any remaining questions or concerns you may have. Over the coming period, We are fully committed to clarifying any points of uncertainty and ensuring that our paper meets your expectations. We sincerely hope that this process will not only lead to a more favorable evaluation from you, but also allow us to further enhance the manuscript.
> Given the extended timeframe, We would greatly appreciate it if you could share any specific issues or concerns you still have regarding the paper—separate from the feedback of other reviewers. This will allow us to provide you with the clearest possible explanations and make further improvements, with the goal of increasing your satisfaction with our work.
> Once again, We want to express our deep gratitude for your valuable contribution to the paper. We look forward to continuing this discussion and developing this paper from your insight.
> Warm regards,

---

### Author Response · Authors · 2024-11-22

We have revised the manuscript based on feedback regarding typos and the overall structure provided by all reviewers. We deeply appreciate your extensive comments and suggestions, which have significantly improved the quality of our paper. We hope you will review the revised manuscript and provide further evaluation. To maintain readability, we did not respond individually to every comment. Instead, we have addressed only questions requiring clarification and acknowledged that your feedback was correct and appropriately implemented elsewhere. For general comments that apply broadly, we have provided an official response in the common comment section.

We reorganized the manuscript’s overall structure. Specifically, we rewrote the theorems to use only symbols defined within the main paper to enhance readability. We combined and enriched the Introduction and Related Works sections to better provide context and background for the study. To ensure clarity, we explicitly defined our metrics as the \textit{Alignment Score} and the \textit{Elasticity Score} and categorized the phenomena as \textit{Alignment} and \textit{Local Elasticity}. Informal explanations of Theorems 1.3 and 1.4 were modified, replacing ambiguous equations with clear textual descriptions. Additionally, we increased the font sizes in all figures for improved readability. We also added experimental results for a two-layer network to validate the theory and included additional experiments in practical settings using ResNet.

Based on your comments regarding the randomness in network parameters in Theorems 5.3 and 5.4, we revised our approach by taking the expectation over neural networks. As a result, the measurement of \textit{Alignment} and \textit{Local Elasticity} now depends on $|\beta^\top \mu|$ instead of $\beta^\top \mu$. This revision aligns with our original claims while also incorporating the symmetry in the definition of $\beta$, providing an improved result. Please refer to the revised paper for further details.

Thank you once again for your invaluable input. We look forward to your feedback on the revised manuscript.

---

### Meta-Review · Area_Chair_7JLB · 2024-12-20

**Metareview:**

The manuscript investigates alignment and local elasticity in two-layer neural networks, analyzing their emergence under train-unseen similarity conditions. While the theoretical framework is grounded in solid mathematics and extends prior work to Sub-Gaussian data and classification tasks, the consensus seem to be that the paper faces significant issues.

Strengths:
-Theoretical analysis of feature learning mechanisms.
-Extension to new data settings.

Weaknesses:
-Poor clarity and organization hinder comprehension.
-Incremental contributions with limited novelty.
-Experimental scope restricted to simple setups, limiting practical relevance.
-Major reviewer concerns about clarity and broader applicability remain unaddressed.

While the paper touches on important topics in feature learning, the incremental contributions, poor clarity, and restricted experimental scope limit its potential impact. I recommend the author to improve their manuscript and resubmit at a later conference with an improved version.

**Additional Comments On Reviewer Discussion:**

The reviewer discussions highlighted recurring issues regarding clarity and scope. Reviewers VPAi and Cqo2 expressed concerns about the lack of novelty and poor exposition, which were only partially addressed in the revised manuscript. Reviewer j5DU acknowledged the theoretical foundation but emphasized the limited generalizability of the results. While the authors attempted to address some technical points, the core concerns—novelty, clarity, and experimental validation—were not resolved to the reviewers’ satisfaction. As such, the overall ratings remained consistent, with most reviewers recommending rejection.

---

### Decision · Program_Chairs · 2025-01-22

Reject